# Generalizable and Composable Multi-Model Embedding Translation

**Beining Yang** [1]   **Yang Cao** [1]

## Abstract

Embedding translation enables interoperability across embedding models, allowing embedding vectors to be reused without costly re-embedding. However, existing methods are typically evaluated under simplified pairwise and in-domain settings and behave as black boxes at inference time, leading to unreliable performance under out-of-distribution (OOD) inputs, multi-model mixing, and composed translations. We analyze embedding translation from a geometric perspective and derive an interpretable error bound that explains systematic error amplification under OOD inputs, mixing and chaining. Building on this, we propose a geometry-aware confidence metric and a Hierarchical Mixture of Experts (`H-MoE`) framework with localized, parameter-efficient adaptation. Following MTEB leaderboard, we conduct large-scale experiments over **10** embedding models and **6** benchmarks across **90** translation directions. `H-MoE` outperforms *every* baseline for *every* model pair over *every* benchmark under OOD scenarios. Furthermore, multi-model mixing and chaining only degrade our performance in Recall@100 by $0.5\% \sim 2.6\%$, compared to $7.2\% \sim 92.3\%$ recall drop by existing methods. Code is available at https://github.com/DBgroup-Edinburgh/embedding-translation.

## 1. Introduction

Embedding models map unstructured objects (*e.g.,* text) into high-dimensional vectors and enable efficient similarity search via *e.g.,* vector databases. In practice, embedding models evolve rapidly (provider switches, fine-tuning, frequent upgrades), yet different models induce incompatible vector spaces: a database built with one model cannot be directly queried with vectors from another model without

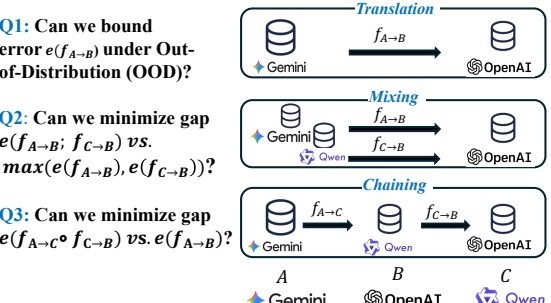

Q1: **Can we bound error** $e(f_{A \to B})$ **under Out-of-Distribution (OOD)?**

Q2: **Can we minimize gap** $e(f_{A \to B};\ f_{C \to B})$ *vs.* $max(e(f_{A \to B}), e(f_{C \to B}))$?

Q3: **Can we minimize gap** $e(f_{A \to C} \circ f_{C \to B})$ *vs.* $e(f_{A \to B})$?

*Figure 1.* **Cross-model embedding interoperability**

costly re-embedding. *Embedding translation* aims to bridge this gap by learning a function that maps embeddings from one model space to another, enabling cross-model queries.

*Prior art*. To this end, there has been work on *embedding translation*, which learns a function to align one embedding space with another. Earlier studies focus on structure-preserving mappings such as Procrustes Analysis (`PA`) (Wang & Mahadevan, 2008)), Canonical Correlation Analysis (`CCA`) (Artetxe et al., 2016)), and Gromov–Wasserstein learning (`GW`) (Chen et al., 2020)), targeting cross-lingual dictionary induction (Mikolov et al., 2013; Artetxe et al., 2016; Smith et al., 2017; Joulin et al., 2018). The rationale is that different embedding spaces are structurally consistent as they encode the same semantic space of words across languages. Recent studies exploit expressive nonlinear and adversarial architectures, *e.g.,* Embedding Converter (`EmbConv`) (Yoon & Arik, 2025) and `Vec2Vec` (Jha et al., 2025), to accommodate complex cross-space mappings that deviate from the global structural consistency.

**Limitations**. There are still several key areas unaddressed.

*Out-of-distribution vectors*. Existing translators are trained and evaluated in a simplified pairwise, in-domain regime: two models embed the same corpus and the translator is fit on a subset, mirroring classic cross-lingual alignment. In practice, embeddings come from heterogeneous and evolving data sources (*e.g.,* different vector databases or domains), each with its own similarity geometry, so test embeddings are often out-of-distribution (OOD) *w.r.t.* training data.

As shown in Fig. 2a, current translators can fail catastrophically under OOD inputs, and scaling up training data does not prevent these drops. While some methods (*e.g.,*

[1]University of Edinburgh, Edinburgh, United Kingdom Correspondence to: Yang Cao <Yang.Cao@ed.ac.uk>. *Proceedings of the $43^{rd}$ International Conference on Machine Learning*, Seoul, South Korea. PMLR 306, 2026. Copyright 2026 by the author(s).

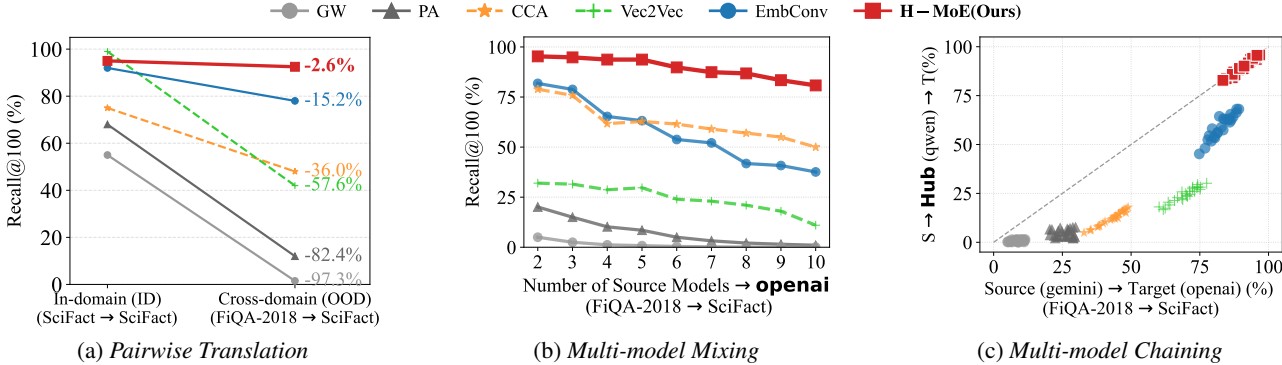

(a) *Pairwise Translation*          (b) *Multi-model Mixing*          (c) *Multi-model Chaining*

*Figure 2.* **Embedding translation scenarios.** (a) *Pairwise Translation*: evaluating translators from `gemini` → `openai` under in-domain (ID) and cross-domain (OOD) shifts; FiQA-2018 → SciFact means we train translator over FiQA-2018 and use it to translate `gemini` embeddings of SciFact to `openai` of SciFact. (b) *Multi-model mixing*: translating embeddings from multiple source models into a unified `openai` target space. (c) *Multi-model chaining*: the x-axis reports direct translation performance (`gemini` → `openai`), while the y-axis shows chained results via the Hub model `qwen`. Detailed settings for each scenario are provided in Appendix A. Results show that our method (HMoE) avoids the significant recall@100 drop in ODD translation, and is substantially more robust in both mixing and chaining scenarios.

`EmbConv`) report OOD results, translation is still treated as a black-box: they provide little insight into how errors arise under shift and, critically, no pre-translation reliability signal. This forces systems to translate every input blindly and only detect failures after observing degraded retrieval.

*Multi-model translation.* Prior work studies embedding translation in isolation for a single source-target pair. In practice, many fine-tuned and frequently updated embedding models coexist ([Muennighoff et al.](), 2023), so translated embeddings from multiple sources may be stored in one target space (*mixing*), and missing pairwise translators often necessitate multi-hop translation steps such as A → Hub → B (*chaining*). As shown in Figures 2b-2c, both patterns can amplify residual errors under composition, yet current methods degrade sharply in both settings.

*Where these settings arise.* The three settings above are not isolated edge cases: they arise naturally in two common deployments. *(i) Staged migration of large or legacy vector databases*, where the historical corpus is already embedded with model $A$ while queries or new documents switch to model $B$ because of provider changes, fine-tuning, or upgrades; full re-embedding is often infeasible (API/operational limits, or raw data being unavailable), so teams collect only a small paired set to train $f_{A \to B}$ while deployment later reaches broader or different corpora, which is exactly the OOD regime. *(ii) Federated or multi-tenant retrieval*, where autonomous vector stores use different embedders, so old and new vectors coexist (mixing), and when a direct translator is unavailable systems must rely on existing $A \to$ Hub and Hub $\to B$ translators (chaining). Importantly, inference-time translation does not require access to the original paired training corpus: our translation-confidence signal only needs a source-side reference set (or its ANN index), which can be shipped with the translator.

**Contributions**. We study embedding translation beyond the standard pairwise, in-domain setting, focusing on the more challenging scenarios in Fig. 1: OOD inputs, multi-model mixing, and multi-hop chaining. Rather than treating translation as black-box regression, we analyze how translation error arises from source-space geometry and how residual errors interact when translations are mixed or chained.

*(1) Geometric OOD error bound (Section 2.2).* We formalize translation as learning a mapping $f_{A \to B}$ between two embedding spaces and study its behavior under distribution shift. Under mild regularity (Lipschitz) assumptions, we derive a pointwise bound for an unseen embedding $\mathbf{x}$: $e(\mathbf{x}) \le (L + L^*)\delta(\mathbf{x}) + e(\mathbf{x_{nn}})$, where $e(\mathbf{x})$ is the translation error of $\mathbf{x}$ relative to an idealized reference translation, $\mathbf{x_{nn}}$ is the nearest training reference point in the source space, and $\delta(\mathbf{x})$ is the corresponding distance-to-reference. The bound separates error into a training term (*i.e.,* local fit around $\mathbf{x_{nn}}$) and a coverage term (*i.e.,* extrapolation controlled by $\delta(\mathbf{x})$), explaining why translators trained i.i.d. can fail systematically on OOD embedding inputs.

*(2) Translation confidence (Section 2.3).* The bound implies that translation is strongly affected by distance-to-reference. We thus develop a translator-independent confidence purely from source-space geometry, enabling systems to estimate *translatability before translation* without access to target-side labels or post-translation evaluation.

*(3) Generalizable translation (Section 3).* Guided by the bound, we design `H-MoE`, a hierarchical mixture-of-experts translator that preserves the expressiveness of nonlinear models while reducing local expansion and extrapolation. `H-MoE` organizes the source embedding space into a hierarchy of regions, each with a specialized translator. It routes test embedding reliably by traversing the hierarchy, and op-

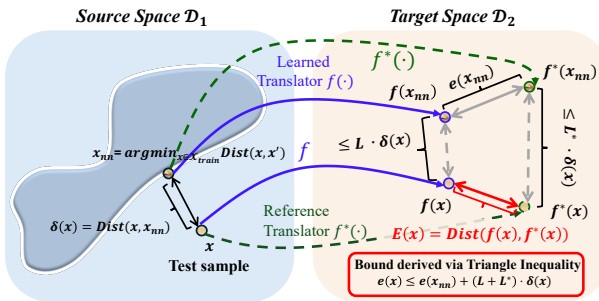

*Figure 3.* **Geometric illustration of the translation error bound.** For a test vector **x** at distance $\delta(\mathbf{x})$ from its nearest training neighbor $\mathbf{x}_{nn}$, we bound the error between the **learned** $f$ and ideal **reference** $f^*$ by decomposing it into the empirical error $e(\mathbf{x}_{nn})$ at the nearest neighbor and the Lipschitz expansion $(L + L^*)\delta(\mathbf{x})$.

timizes parameter-efficiency via Low-Rank Adaptation.

*(4) Multi-model consistency and transitivity (Section 4).* We characterize why multi-model settings are harder than pairwise translation: mixing embeddings from different sources distort cross-source distances when different translators introduce residuals in incompatible directions, and chaining can accumulate error when propagated upstream residuals align with downstream residuals. We address both with a unified geometric principle: regularize residual errors to be *directionally predictable* using principal directions of the target embedding distribution, yielding robust *consistency* for mixing and improved *transitivity* for chaining.

*(5) Library and large-scale evaluation (Section 5).* We implement our approach as a unified library supporting all **90** translation directions among **10** embedding models. We evaluate on **6** BEIR benchmarks (Thakur et al., 2021), covering cross-dataset OOD, multi-model mixing, and chaining. while existing baselines suffer severe degradation, H−MoE maintains stable performance. In particular, we achieve state-of-the-art results under OOD translation and limits degradation under multi-model settings to $0.5\% \sim 2.6\%$, compared to $7.2\% \sim 92.3\%$ for prior methods.

**Conflict of Interest Disclosure.** The authors declare no financial conflicts of interest. Both authors are affiliated solely with the University of Edinburgh, an academic institution. The embedding models evaluated in this work (*e.g.,* openai, gemini, qwen, gritlm) are independent third-party systems with which the authors have no financial or employment relationship, and no proprietary models developed by the authors' institution are evaluated as part of this study.

## 2. Bounding Translation Error

In this section, we derive a pointwise upper bound on embedding-translation error under distribution shift. The bound separates error into (i) a *training* term that captures translator's quality over training set, and (ii) a data *coverage* term

that is dependent on the geometry of source-side dataset.

### 2.1. Preliminaries

Let $\mathcal{O}$ be a universe of objects. Two embedding models $\text{emb}_A$ and $\text{emb}_B$ map each $o \in \mathcal{O}$ to a source embedding $\mathbf{x} := \text{emb}_A(o) \in \mathbb{R}^{d_A}$ and a target embedding $\mathbf{y} := \text{emb}_B(o) \in \mathbb{R}^{d_B}$. Embedding translation learns $f_{A \to B} : \mathbb{R}^{d_A} \to \mathbb{R}^{d_B}$ so that $f_{A \to B}(\text{emb}_A(o))$ is close to $\text{emb}_B(o)$ for the same object $o$, given paired training data $\{(\mathbf{x}_i, \mathbf{y}_i)\}_{i=1}^n$.

Let $\mathcal{O}_{\text{train}}$ and $\mathcal{O}_{\text{test}}$ be training and test collections, and define the corresponding source-embedding sets $\mathcal{X}_{\text{train}} := \{\text{emb}_A(o) \mid o \in \mathcal{O}_{\text{train}}\}$, also referred to as the reference set or pool, and $\mathcal{X}_{\text{test}} := \{\text{emb}_A(o) \mid o \in \mathcal{O}_{\text{test}}\}$ as the testing set. Out-of-distribution (OOD) means $\mathcal{O}_{\text{test}}$ differ substantially from $\mathcal{O}_{\text{train}}$. We use Euclidean distance $\text{Dist}(\mathbf{u}, \mathbf{v}) := \|\mathbf{u} - \mathbf{v}\|_2$ (see Appendix B.1 for cosine variant).

### 2.2. A pointwise upper bound under distribution shift

We posit an (unknown) ideal alignment $f^* : \mathbb{R}^{d_A} \to \mathbb{R}^{d_B}$ that exactly matches training pairs, *i.e.,* $f^*(\text{emb}_A(o)) = \text{emb}_B(o)$ for all $o \in \mathcal{O}_{\text{train}}$. We measure pointwise translation error relative to $f^*$ by $e(\mathbf{x}) := \|f(\mathbf{x}) - f^*(\mathbf{x})\|_2$.

**Distance-to-reference.** Given a test embedding $\mathbf{x} \in \mathcal{X}_{\text{test}}$, let $\mathbf{x}_{\text{nn}}$ denote its nearest neighbor in the source-side training reference set, *i.e.,* $\mathbf{x}_{\text{nn}} := \arg\min_{\mathbf{x}' \in \mathcal{X}_{\text{train}}} \|\mathbf{x} - \mathbf{x}'\|_2$, and define the *distance-to-reference* as $\delta(\mathbf{x}) := \|\mathbf{x} - \mathbf{x}_{\text{nn}}\|_2$.

Intuitively, $\delta(\mathbf{x})$ indicates how much generalization power we would require from $f$ to translate it. When $\delta(\mathbf{x})$ is large, the test point $\mathbf{x}$ lies far from the region covered by the training reference set, so translation requires more extrapolation.

**Lipschitz continuity.** Following the standard regularity condition, we assume that the learned translator $f$ is $L$-Lipschitz and the ideal reference translator $f^*$ is $L^*$-Lipschitz: for all $\mathbf{x}, \mathbf{x}' \in \mathbb{R}^{d_A}$, we have $\|f(\mathbf{x}) - f(\mathbf{x}')\|_2 \leq L\|\mathbf{x} - \mathbf{x}'\|_2$ and $\|f^*(\mathbf{x}) - f^*(\mathbf{x}')\|_2 \leq L^*\|\mathbf{x} - \mathbf{x}'\|_2$. Intuitively, this asserts that perturbations in the source space should not cause arbitrarily large changes in the translated embedding.

**Theorem 1:** *For any* $\mathbf{x} \in \mathcal{X}_{\text{test}}$*, the translation error* $e(\mathbf{x})$ *of* $\mathbf{x}$ *by translator* $f$ *relative to* $f^*$ *is bounded by*

$$e(\mathbf{x}) \leq (L + L^*)\,\delta(\mathbf{x}) + e(\mathbf{x}_{\text{nn}}),$$

*where* $e(\mathbf{x}_{\text{nn}}) := \|f(\mathbf{x}_{\text{nn}}) - f^*(\mathbf{x}_{\text{nn}})\|_2$. ☐

As illustrated in Fig. 3, Theorem 1 follows by anchoring $\mathbf{x}$ to $\mathbf{x}_{\text{nn}}$ and applying the triangle inequality and Lipschitz continuity (see Appendix B for full proof). It makes the decomposition explicit: $e(\mathbf{x}_{\text{nn}})$ is a *training* term (local fit), while $(L + L^*)\delta(\mathbf{x})$ is a *coverage* term that grows with distance-to-reference. In particular, if $\delta(\mathbf{x}) = 0$ the bound reduces to training error; conversely, even with small train-

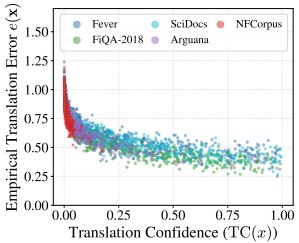

(a) *Fix method (*`H-MoE`*), comparing all datasets to SciFact.*

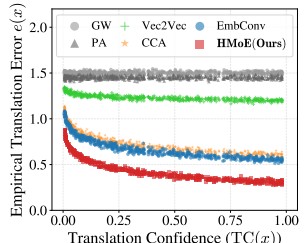

(b) *Fix dataset (FiQA-2018 → SciFact), comparing methods.*

*Figure 4.* **Translation confidence (**`TC`**) vs. empirical translation error** $e(\mathbf{x})$ (`gemini` → `openai`): (a) the concentration of NFCorpus near TC ≈ 0 reflects significant domain divergence from the SciFact samples; (b) on FiQA-2018 → SciFact, the high error of `GW` and `PA` empirically validates the importance of the reconstruction term $e(r(o))$ in Theorem 1, while `H-MoE` achieves lowest $e(x)$.

ing error, large $\delta(\mathbf{x})$ can lead to large OOD translation error.

### 2.3. Translation Confidence Metric (TC(x))

Theorem 1 suggests that the main per-point factor governing OOD translation quality is $\delta(\mathbf{x})$: the remaining terms $L$, $L^*$, and $e(\mathbf{x}_{nn})$ are either global properties of the translators or small after successful training, while $\delta(\mathbf{x})$ varies substantially across test inputs. This motivates a translator-independent per-input metric that decides translatability of a given test vector based solely on source-space geometry.

**Translation confidence**. For any test embedding $\mathbf{x} \in \mathcal{X}_{\text{test}}$, we use its distance-to-reference $\delta(\mathbf{x}) = \min_{\mathbf{x}' \in \mathcal{X}_{\text{train}}} \|\mathbf{x} - \mathbf{x}'\|$ as the core indicator of uncertainty. To ensure the metric is comparable across different datasets and embedding scales, we normalize this distance by the intrinsic scale of the training reference set. We estimate this scale $\sigma_{data}$ as the standard deviation of pairwise distances within the training set: $\sigma_{\text{data}} := \sqrt{\text{Var}_{\mathbf{x}_i,\mathbf{x}_j \in \mathcal{X}_{\text{train}}, i \neq j}\big[\|\mathbf{x}_i - \mathbf{x}_j\|\big]}$.

We then define the *Translation Confidence* (TC) for a query object $o$ with embedding $\mathbf{x} := \text{emb}_A(o)$ as: $\text{TC}(\mathbf{x}) := \exp\left(-\dfrac{\delta(\mathbf{x})}{\sigma_{\text{data}}}\right)$. Note that $\text{TC}(\mathbf{x}) \in (0, 1]$, where higher values indicate better coverage, *i.e.*, the query lies near the training manifold and is expected to be more stable.

As shown in Fig. 4, translation confidence is indeed an accurate indicator of the empirical translation error when translating `gemini` to `openai` embeddings across (a) varying training set (Fig. 4a) and (b) different translators (Fig. 4b).

**Implications**. (1) In practice, $\delta(\mathbf{x})$ can be efficiently computed by querying a fast nearest-neighbor index (*e.g.,* FAISS (Johnson et al., 2021)) built on $\mathcal{X}_{\text{train}}$. The scale $\sigma_{\text{data}}$ is computed once offline (typically using a random subset of pairs) and cached. This makes the confidence score $\text{TC}(\mathbf{x})$ available in real time and does not require any labels

or ground-truth embeddings in the target space.

(2) Because $\text{TC}(\mathbf{x})$ is computed *before* translation, it enables abstention policies such as translating only if $\text{TC}(\mathbf{x})$ is greater than a threshold (determined by the application scenario), otherwise falling back to recomputing the target embedding directly. As we will see shortly in Section 5, $\text{TC}(\mathbf{x})$ correlates strongly with downstream retrieval performance (*e.g.,* Recall@K) under distribution shift, supporting its use as a practical translation quality indicator.

**Scope and calibration.** TC is designed for online reliability assessment under a *fixed* deployed translator and reference pool. For a given pool $\mathcal{X}_{\text{train}}$, $\sigma_{\text{data}}$ is constant, and $\text{TC}(\mathbf{x}) = \exp(-\delta(\mathbf{x})/\sigma_{\text{data}})$ is a monotone transform of $\delta(\mathbf{x})$. Hence, it preserves the ranking of risky inputs, and only the operating threshold needs to be calibrated for each reference pool. Directly comparing raw TC values across reference pools with substantially different scales or cluster structures is therefore not the intended use. For heterogeneous multi-domain pools, we further provide a local-$k$NN normalization variant as a natural extension. Appendix C reports cross-pool evidence showing that the rank correlation between TC and per-vector empirical translation error remains strong as the reference pool is enlarged.

## 3. Hierarchical Mixture-of-Experts Translator

Theorem 1 bounds the translation error at a test embedding $\mathbf{x}$ by a training term $e(\mathbf{x}_{nn})$ and a coverage term $(L + L^*)\delta(\mathbf{x})$. For a fixed training set, $\delta(\mathbf{x})$ is determined by how well the training data covers the source space, and cannot be changed at inference time. Our design focus in this section is thus translator-side: (i) reduce the empirical error on training set and (ii) reduce the expansion factor corresponding to $L$.

**Impact of Lipschitz expansion**. We first examine the impact of the expansion factor on the design of translators. Since computing a certified global Lipschitz constant for a neural network is NP-hard (Fazlyab et al., 2019; Virmaux & Scaman, 2018), we use an empirical Lipschitz proxy following (Gamba et al., 2023). Specifically, for a differentiable translator $f : \mathbb{R}^{d_A} \to \mathbb{R}^{d_B}$, a local empirical Lipschitz constant can be estimated as $\hat{L}_{\text{Jac}} = \max_{\mathbf{x}_i \in \mathcal{X}} \|J_f(\mathbf{x}_i)\|_2$, where $J_f(\mathbf{x})$ is the Jacobian of $f$ and $\|\cdot\|_2$ is the spectral norm. This Lipschitz proxy can be efficiently approximated using power iteration with Jacobian–vector products.

Fig. 5a plots $\hat{L}_{\text{Jac}}(f)$ as we increase the size (and coverage) of the training reference set. Orthogonal Procrustes (`PA`) has constant 1 Lipschitz value because it is an isometry, but its rigid form can underfit cross-model alignment. In contrast, learned nonlinear translators are more expressive, yet their empirical expansion grows with dataset size, indicating a larger effective Lipschitz term that can amplify OOD deviations in Theorem 1. This implies that a single global transla-

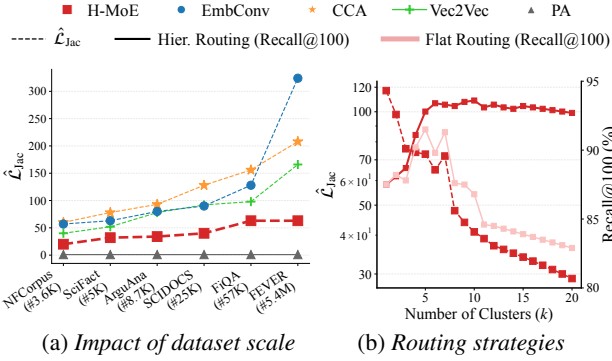

(a) *Impact of dataset scale*  (b) *Routing strategies*

*Figure 5.* **Analysis of empirical Lipschitz constants $\hat{L}_{\mathrm{Jac}}$.** (a) $\hat{L}_{\mathrm{Jac}}$ vs. dataset scales; (b) The impact of query routing (*Flat* vs. *Hierarchical*) on Lipschitz constants and retrieval performance.

tor must accommodate progressively more complex geometry, and thus inevitably getting worse with OOD inputs.

This motivates a key design principle for embedding translation: preserve the expressiveness of nonlinear translators while controlling local expansion by replacing a single monolithic mapping with multiple region-wise translators, each with lower *local* Lipschitz complexity. We then translate a test point $\mathbf{x}$ by routing it to a nearby local translator.

**Local translators and query routing**. We next formalize the idea of local translators. Let $\mathcal{O}_{\mathrm{train}}$ be the training objects, and for each $o \in \mathcal{O}_{\mathrm{train}}$ let $\mathbf{x}_o = \mathrm{emb}_A(o)$ and $\mathbf{y}_o = \mathrm{emb}_B(o)$. We partition the training set into regions (clusters) in the *source* embedding space ($\mathrm{emb}_A$).

Suppose we have a family of experts indexed by $i$, where expert $i$ is trained on a subset $\mathcal{O}_i \subseteq \mathcal{O}_{\mathrm{train}}$ and defines a translator $f_i$. For a test point $\mathbf{x}$, define the expert-specific distance-to-reference $\delta_i(\mathbf{x}) := \min_{o \in \mathcal{O}_i} \|\mathbf{x} - \mathbf{x}_o\|_2$. Repeating the same argument as in Theorem 1 yields an expert-specific bound involving $\delta_i(\mathbf{x})$ and the local expansion behavior of $f_i$. This highlights a concrete routing objective: we would like to apply an expert whose training region is close to the query (small $\delta_i(\mathbf{x})$) and whose mapping is locally non-expansive (small empirical expansion).

However, naive routing introduces its own failure mode: if we always select the single nearest cluster, *i.e.,flat* nearest-centroid routing, small perturbations can flip the chosen expert near cluster boundaries, leading to unstable translations.

**Hierarchical routing**. To make translation stable, we build a hierarchy over the source training embeddings via hierarchical clustering (Pedregosa et al., 2011; Nielsen, 2016). This produces a rooted binary tree with $K$ leaf clusters (a partition of the training data) and $2K - 1$ total nodes. Each node corresponds to a subset of training objects; leaves represent fine regions, while internal nodes are unions of their descendant leaves (coarser regions). We train one expert

translator $f_v$ for each node $v$. At inference time, routing proceeds top-down: we descend the tree while the query is clearly closer to one child than the other; otherwise we stop early and use the current node's (coarser) expert. Unlike learned top-1 gating or flat nearest-centroid routing, which must always commit to a single fine-grained expert and can therefore switch abruptly when a query lies near a cluster boundary, our rule is purely geometry-based on the source space and *backs off* to a coarser expert whenever the child-level decision is ambiguous. Together with the fact that each internal-node expert is trained on the union of its descendant leaves, this guarantees that a boundary query is served by an expert whose training support actually covers it, which empirically gives the stability advantage reported in Fig. 5b.

As shown in Fig. 5b, this reduces translation boundary instability: its retrieval performance improves steadily for small to moderate numbers of clusters, while the naive flat strategy deteriorates when the partition becomes overly fine-grained. Moreover, it also shows that increasing the number of clusters consistently reduces the empirical Lipschitz constant, confirming the effectiveness of localized modeling.

**Hierarchical Mixture-of-Experts Translation**. Guided by the above analysis, we propose a Hierarchical Mixture-of-Experts (H-MoE) embedding translator, using local translator experts and hierarchical routing described above.

Each expert $f_i$ is implemented as a lightweight feed-forward translator and is trained only on samples within its assigned cluster $\mathcal{C}_i$. Concretely, the shared base translator is a 4-layer MLP with SELU activations (the same backbone as EmbConv), and each tree node adds a node-specific LoRA adapter on top, so the additional capacity comes from localized low-rank adaptation rather than from instantiating a full network per expert. The training objective combines a pointwise regression loss with a local structure-preserving loss: $\mathcal{L} = \mathcal{L}_{\mathrm{reg}} + \alpha\,\mathcal{L}_{\mathrm{local}}$. The regression loss enforces pointwise alignment between translated and target embeddings:

$$\mathcal{L}_{\mathrm{reg}} = \sum_{t \in \mathcal{C}_i} \mathrm{Dist}(f_i(\mathrm{emb}_1(t)),\, \mathrm{emb}_2(t))\,.$$

Following EmbConv (Yoon & Arik, 2025), to preserve neighborhood geometry within each localized region, we adopt a local similarity loss that penalizes discrepancies in pairwise distances between a sample $t_1$ and its $m$ nearest neighbors $\mathcal{NN}_m(t_1)$ (with $m = 100$):

$$\mathcal{L}_{\mathrm{local}} = \sum_{t_1 \in \mathcal{C}_i} \sum_{t_2 \in \mathcal{NN}_m(t_1)} \Big| \mathrm{Dist}\big(f_i(\mathrm{emb}_1(t_1)), f_i(\mathrm{emb}_1(t_2))\big) \\ - \mathrm{Dist}\big(\mathrm{emb}_2(t_1), \mathrm{emb}_2(t_2)\big) \Big|.$$

We omit global similarity losses used in prior work (*e.g.,* EmbConv), as each expert operates on a localized manifold where global-scale distance preservation is unnecessary.

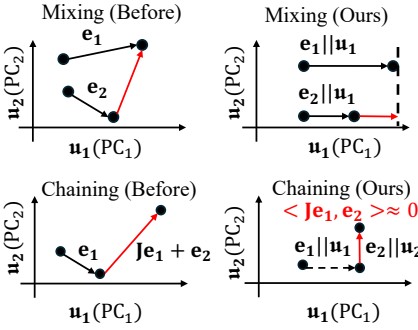

*Figure 6.* **Unified directional residual error control.** *Top (mixing)*: make residuals from different source translators directionally consistent by concentrating them along $\mathbf{u}_1$ (PC$_1$). *Bottom (chaining)*: reduce constructive accumulation in two-hop translation by allocating the downstream residual to an orthogonal direction $\mathbf{u}_2$ (PC$_2$).

**Parameter-efficient optimization**. Training $2K - 1$ independent neural translators would scale parameters linearly with the number of experts. To avoid this, we adopt Low-Rank Adaptation (LoRA) (Hu et al., 2022). Each expert introduces cluster-specific low-rank updates on top of a shared base translator, enabling localized specialization with minimal additional parameters. The rank of the low-rank adaptation is set to $r$, which is shared across all experts. Training and inference details are provided in Appendix E.

## 4. Multi-Model Mixing and Chaining

We consider two common multi-model usage patterns. *Mixing* stores embeddings translated from multiple source models in a single target space. *Chaining* composes translators across multiple hops (*e.g.*, $A \rightarrow$ Hub $\rightarrow B$) when a direct translator is unavailable. In both cases, even if each translator has low average error in isolation, their residuals can interact and distort geometry in the shared target space.

Our key idea, as illustrated in Fig. 6, is to make residuals *directionally predictable*: (i) residuals from different sources should be compatible under mixing, and (ii) residuals from different hops should not add constructively under chaining.

**Principal directions**. We extract two stable directions from the target embedding space. Specifically, we run PCA on target embeddings of training objects $\{\mathrm{emb}_t(o) \mid o \in \mathcal{O}_{\mathrm{train}}\}$ and denote by $\mathbf{u}_1$ and $\mathbf{u}_2$ the first two principal directions (PC$_1$ and PC$_2$). By construction, $\mathbf{u}_1 \perp \mathbf{u}_2$. We use them as low-dimensional "error channels" to shape residuals. We emphasize that this is not an assumption that translation residuals must lie along principal components: $\mathbf{u}_1$ and $\mathbf{u}_2$ are simply the most stable, data-driven, geometry-aware basis shared across translators of the same target space. $\mathbf{u}_1$ provides a *common channel* so that residuals from different source translators become more directionally compatible under mixing, while $\mathbf{u}_2 \perp \mathbf{u}_1$ provides the simplest *orthog-*

*onal channel* for the downstream hop in chaining, so the propagated upstream and the downstream residuals occupy decoupled directions to first order.

### 4.1. Consistent Mixing: Aligning along $\mathbf{u}_1$

Let $\mathcal{S}$ be a set of source models and let $t$ be a common target model. For each $s \in \mathcal{S}$, we train a translator $f_{s \rightarrow t}$ into the target space. For each $o \in \mathcal{O}_{\mathrm{train}}$, its *residual error* in the target space is $\mathbf{e}_{s \rightarrow t}(o) := f_{s \rightarrow t}(\mathrm{emb}_s(o)) - \mathrm{emb}_t(o)$.

Mixing is challenging because cross-source distances depend on differences between residuals. For two translated embeddings $\hat{\mathbf{y}}_1 = \mathrm{emb}_t(o_1) + \mathbf{e}_1$ and $\hat{\mathbf{y}}_2 = \mathrm{emb}_t(o_2) + \mathbf{e}_2$,

$$\|\hat{\mathbf{y}}_1 - \hat{\mathbf{y}}_2\|_2^2 = \|\mathbf{y}_1 - \mathbf{y}_2\|_2^2 + \|\mathbf{e}_1 - \mathbf{e}_2\|_2^2 + 2\langle \mathbf{y}_1 - \mathbf{y}_2, \ \mathbf{e}_1 - \mathbf{e}_2 \rangle,$$

where $\mathbf{y}_i := \mathrm{emb}_t(o_i)$. The latter two terms are induced by residuals; if residual directions vary arbitrarily across source translators, they introduce large distortions and accumulate quadratically: $\mathbb{E}\|\mathbf{e}_1 - \mathbf{e}_2\|_2^2 \approx \mathbb{E}\|\mathbf{e}_1\|_2^2 + \mathbb{E}\|\mathbf{e}_2\|_2^2$.

**Consistent mixing**. To improve cross-source compatibility, we direct residuals from all sources to concentrate along the same direction $\mathbf{u}_1$ by penalizing their component orthogonal to $\mathbf{u}_1$. For each translator $f_{s \rightarrow t}$, we add the regularizer

$$\mathcal{L}_{\mathrm{dir}}^{mix} = \mathbb{E}_{o \sim \mathcal{O}_{\mathrm{train}}} \left[ \left\| \mathbf{e}_{s \rightarrow t}(o) - (\mathbf{e}_{s \rightarrow t}(o)^\top \mathbf{u}_1) \mathbf{u}_1 \right\|_2^2 \right].$$

This directs the residual so that different source translators introduce more consistent distortions when mixed.

### 4.2. Transitive Chaining: Orthogonal Decoupling via $\mathbf{u}_2$

We next consider chaining $s \xrightarrow{f_{s \rightarrow h}}$ Hub $\xrightarrow{f_{h \rightarrow t}} t$, *i.e.,* translation from source model $s$ to a target model $t$ via an intermediate hub model $h$ using two translators, $f_{s \rightarrow h}$ and $f_{h \rightarrow t}$. For a test object $o$, define $\mathbf{z} = \mathrm{emb}_h(o)$ and $\mathbf{y} = \mathrm{emb}_t(o)$. The chained output is $\hat{\mathbf{y}} = f_{h \rightarrow t}(f_{s \rightarrow h}(\mathrm{emb}_s(o)))$.

Let the first-hop residual in hub space be $\mathbf{e}_1 := f_{s \rightarrow h}(\mathrm{emb}_s(o)) - \mathbf{z}$, and the second-hop residual (evaluated at the clean hub embedding) be $\mathbf{e}_2 := f_{h \rightarrow t}(\mathbf{z}) - \mathbf{y}$. A first-order Taylor approximation around $\mathbf{z}$ gives

$$f_{h \rightarrow t}(\mathbf{z} + \mathbf{e}_1) \approx f_{h \rightarrow t}(\mathbf{z}) + \mathbf{J}_{f_{h \rightarrow t}}(\mathbf{z}) \, \mathbf{e}_1,$$

where $\mathbf{J}_{f_{h \rightarrow t}}(\mathbf{z})$ is the Jacobian of $f_{h \rightarrow t}$ *w.r.t.* its input evaluated at $\mathbf{z} = \mathrm{emb}_h(o)$. Thus the chained error in the target space satisfies $\hat{\mathbf{y}} - \mathbf{y} \approx \mathbf{J}_{f_{h \rightarrow t}}(\mathbf{z}) \mathbf{e}_1 + \mathbf{e}_2$, and

$$\|\hat{\mathbf{y}} - \mathbf{y}\|_2^2 \approx \|\mathbf{J}_{f_{h \rightarrow t}}(\mathbf{z}) \mathbf{e}_1\|_2^2 + \|\mathbf{e}_2\|_2^2 + 2\langle \mathbf{J}_{f_{h \rightarrow t}}(\mathbf{z}) \mathbf{e}_1, \ \mathbf{e}_2 \rangle.$$

The inner-product term captures *constructive accumulation*: when the propagated upstream residual and the downstream residual are aligned, the cross term becomes large and chaining deviates substantially from a direct translation.

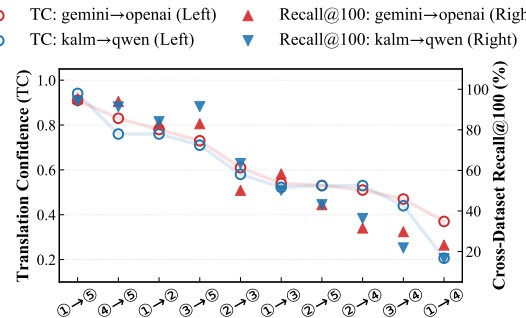

*Figure 7.* **Translation Confidence (TC($\mathbf{x}$)) scores** vs. **Recall@100 across $\mathcal{O}_{\text{train}} \rightarrow \mathcal{O}_{\text{test}}$ pairs.** (Datasets are indexed by ① SciDocs, ② ArguAna, ③ FiQA-2018, ④ NFCorpus, and ⑤ SciFact.)

**Orthogonal decoupling.** To promote translation transitivity, we encourage these two contributions to occupy different target-space directions. Specifically, we (i) apply the same projection penalty as $\mathcal{L}_{\text{mix}}$ to the *propagated upstream perturbation* in the target space, $f_{h \rightarrow t}(\mathbf{z}+\mathbf{e}_1) - f_{h \rightarrow t}(\mathbf{z})$ (which equals $\mathbf{J}_{f_{h \rightarrow t}}(\mathbf{z})\mathbf{e}_1$ to first order), encouraging it to concentrate along $\mathbf{u}_1$; and (ii) regularize the second-hop residual to concentrate along the orthogonal direction $\mathbf{u}_2$ via

$$\mathcal{L}_{\text{dir}}^{chain} = \mathbb{E}_{o \sim \mathcal{O}_{\text{train}}} \left[ \left\| \mathbf{e}_2 - (\mathbf{e}_2^\top \mathbf{u}_2)\mathbf{u}_2 \right\|_2^2 \right],$$

where $\mathbf{e}_2 = f_{h \rightarrow t}(\text{emb}_h(o)) - \text{emb}_t(o)$. By directing residual contributions to orthogonal channels, we reduce the expected cross term $\langle \mathbf{J}_{f_{h \rightarrow t}}(\mathbf{z})\mathbf{e}_1, \ \mathbf{e}_2 \rangle$ in chained error, which stabilizes multi-hop composition and improves transitivity. Combining the $\mathcal{L}_{\text{reg}}$ and $\mathcal{L}_{\text{local}}$, the final loss become $\mathcal{L} = \mathcal{L}_{\text{reg}} + \alpha \mathcal{L}_{\text{local}} + \beta \mathcal{L}_{\text{dir}}$, we ablate each term and test the sensitivity in the Sec. 5 and Appendix K.

**Putting it all together**. H-MoE combines all components into a risk-aware translation pipeline. (i) TC flags a test vector as risky *before* translation when it is far from anything seen during training, so a downstream system can abstain or fall back to re-embedding. (ii) Based on Theorem 1, local experts of H-MoE allow references to cover more OOD inputs that otherwise would be risky to translate. Further, hierarchical routing makes the per-query expert choice stable: when a query falls cleanly inside one child cluster, it is served by the corresponding fine expert; when it is near a boundary, the router backs off to a coarser expert whose training support actually covers it. (iii) For mixing and chaining, directional regularization shapes residual errors so that errors from different translators occupy compatible directions rather than interfere constructively. All these methods make H-MoE robust for both OOD and mixing/chaining scenarios.

## 5. Experimental Study

**Main findings**. Across 10 embedding models, 6 BEIR datasets, and 90 directed translation directions, we find:

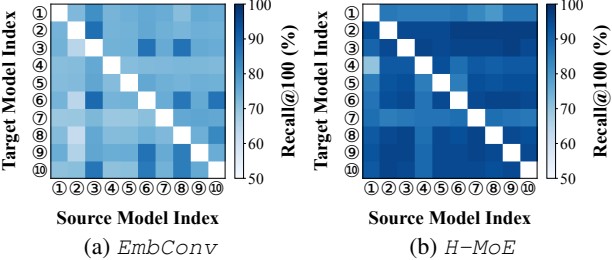

(a) *EmbConv*    (b) *H-MoE*

*Figure 8.* **Heatmap of cross-model translation performance.** (All results trained on the FiQA-2018 and test on the SciFact. The indices follow: ① e5, ② gemini, ③ gritlm, ④ kalm, ⑤ linq, ⑥ mistral, ⑦ nemotron, ⑧ openai, ⑨ qwen, and ⑩ sfr.)

- *Translation risk estimate*: Translation confidence (TC) strongly correlates with Recall@100 over translated embeddings under cross-dataset evaluation (Fig. 7; § 5.2).

- *Best OOD pairwise translation*: H-MoE has the top Recall@100 across all cases, improving by 12.0% on average over the strongest baseline (Table 1, Fig. 8; § 5.3).

- *Robust multi-model interoperability*: mixing and chaining degrade H-MoE by only 0.5%–2.6% Recall@100, while baselines can drop by up to 36.6% (mixing) and 36.8% (chaining) (Fig. 9; § 5.4).

- *Efficiency:* H-MoE translates 1M vectors in < 1 s with negligible overhead with more experts (Table 3; § 5.5).

### 5.1. Experimental setup

**Datasets**. Following BEIR (Thakur et al., 2021), we evaluated on six retrieval datasets: SciFact (Wadden et al., 2020) and Fever (Thorne et al., 2018) (fact checking), SciDocs (Cohan et al., 2020) (citation prediction), ArguAna (Wachsmuth et al., 2018) (argument retrieval), FiQA-2018 (Maia et al., 2018) (financial QA), and NFCorpus (Boteva et al., 2016) (biomedical IR). Dataset statistics are in Appendix F. We report Recall@100, which is stable for methods whose performance can be volatile under cross-dataset shift.

**Embedding models**. Following MTEB leaderboard (Muennighoff et al., 2023), we used 10 major embedding models: linq (Kim et al., 2024), e5 (Wang et al., 2024), sfr (Meng et al., 2024), gritlm (Muennighoff et al., 2024), kalm (Zhao et al., 2025), nemotron (Babakhin et al., 2025), qwen (Zhang et al., 2025), openai (OpenAI, 2022), mistral (Jiang et al., 2023), and gemini (Google, 2025). This yields 90 translation directions (details in Appendix G).

**Configuration**. We set up translation tasks as follows.

*Pairwise OOD translation*. A pairwise translation aims to translate two different representations (by models $s$ and $t$) of the same test corpora following a direction $s \rightarrow t$. We trained translators on a training dataset that is different from the test corpora, generating domain shifts. We use $\mathcal{O}_{\text{train}} \rightarrow \mathcal{O}_{\text{test}}$ to represent that we train translator over corpora

*Table 1.* **Cross-data Translation performance (Recall@100).** We adopt the setting from Fever to multiple target datasets (five runs). **Bold** and underline denote the best and runner-up average results in each column, respectively. Values highlighted in red indicate high-variance outcomes ($std > 10$). We only report **7** pairs involving all 10 models due to space limit; the remaining **83** pairs are reported in Appendix J.

| Method | Dataset | gemini→openai | linq→gritlm | qwen→gemini | linq→e5 | mistral→openai | e5→sfr | nemotron→linq |
|---|---|---|---|---|---|---|---|---|
| GW | SciDocs | $0.1_{\pm 0.0}$ | $0.2_{\pm 0.0}$ | $0.9_{\pm 0.2}$ | $0.1_{\pm 0.0}$ | $0.1_{\pm 0.0}$ | $0.1_{\pm 0.0}$ | $0.4_{\pm 0.1}$ |
| | FiQA-2018 | $0.0_{\pm 0.0}$ | $0.0_{\pm 0.0}$ | $0.5_{\pm 0.1}$ | $0.2_{\pm 0.0}$ | $0.0_{\pm 0.0}$ | $0.0_{\pm 0.0}$ | $0.0_{\pm 0.0}$ |
| | ArguAna | $1.1_{\pm 0.1}$ | $1.6_{\pm 0.2}$ | $0.7_{\pm 0.2}$ | $1.0_{\pm 0.1}$ | $1.0_{\pm 0.1}$ | $0.8_{\pm 0.1}$ | $2.9_{\pm 0.4}$ |
| | **Avg.** | 0.4 | 0.6 | 0.7 | 0.4 | 0.4 | 0.3 | 1.1 |
| PA | SciDocs | $0.1_{\pm 0.0}$ | $10.0_{\pm 7.8}$ | $0.2_{\pm 0.0}$ | $0.1_{\pm 0.0}$ | $\color{red}13.4_{\pm 5.5}$ | $0.2_{\pm 0.0}$ | $\color{red}18.4_{\pm 6.5}$ |
| | FiQA-2018 | $0.0_{\pm 0.0}$ | $\color{red}13.3_{\pm 12.0}$ | $0.5_{\pm 0.1}$ | $0.4_{\pm 0.1}$ | $\color{red}16.9_{\pm 9.6}$ | $0.0_{\pm 0.0}$ | $\color{red}16.0_{\pm 10.1}$ |
| | ArguAna | $1.1_{\pm 0.1}$ | $\color{red}18.4_{\pm 14.9}$ | $1.1_{\pm 0.1}$ | $1.1_{\pm 0.1}$ | $\color{red}35.1_{\pm 16.2}$ | $1.0_{\pm 0.2}$ | $\color{red}20.1_{\pm 15.0}$ |
| | **Avg.** | 0.4 | 13.9 | 0.6 | 0.5 | 21.8 | 0.4 | 18.2 |
| CCA | SciDocs | $24.1_{\pm 3.0}$ | $40.5_{\pm 5.6}$ | $29.5_{\pm 4.5}$ | $24.1_{\pm 3.3}$ | $24.8_{\pm 3.9}$ | $44.1_{\pm 7.3}$ | $38.8_{\pm 5.5}$ |
| | FiQA-2018 | $47.8_{\pm 7.4}$ | $\color{red}75.9_{\pm 11.0}$ | $30.2_{\pm 5.0}$ | $\color{red}72.8_{\pm 10.5}$ | $49.7_{\pm 6.7}$ | $\color{red}79.0_{\pm 12.2}$ | $\color{red}69.3_{\pm 10.3}$ |
| | ArguAna | $\color{red}73.8_{\pm 12.3}$ | $73.8_{\pm 9.8}$ | $\color{red}75.0_{\pm 10.5}$ | $\color{red}73.8_{\pm 11.9}$ | $71.8_{\pm 9.4}$ | $\color{red}79.0_{\pm 13.8}$ | $\color{red}76.9_{\pm 10.1}$ |
| | **Avg.** | 48.6 | 63.4 | 44.9 | 56.9 | 48.8 | 67.4 | 61.7 |
| Vec2Vec | SciDocs | $20.3_{\pm 5.8}$ | $20.3_{\pm 6.2}$ | $24.7_{\pm 6.6}$ | $38.4_{\pm 8.6}$ | $\color{red}25.0_{\pm 13.9}$ | $\color{red}45.0_{\pm 10.3}$ | $27.8_{\pm 6.8}$ |
| | FiQA-2018 | $\color{red}31.1_{\pm 10.5}$ | $31.1_{\pm 8.7}$ | $\color{red}20.3_{\pm 12.4}$ | $\color{red}23.4_{\pm 11.1}$ | $11.1_{\pm 9.5}$ | $31.1_{\pm 9.9}$ | $31.1_{\pm 6.5}$ |
| | ArguAna | $\color{red}37.2_{\pm 12.6}$ | $\color{red}14.8_{\pm 13.0}$ | $\color{red}36.6_{\pm 14.2}$ | $\color{red}33.4_{\pm 13.0}$ | $\color{red}37.2_{\pm 11.2}$ | $\color{red}36.6_{\pm 10.7}$ | $\color{red}42.9_{\pm 9.9}$ |
| | **Avg.** | 29.5 | 22.1 | 27.2 | 31.7 | 24.4 | 37.6 | 33.9 |
| EmbConv | SciDocs | $57.0_{\pm 1.3}$ | $57.0_{\pm 1.6}$ | $52.6_{\pm 1.9}$ | $48.6_{\pm 1.6}$ | $50.6_{\pm 1.3}$ | $38.2_{\pm 0.9}$ | $57.0_{\pm 2.2}$ |
| | FiQA-2018 | $78.3_{\pm 2.4}$ | $73.6_{\pm 2.0}$ | $72.3_{\pm 2.3}$ | $71.9_{\pm 1.6}$ | $71.9_{\pm 2.2}$ | $59.7_{\pm 1.6}$ | $52.5_{\pm 1.1}$ |
| | ArguAna | $79.6_{\pm 1.8}$ | $79.6_{\pm 2.7}$ | $78.6_{\pm 3.0}$ | $78.4_{\pm 2.1}$ | $87.6_{\pm 1.8}$ | $77.4_{\pm 1.9}$ | $76.3_{\pm 2.9}$ |
| | **Avg.** | 71.6 | 70.1 | 67.8 | 66.3 | 70.0 | 58.4 | 61.9 |
| **H-MoE** (Ours) | SciDocs | $56.7_{\pm 2.5}$ | $58.2_{\pm 3.6}$ | $61.1_{\pm 4.4}$ | $67.6_{\pm 3.4}$ | $73.2_{\pm 1.1}$ | $72.1_{\pm 4.5}$ | $62.1_{\pm 1.0}$ |
| | FiQA-2018 | $88.3_{\pm 0.8}$ | $86.7_{\pm 6.1}$ | $79.8_{\pm 0.1}$ | $77.8_{\pm 2.3}$ | $85.0_{\pm 6.7}$ | $80.3_{\pm 0.6}$ | $80.3_{\pm 3.5}$ |
| | ArguAna | $92.4_{\pm 5.9}$ | $91.6_{\pm 1.3}$ | $92.4_{\pm 2.7}$ | $93.6_{\pm 1.7}$ | $92.9_{\pm 4.1}$ | $93.1_{\pm 2.4}$ | $93.1_{\pm 1.6}$ |
| | **Avg.** | **79.1** | **78.8** | **77.8** | **79.7** | **83.7** | **81.8** | **78.5** |

$\mathcal{O}_{train}$ and use it to translate different embeddings of $\mathcal{O}_{test}$. In all cases, we use full dataset for both $\mathcal{O}_{train}$ and $\mathcal{O}_{test}$.

*Mixing and chaining*. We evaluated multi-model mixing and two-hop chaining using setup in Fig. 9 (see Appendix H).

**Baselines**. We compared to linear translators PA, CCA, and GW, and neural translators Vec2Vec and EmbConv. For GW, PA, CCA, and EmbConv we followed the original papers; for Vec2Vec we used the authors' implementation.

**Implementation**. We implemented our method as a unified library with precomputed embeddings for all documents and queries across 10 models and 6 datasets (60 embedding sets, 900 GB). Open-weight models were executed locally under default settings, with a maximum input length of 4096 tokens to fit GPU memory; For H-MoE, we used $k=8$ clusters on Fever and $k=4$ otherwise, and fixed the LoRA rank to $r=8$. The hyper-parameters are set to $\tau = 0.8, \alpha = 0.5, \beta = 0.7$. All experts were trained with a learning rate of $10^{-4}$ and early stopping. Experiments were conducted on an AMD EPYC 7713P CPU with 120 GB RAM and an NVIDIA A100 80 GB GPU.

### 5.2. Effectiveness of Translation Confidence (TC(x))

In section 2, we have already reported per-vector TC. Here we further evaluated whether TC tracks translation quality at the dataset level. For two representative translation directions openai → gemini and kalm → qwen, we com-

puted dataset-level confidence by averaging TC over the corpus embeddings to be translated in each $\mathcal{O}_{train} \rightarrow \mathcal{O}_{test}$ dataset transfer pair, and compare it with the Recall@100.

Figure 7 shows a strong positive relationship: recall decreases consistently as TC drops across both model pairs. Notably, transfers involving NFCorpus (biomedical) yield low TC and substantial performance degradation, reflecting its domain mismatch with the other datasets. This validates TC as a practical, model-agnostic indicator for deciding when to translate (or abstain) under distribution shift.

### 5.3. Cross-Dataset (OOD) Pairwise Translation

Tables 1 and App. J confirm the robust generalization of H-MoE under substantial domain shifts. Although trained solely on Fever, H-MoE establishes a new state of the art across all 90 evaluated pairs, outperforming the strongest baseline (EmbConv) by an average of $14.9\%$ in Recall@100. Importantly, H-MoE exhibits markedly improved stability by avoiding high-variance results ($std > 10$) common to baselines such as Vec2Vec and PA.

Figure 8 shows cross-dataset results for the FiQA-2018 → SciFact direction across all translation pairs, comparing H-MoE with the strongest baseline, EmbConv. H-MoE consistently exhibits stronger performance, as reflected by deeper saturation. Although minor variation appears across certain directions (*e.g.,* bridging encoder-only (① e5) and LLM-based embeddings), recall remains above $90\%$ for

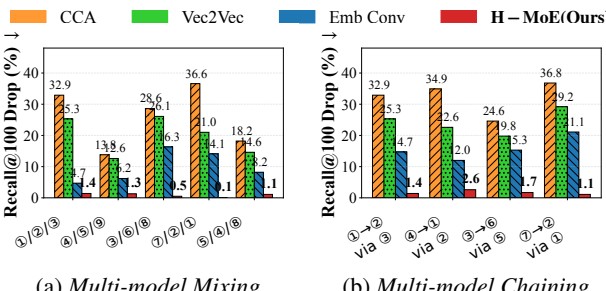

(a) *Multi-model Mixing*  (b) *Multi-model Chaining*

*Figure 9.* **Multi-model performance degradation.** Evaluation under FiQA-2018 → SciFact setting. (a) **Mixing**: Co-existence of original and translated embeddings (*e.g.,* ① along with ② → ① and ③ → ① in a single retrieval database. (b) **Chaining**: Cumulative error in multi-hop translation (*e.g.,* ① → ② via ③). Bars represent drop relative to direct translation. Indices follow the Fig. 8.

*Table 2.* **Ablation study of architectural capacity and geometric constraints.** $P_1$: gemini→openai, $P_2$: kalm→qwen. Values denote Recall@100 (OOD) and performance degradation ($\Delta \downarrow$) for Interoperability. All metrics are reported as $P_1$ / $P_2$.

| Variant | Arch. | | Loss | | OOD ($\uparrow$) | Mixing ($\downarrow$) | Chaining ($\downarrow$) |
|---|---|---|---|---|---|---|---|
| | Mono | MoE | $\mathcal{L}_{loc}$ | $\mathcal{L}_{dir}$ | $P_1$ / $P_2$ | $P_1$ / $P_2$ | $P_1$ / $P_2$ |
| (1) Base | ✓ | – | – | – | 83.8/85.9 | 28.8/17.5 | 24.6/27.3 |
| (2) + Constraint | ✓ | – | ✓ | ✓ | 86.7/87.4 | 3.8/1.1 | 2.5/1.9 |
| (3) + Capacity | – | ✓ | ✓ | – | 89.8/91.1 | 11.1/13.4 | 16.7/18.9 |
| **(4) H-MoE** | – | ✓ | ✓ | ✓ | **93.4 / 94.2** | **2.3/0.8** | **1.8/1.4** |

most pairs. These results demonstrate that H-MoE reliably translates between heterogeneous models with minimal degradation, highlighting its practical utility.

### 5.4. Multi-Model Interoperability Analysis

**Mixing**. Figure 9a shows that aggregating embeddings from heterogeneous source models into a shared target index causes severe performance degradation for baseline methods, with Recall@100 dropping by up to 36.6%. In contrast, H-MoE typically degrades by less than 2%. This observation aligns with our theoretical analysis in Sec. 4.

**Chaining**. We further evaluated 2-hop chaining. As shown in Fig. 9b, baselines suffer substantial error amplification. In particular, EmbConv incurs a 21.1% performance drop, indicating that strong pairwise performance does not imply robust composition. By contrast, H-MoE limits chaining degradation to within about 3% of direct translation, validating our orthogonal decoupling strategy from Section 4.2.

### 5.5. Ablation and Efficiency

We carried out an ablation and efficiency study using FiQA-2018 → SciFact as $\mathcal{O}_{train}$ → $\mathcal{O}_{test}$ dataset transfer pair (See also Appendix K for a sensitivity study).

**Ablation Study.** Table 2 demonstrates that both architectural capacity and geometric controls are critical. Incorporating geometric constraints ($\mathcal{L}_{dir}$) substantially improves

*Table 3.* **Inference Efficiency and Scalability Analysis.** We fix the LoRA rank $r = 8$ and report performance here. Latency denotes amortized per-sample processing time; Total Time represents end-to-end conversion for $10^6$ vectors on a single NVIDIA A100 GPU.

| Method | Params ($\downarrow$) | Latency / $\mu$s $\downarrow$ | | Total Time / s ($\downarrow$) | Throughput ($\uparrow$) |
|---|---|---|---|---|---|
| | (M) | $10^3$ | $10^6$ | $10^6$ | (M s/s) |
| EmbConv | 0.62 | 0.71 | 0.61 | 0.61 s | 1.64 |
| H-MoE ($k = 4$) | 0.79 | 0.81 | 0.72 | 0.72 | 1.39 |
| H-MoE ($k = 6$) | 0.96 | 0.83 | 0.73 | 0.73 | 1.37 |
| H-MoE ($k = 8$) | 1.22 | 0.85 | 0.75 | 0.75 | 1.33 |

stability, reducing mixing and chaining errors (*e.g.,* from 28.8% to 3.8% on $P_1$) relative to the (1), underscoring the importance of directional residual alignment. Increasing architectural capacity with MoE (3) yields notable OOD gains (*e.g.,* +6.0% on $P_1$) over the single varaint, indicating local experts more effective and align with our analysis.

**Efficiency**. Table 3 evaluates the computational efficiency of our method. Although the hierarchical strategy theoretically instantiates $2k - 1$ experts (*e.g.,* 15 experts for $k = 8$), LoRA parameterization restricts the parameter count to about $2.0\times$ that of the monolithic baseline (1.22M vs. 0.62M). Inference efficiency is similarly preserved: despite the added routing logic, latency increases modestly (0.75µs vs. 0.61µs), demonstrating scalability of H-MoE.

## 6. Conclusion

We study embedding translation under out-of-distribution (OOD) inputs, multi-model mixing, and multi-hop chaining, which are critical for practical cross-model interoperability. We derive a geometric pointwise error bound that explains performance degradation under OOD shift and motivates a translator-independent confidence score. Building on this analysis, we propose H-MoE, a hierarchical mixture-of-experts translator that is robust under OOD and improves both consistency and transitivity. Extensive experiments across 10 embedding models, 6 datasets, and 90 translation directions demonstrate strong OOD translation performance and substantially reduced degradation in multi-model scenarios. Future work includes tighter end-to-end retrieval analysis and extensions to multimodal and evolving spaces.

**Limitations**. H-MoE still depends on the coverage of the source-side reference pool. When inputs are far from all training references, translation can be unreliable; while TC should flag those risky cases, it is up to the users to decide whether to translate or provide a fallback, *e.g.,* re-embedding. Further, our experiments focus on text embedding models and retrieval metrics; evaluating downstream end-to-end RAG pipelines may introduce additional generator and reranker effects, which requires further systematic studies. We also mainly study two-hop chaining, while longer translation chains and continuously evolving embedding spaces may offer opportunities for deep future work.

## Impact Statement

This paper presents work whose goal is to advance the field of Machine Learning. There are many potential societal consequences of our work, none of which we feel must be specifically highlighted here.

## Acknowledgements

We thank the anonymous ICML reviewers and area chair for their constructive feedback. This work is supported by RAEng RF\201920\19\319 and the Huawei-Edinburgh Joint Lab.

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

# A. Embedding Translation Scenario Settings

## A.1. Preliminaries

Let $\mathcal{O}$ denote a universe of objects. We consider two embedding models: a *source model* $\mathrm{emb}_A$ and a *target model* $\mathrm{emb}_B$, which map each object $o \in \mathcal{O}$ to a source embedding $\mathbf{x} := \mathrm{emb}_A(o) \in \mathbb{R}^{d_A}$ and a target embedding $\mathbf{y} := \mathrm{emb}_B(o) \in \mathbb{R}^{d_B}$, respectively.

**Embedding Translation.** The goal of embedding translation is to learn a function $f_{A \to B} : \mathbb{R}^{d_A} \to \mathbb{R}^{d_B}$ such that the translated source embedding approximates the target embedding of the same object. The translator $f_{A \to B}$ is trained using a paired dataset defined on a training collection $\mathcal{O}_{\mathrm{train}}$:

$$\mathcal{D}_{\mathrm{train}} := \{(\mathrm{emb}_A(o), \mathrm{emb}_B(o)) \mid o \in \mathcal{O}_{\mathrm{train}}\}. \tag{1}$$

The training objective can be minimize the distance $\mathrm{Dist}(f_{A \to B}(\mathbf{x}), \mathbf{y})$ over all pairs in $\mathcal{D}_{\mathrm{train}}$.

## A.2. Translation Scenarios

All scenarios are evaluated on a held-out test collection $\mathcal{O}_{\mathrm{test}}$, where $\mathcal{O}_{\mathrm{train}} \cap \mathcal{O}_{\mathrm{test}} = \emptyset$. Evaluation is conducted via a retrieval-based protocol, as detailed below.

**Pairwise Translation Scenario.** This scenario evaluates a single translator $f_{A \to B}$ between one source model and one target model. The evaluation proceeds as follows:

1. *Corpus Translation.* The learned translator $f_{A \to B}$ is applied to all objects in $\mathcal{O}_{\mathrm{test}}$. For each object $o \in \mathcal{O}_{\mathrm{test}}$, we compute the translated embedding $\hat{\mathbf{y}}_o := f_{A \to B}(\mathrm{emb}_A(o))$. The resulting translated corpus is $\hat{\mathcal{Y}}_{\mathrm{test}} := \{\hat{\mathbf{y}}_o \mid o \in \mathcal{O}_{\mathrm{test}}\}$.

2. *Retrieval.* For each query object $q \in \mathcal{O}_{\mathrm{test}}$, its ground-truth target embedding $\mathrm{emb}_B(q)$ is used as the query. Nearest neighbor search is performed over $\hat{\mathcal{Y}}_{\mathrm{test}}$.

3. *Metric.* Candidates in $\hat{\mathcal{Y}}_{\mathrm{test}}$ are ranked by their distance to $\mathrm{emb}_B(q)$. Performance is measured by Recall@100, which checks whether the ground-truth answer (by index) for $q$ appears in the top-100 retrieved results.

**Multi-model Mixing Scenario.** This scenario extends pairwise translation to multiple source models that share a common target space. We train multiple translators covering different source–target pairs (e.g., $\mathrm{emb}_A \to \mathrm{emb}_B$, $\mathrm{emb}_C \to \mathrm{emb}_B$, $\mathrm{emb}_D \to \mathrm{emb}_B$). At inference time, the translated corpus is constructed as follows:

- The test collection $\mathcal{O}_{\mathrm{test}}$ is partitioned into disjoint subsets, with each subset assigned to a specific translator.

- Each translator is applied to its assigned subset to produce translated embeddings in the shared target space.

- Translated embeddings from all translators are merged into a single mixed search corpus.

- Retrieval and evaluation follow the same protocol and metric as in the pairwise translation scenario.

**Multi-model Chaining Scenario.** This scenario evaluates sequential composition of translators through an intermediate embedding space. Specifically, we consider a two-hop translation pipeline of the form $\mathrm{emb}_A \to \mathrm{emb}_B \to \mathrm{emb}_C$. The evaluation protocol largely follows the pairwise scenario:

- *Sequential Translation.* For each object $o \in \mathcal{O}_{\mathrm{test}}$, we first apply $f_{A \to B}$ to obtain an intermediate embedding, followed by $f_{B \to C}$ to produce the final translated embedding.

- *Corpus Construction.* Final translated embeddings for all objects form the search corpus in the target space.

- *Retrieval.* For each query object $q \in \mathcal{O}_{\mathrm{test}}$, retrieval is performed using its ground-truth target embedding as the query.

- *Metric.* Performance is evaluated using Recall@100, consistent with the pairwise translation scenario.

# B. Proof of Theorem 1.

We first briefly recall the key notations defined in Section 3. Let $f$ be the learned translator and $f^\star$ be the ground-truth translator. For any test sample $\mathbf{x} \in \mathcal{X}_{test}$, we denote its nearest neighbor in the training reference set $\mathcal{X}_{train}$ as:

$$\mathbf{x}_{nn} = \arg \min_{\mathbf{x}' \in \mathcal{X}_{train}} \|\mathbf{x} - \mathbf{x}'\|.$$

The distance to the training manifold is $\delta(\mathbf{x}) = \|\mathbf{x} - \mathbf{x}_{nn}\|$. Furthermore, $f$ and $f^\star$ are assumed to be $L$-Lipschitz and $L^\star$-Lipschitz continuous, respectively.

**Proof:** By adding and subtracting $f(\mathbf{x}_{nn})$ and $f^\star(\mathbf{x}_{nn})$ inside the norm and applying the triangle inequality, we can decompose the pointwise translation error into three distinct components:

$$
\begin{aligned}
e(\mathbf{x}) &= \|f(\mathbf{x}) - f^\star(\mathbf{x})\| \\
&= \|f(\mathbf{x}) - f(\mathbf{x}_{nn}) + f(\mathbf{x}_{nn}) - f^\star(\mathbf{x}_{nn}) + f^\star(\mathbf{x}_{nn}) - f^\star(\mathbf{x})\| \\
&\leq \underbrace{\|f(\mathbf{x}) - f(\mathbf{x}_{nn})\|}_{\text{Term (I)}} + \underbrace{\|f(\mathbf{x}_{nn}) - f^\star(\mathbf{x}_{nn})\|}_{\text{Term (II)}} + \underbrace{\|f^\star(\mathbf{x}_{nn}) - f^\star(\mathbf{x})\|}_{\text{Term (III)}}.
\end{aligned}
\tag{2}
$$

We now bound each term individually based on our assumptions:

- **Term (I):** By the $L$-Lipschitz continuity of the learned translator $f$, we have:

$$\|f(\mathbf{x}) - f(\mathbf{x}_{nn})\| \leq L\|\mathbf{x} - \mathbf{x}_{nn}\| = L\delta(\mathbf{x}).$$

- **Term (II):** By definition, this term represents the translation error at the training point $\mathbf{x}_{nn}$, which we denote as $e(\mathbf{x}_{nn})$.

- **Term (III):** Similarly, by the $L^\star$-Lipschitz continuity of the ground-truth translator $f^\star$, we have:

$$\|f^\star(\mathbf{x}_{nn}) - f^\star(\mathbf{x})\| \leq L^\star\|\mathbf{x}_{nn} - \mathbf{x}\| = L^\star\delta(\mathbf{x}).$$

Substituting these bounds back into Equation (2), we obtain:

$$
\begin{aligned}
e(\mathbf{x}) &\leq L\delta(\mathbf{x}) + e(\mathbf{x}_{nn}) + L^\star\delta(\mathbf{x}) \\
&= (L + L^\star)\delta(\mathbf{x}) + e(\mathbf{x}_{nn}).
\end{aligned}
$$

This completes the proof. □

## B.1. Extension to Cosine Distance for Normalized Embeddings

In practice, embeddings are often $L_2$-normalized such that $\|\mathbf{x}\| = 1$ for all $\mathbf{x} \in \mathcal{X}$. In this context, we define the cosine distance between two vectors $\mathbf{u}$ and $\mathbf{v}$ as $d_{cos}(\mathbf{u}, \mathbf{v}) = 1 - \mathbf{u}^\top \mathbf{v}$. We now show that the Euclidean-based bound in Theorem 1 directly constrains the translation error in terms of cosine distance.

**Relationship between Euclidean and Cosine Distance** For any two unit vectors $\mathbf{u}, \mathbf{v}$ where $\|\mathbf{u}\| = \|\mathbf{v}\| = 1$, the following identity holds:

$$\|\mathbf{u} - \mathbf{v}\|^2 = \|\mathbf{u}\|^2 + \|\mathbf{v}\|^2 - 2\mathbf{u}^\top \mathbf{v} = 2 - 2\mathbf{u}^\top \mathbf{v} = 2d_{cos}(\mathbf{u}, \mathbf{v}). \tag{3}$$

This implies that $d_{cos}(\mathbf{u}, \mathbf{v}) = \frac{1}{2}\|\mathbf{u} - \mathbf{v}\|^2$.

**Translation Error in Cosine Space** Let $e_{cos}(\mathbf{x}) = d_{cos}(f(\mathbf{x}), f^\star(\mathbf{x}))$ be the translation error measured by cosine distance. Assuming the outputs of the translators $f$ and $f^\star$ are also normalized, we can apply the result from Theorem 1:

$$
\begin{aligned}
e_{cos}(\mathbf{x}) &= \frac{1}{2}\|f(\mathbf{x}) - f^\star(\mathbf{x})\|^2 \\
&= \frac{1}{2}e(\mathbf{x})^2 \\
&\leq \frac{1}{2}\left[(L + L^\star)\delta(\mathbf{x}) + e(\mathbf{x}_{nn})\right]^2.
\end{aligned}
\tag{4}
$$

where $\delta(\mathbf{x}) = \sqrt{2d_{cos}(\mathbf{x}, \mathbf{x}_{nn})}$ represents the distance to the training set in the source space.

**Conclusion**    Equation (4) demonstrates that although the cosine distance itself does not satisfy the triangle inequality, the translation error in cosine space is strictly bounded by the proximity of the test sample to the training manifold $\mathcal{X}_{train}$ in the source space. As $\delta(\mathbf{x}) \to 0$ (i.e., the test sample approaches a training sample) and the empirical error $e(\mathbf{x}_{nn}) \to 0$, the cosine-based translation error $e_{cos}(\mathbf{x})$ is guaranteed to vanish.

## C. Translation Confidence Across Reference Pools

This appendix complements Section 2.3 by stress-testing $\text{TC}(\mathbf{x})$ on heterogeneous, mixed-domain reference pools, addressing the concern that pooling distinct-domain datasets could expand $\sigma_{\text{data}}$ and push TC values uniformly toward 1.0.

**Setup.**    We progressively enlarge the training/reference pool of the translator $f_{\text{GEMINI}\to\text{OPENAI}}$ in three stages: FiQA-2018 only, FiQA-2018+SciDocs, and FiQA-2018+SciDocs+ARGUANA, by retraining the translator each time and keeping the evaluation set fixed. For each pool we report the mean TC over evaluation embeddings and the Pearson correlation between $\text{TC}(\mathbf{x})$ and the per-vector empirical translation error $e(\mathbf{x})$. We also evaluate a local-$k$NN variant of the normalization in which $\sigma_{\text{data}}$ is replaced by the median nearest-neighbor distance inside each query's local region.

**Results.**    Table 4 summarizes the results.

*Table 4.* **Translation confidence TC as the reference pool becomes more heterogeneous. All Pearson correlations are significant at** $p < 0.001$**.**

| Reference Pool | Global $\sigma_{\text{data}}$ | | Local-$k$NN normalization | |
|---|---|---|---|---|
| | Mean TC | Pearson | Mean TC | Pearson |
| FiQA-2018 | 0.51 | 0.81 | 0.62 | 0.81 |
| FiQA-2018 + SciDocs | 0.48 | 0.78 | 0.55 | 0.77 |
| FiQA-2018 + SciDocs + ARGUANA | 0.31 | 0.78 | 0.51 | 0.76 |

**Discussion.**    We do not observe the hypothesized inflation of TC toward 1.0: the mean TC actually *decreases* ($0.51 \to 0.48 \to 0.31$) as the pool becomes more heterogeneous, because the larger pool also enlarges $\sigma_{\text{data}}$ but $\delta(\mathbf{x})$ does not shrink proportionally. Crucially, the rank correlation between $\text{TC}(\mathbf{x})$ and the empirical error $e(\mathbf{x})$ remains strong across all pools (Pearson $0.81/0.78/0.78$), confirming the intended use of TC in Section 2.3: for a fixed deployed translator, TC preserves the ranking of risky inputs and only the operating threshold needs per-pool calibration. The local-$k$NN variant yields comparable correlations ($0.81/0.77/0.76$) while keeping the absolute TC scale more stable across pools, providing a practical extension for mixed-domain deployments.

## D. Additional Empirical Results

This appendix collects empirical results that complement the main experiments and address questions raised during the review period. All results follow the `gemini → openai` setting of Table **??**, using the same translator backbone and training hyperparameters.

### D.1. In-Domain and Multi-Metric Comparison

The main paper reports R@100 because it is the most stable indicator under strong cross-dataset shift. Table 5 reports the same comparison under R@10, NDCG@10, and MRR@10 on FiQA-2018, SciDocs, and ARGUANA, together with the in-domain (ID) numbers in which $1/3$ of the corpus is used to train the translator and the remainder is used for evaluation. `H-MoE` dominates all baselines across all four metrics on both ID and OOD inputs, so the OOD gains in Table **??** are not bought at the cost of ID quality.

### D.2. Statistical Significance

We assess the significance of the `H-MoE` vs. `EmbConv` gap with a paired bootstrap over the full query set of each of FiQA-2018, SciDocs, and ARGUANA; Table 6 reports the estimated $95\%$ confidence intervals on the per-query gain $\Delta =$

*Table 5.* **Multi-metric retrieval on FiQA-2018, SciDocs, and ARGUANA (gemini → openai). ID uses** $1/3$ **of the corpus to train the translator and the remainder for evaluation; OOD trains on a different corpus and evaluates without further adaptation. N@10 abbreviates NDCG@10 and M@10 abbreviates MRR@10.**

| Type | Method | FiQA-2018 | | | | SciDocs | | | | ARGUANA | | | |
|---|---|---|---|---|---|---|---|---|---|---|---|---|---|
| | | R@100 | R@10 | N@10 | M@10 | R@100 | R@10 | N@10 | M@10 | R@100 | R@10 | N@10 | M@10 |
| ID | CCA | 83.4 | 82.3 | 73.5 | 70.8 | 55.2 | 49.8 | 35.9 | 41.9 | 81.6 | 52.2 | 48.1 | 38.8 |
| | Vec2Vec | 86.5 | 81.3 | 71.3 | 66.4 | 55.8 | 47.2 | 34.2 | 38.4 | 44.8 | 39.4 | 30.4 | 27.2 |
| | EmbConv | 89.3 | 82.6 | 75.4 | 71.7 | 58.6 | 52.5 | 37.5 | 45.8 | 83.8 | 75.8 | 71.4 | 54.3 |
| | H-MoE | **93.4** | **87.5** | **81.8** | **78.3** | **62.1** | **60.2** | **46.7** | **51.3** | **92.8** | **89.4** | **83.1** | **70.5** |
| OOD | CCA | 47.8 | 32.8 | 21.7 | 18.3 | 24.1 | 17.4 | 13.5 | 10.1 | 73.8 | 38.4 | 31.4 | 24.7 |
| | Vec2Vec | 31.1 | 11.4 | 12.3 | 7.2 | 20.3 | 5.8 | 4.3 | 3.4 | 37.2 | 14.8 | 11.9 | 8.6 |
| | EmbConv | 78.3 | 54.8 | 28.4 | 21.6 | 57.0 | 26.3 | 18.3 | 15.3 | 79.6 | 58.7 | 46.8 | 33.1 |
| | H-MoE | **88.3** | **61.7** | **44.2** | **39.7** | **56.7** | **35.1** | **24.2** | **21.8** | **92.4** | **72.6** | **61.3** | **47.8** |

`H-MoE − EmbConv`. Across all four metrics and three datasets, the lower CI endpoint stays above zero, and gains range from +6.1 to +19.7 points.

*Table 6.* **Paired bootstrap** $95\%$ **CI on the per-query gain** $\Delta = $ **H-MoE − EmbConv.**

| Metric | FiQA-2018 | SciDocs | ARGUANA |
|---|---|---|---|
| R@100 | +14.1 [10.2, 17.8] | +7.1 [3.9, 10.0] | +14.5 [10.7, 18.0] |
| NDCG@10 | +16.5 [11.8, 20.9] | +7.1 [3.5, 10.2] | +19.7 [14.1, 24.8] |
| MRR@10 | +18.8 [13.6, 23.4] | +6.1 [2.7, 9.1] | +12.8 [8.1, 17.0] |

### D.3. Translation versus Re-Embedding Compute

Translation operates on existing source-space vectors, whereas re-embedding additionally requires tokenization and a full encoder forward pass over raw text. On a single A100 GPU, re-embedding the SciDocs corpus (25,657 documents) with QWEN3-EMBEDDING-8B takes 171 min, while translating the same number of already-available source embeddings with H-MoE takes 38.43 ms on the same machine — a ratio of roughly $2.7 \times 10^5$. The exact ratio depends on encoder size, document length, batching, and I/O, but the qualitative gap is inherent: translation cost is independent of input text length once source embeddings exist.

### D.4. Underfitting Does Not Explain the OOD Gap

To test the hypothesis that the EmbConv OOD gap is an artifact of overfitting (so an early-stopped EmbConv would close it), we evaluate EmbConv checkpoints at $10\%, 20\%, \dots, 100\%$ of training under the same OOD setting as Table **??**; results in Table 7 show that no early-stopped checkpoint comes close to H-MoE (*e.g.,* 79.6 vs. 92.4 on ARGUANA at the strongest checkpoint). This is consistent with the bound $e(\mathbf{x}) \leq (L + L^*)\delta(\mathbf{x}) + e(\mathbf{x}_{nn})$ in Theorem 1: global underfitting may reduce the expansion ($L$) but leaves the fit term ($e(\mathbf{x}_{nn})$) large, whereas H-MoE reduces both terms by splitting the problem into local regions.

*Table 7.* **OOD R@100 of EmbConv checkpoints at every** $10\%$ **of training, and the converged H-MoE for reference (gemini → openai).**

| EmbConv fraction | 10% | 20% | 30% | 40% | 50% | 60% | 70% | 80% | 90% | 100% | H-MoE |
|---|---|---|---|---|---|---|---|---|---|---|---|
| FiQA-2018 | 34.8 | 51.7 | 73.8 | 74.5 | 71.9 | 68.4 | 72.9 | 74.9 | 77.8 | 78.3 | **88.3** |
| SciDocs | 21.5 | 33.8 | 46.1 | 49.8 | 47.3 | 44.5 | 50.6 | 53.9 | 55.8 | 57.0 | **56.7** |
| ARGUANA | 36.2 | 61.2 | 68.9 | 72.3 | 68.8 | 66.5 | 71.8 | 74.6 | 77.3 | 79.6 | **92.4** |

### D.5. Component Contributions of the Directional Loss

Table 8 isolates the contribution of the directional regularizer $\mathcal{L}_{dir}$ from the local-specialization losses ($\mathcal{L}_{reg} + \alpha\mathcal{L}_{local}$). With only $\mathcal{L}_{dir}$, the model retains the directional control but loses local expansion control: composition becomes much more stable than the global baseline, but pairwise OOD recall drops to 78.4. Conversely, removing $\mathcal{L}_{dir}$ keeps pairwise OOD

recall at 89.8 but increases the mixing drop to 11.1% and the chaining drop to 16.7%. Combining both losses gives the best overall result.

*Table 8.* **Effect of removing each loss family on** `gemini` $\rightarrow$ `openai`. **H-MoE (full) uses both** $\mathcal{L}_{\text{reg}} + \alpha\mathcal{L}_{\text{local}}$ **and** $\mathcal{L}_{\text{dir}}$**.**

| Configuration | OOD R@100 | Mixing drop | Chaining drop |
|---|---|---|---|
| Only $\mathcal{L}_{\text{dir}}$ | 78.4 | 5.7% | 3.5% |
| H-MoE without $\mathcal{L}_{\text{dir}}$ | 89.8 | 11.1% | 16.7% |
| H-MoE (full) | **93.4** | **2.3%** | **1.8%** |

# E. Algorithm of H-MoE

---

**ALGORITHM 1:** Training Phase of H-MoE

---

**Input:** Paired training data $\mathcal{D}_{\text{train}} = \{(\mathbf{x}_n, \mathbf{y}_n)\}_{n=1}^N$ where $\mathbf{x}_n \in \mathbb{R}^{d_A}$, $\mathbf{y}_n \in \mathbb{R}^{d_B}$; Number of leaves $K$.
**Output:** Base parameters $\theta_{\text{base}}$, Expert set $\mathcal{E} = \{(\mathbf{A}_i, \mathbf{B}_i, \boldsymbol{\mu}_i)\}_{i=1}^{2K-1}$.

*// Stage 1: Global Base Alignment*
1 Initialize base translator $f_{\theta_{\text{base}}}$ (e.g., MLP);
2 $\theta_{\text{base}} \leftarrow \arg\min_\theta \sum_{n=1}^N \|f_\theta(\mathbf{x}_n) - \mathbf{y}_n\|_1$;
3 Freeze $\theta_{\text{base}}$;

*// Stage 2: Hierarchical Manifold Partitioning*
4 Perform hierarchical clustering (e.g., Agglomerative) on $\{\mathbf{x}_n\}$ to build binary tree $\mathcal{T}$ with $K$ leaves;
5 Indexing nodes $i \in \{1, \ldots, 2K-1\}$ (Root is node 1);
6 **for** each node $i$ in $\mathcal{T}$ **do**
7 $\quad$ Identify training subset $\mathcal{C}_i \subseteq \mathcal{D}_{\text{train}}$ associated with node $i$;
$\qquad\qquad\qquad\qquad\qquad$ *// Internal nodes contain union of children's data*;
8 $\quad$ Compute centroid $\boldsymbol{\mu}_i = \frac{1}{|\mathcal{C}_i|} \sum_{(\mathbf{x},\mathbf{y}) \in \mathcal{C}_i} \mathbf{x}$;

*// Stage 3: Expert Specialization (LoRA)*
9 **for** each node $i = 1$ **to** $2K - 1$ **do**
10 $\quad$ Initialize LoRA adapters $\mathbf{A}_i \in \mathbb{R}^{r \times d}$, $\mathbf{B}_i \in \mathbb{R}^{d \times r}$;
11 $\quad$ Define expert $f_i(\mathbf{x}) = f_{\theta_{\text{base}}}(\mathbf{x}) + \mathbf{B}_i \mathbf{A}_i \mathbf{x}$;
$\quad$ *// Optimize with combined Regression + Local Structure loss*
12 $\quad$ $\mathbf{A}_i, \mathbf{B}_i \leftarrow \arg\min_{\mathbf{A},\mathbf{B}} \sum_{(\mathbf{x},\mathbf{y}) \in \mathcal{C}_i} (\mathcal{L}_{\text{reg}} + \alpha\mathcal{L}_{\text{local}})$;
13 **return** $\theta_{\text{base}}, \{(\mathbf{A}_i, \mathbf{B}_i, \boldsymbol{\mu}_i)\}_{i=1}^{2K-1}$;

---

**ALGORITHM 2:** Hierarchical Top-Down Routing & Inference

**Input:** Query $\mathbf{x} \in \mathbb{R}^{d_A}$, Tree structure $\mathcal{T}$, Experts $\{f_i\}$, Centroids $\{\boldsymbol{\mu}_i\}$, Confidence Threshold $\tau \in (0, 1]$.
**Output:** Translated embedding $\hat{\mathbf{y}} \in \mathbb{R}^{d_B}$.

   *// Initialize at Root*
1  $v \leftarrow \text{Root}(\mathcal{T})$;
2  **while** $v$ is not a Leaf **do**
3      Let $u_L, u_R$ be the left and right children of $v$;
4      Compute distances to children centroids: $d_L = \|\mathbf{x} - \boldsymbol{\mu}_{u_L}\|_2, \quad d_R = \|\mathbf{x} - \boldsymbol{\mu}_{u_R}\|_2$;
      *// Check ambiguity ratio (stop if query is equidistant)*
5      $\rho = \min(d_L, d_R) / \max(d_L, d_R)$;
6      **if** $\rho > \tau$ **then**
         *// Ambiguous boundary: stop at current coarser expert*
7         **break**;
8      **else**
         *// Clear preference: descend to closer child*
9         **if** $d_L < d_R$ **then**
10           $v \leftarrow u_L$;
11         **else**
12           $v \leftarrow u_R$;

13  $i_{\text{sel}} \leftarrow v$;

   *// Final Translation with Selected Expert*
14  $\hat{\mathbf{y}} = f_{\theta_{\text{base}}}(\mathbf{x}) + \mathbf{B}_{i_{\text{sel}}} \mathbf{A}_{i_{\text{sel}}} \mathbf{x}$;
15  **return** $\hat{\mathbf{y}}$;

## F. Benchmarks

In this work, we use six datasets following the BEIR (Thakur et al., 2021) benchmarks. Here, we give a brief introduction to these works, and give the statistics of all the datasets.

SciFact (Wadden et al., 2020) is a fact-checking dataset in the scientific domain. It requires models to retrieve evidence from a corpus of 5,183 scientific abstracts to verify given claims. The dataset features an average document length of 213.63 words.

NFCorpus (Boteva et al., 2016) is a bio-medical information retrieval dataset containing 3,633 documents. It is designed to test a model's ability to bridge the lexical gap between natural language queries from non-experts and expert-level medical literature.

ArguAna (Wachsmuth et al., 2018) focuses on argument retrieval from online debate forums. It contains 8,674 documents and is unique within the BEIR benchmark for its long-form queries, which have an average length of 192.98 words.

SciDocs (Cohan et al., 2020) is a citation prediction dataset consisting of 25,657 documents. The task involves identifying relevant scientific papers based on article titles, evaluating the model's capacity to capture citation-based relevance.

FiQA-2018 (Maia et al., 2018) targets the financial domain for question answering. It contains 57,638 documents, requiring systems to retrieve relevant financial opinions and information from forum posts and articles.

Fever (Thorne et al., 2018) is a large-scale fact extraction and verification dataset based on Wikipedia. With a corpus of 5,416,568 documents, it tests the scalability and precision of retrieval systems in verifying natural language claims.

## G. Details of Evaluated Embedding Models

In this appendix, we provide technical specifications and architectural details for the embedding models used in our evaluation. Following the current state-of-the-art on the MTEB leaderboard, we selected a diverse range of models from both open-source and proprietary providers. Table 10 summarizes their key statistics.

- **KaLM-Embedding-Gemma3-12B-2511** (Zhao et al., 2025): A state-of-the-art model from Tencent based on Gemma 3-12B. It features 11.76B parameters and supports Matryoshka Representation Learning (MRL), allowing for flexible

*Table 9.* **Detailed statistics of the six selected datasets from the BEIR benchmark (Thakur et al., 2021). The benchmark covers diverse retrieval tasks and domains to evaluate the zero-shot generalization of models. #Corpus denotes the total number of documents in the collection; #Query indicates the number of queries in the test set; Avg. D/Q is the average number of relevant documents per query; Avg. Q and Avg. D represent the average word length of queries and documents, respectively.**

| Dataset | Task | Domain | #Corpus | #Query | Avg. D/Q | Avg. Q | Avg. D |
|---|---|---|---|---|---|---|---|
| SciFact (Wadden et al., 2020) | Fact Checking | Scientific | 5,183 | 300 | 1.1 | 12.37 | 213.63 |
| NFCorpus (Boteva et al., 2016) | Bio-Medical IR | Bio-Medical | 3,633 | 323 | 38.2 | 3.30 | 232.26 |
| ArguAna (Wachsmuth et al., 2018) | Argument | Misc. | 8,674 | 1,406 | 1.0 | 192.98 | 166.80 |
| SciDocs (Cohan et al., 2020) | Citation-Pred. | Scientific | 25,657 | 1,000 | 4.9 | 9.38 | 176.19 |
| FiQA-2018 (Maia et al., 2018) | Question Answ. | Finance | 57,638 | 648 | 2.6 | 10.77 | 132.32 |
| Fever (Thorne et al., 2018) | Fact Checking | Wikipedia | 5,416,568 | 6,666 | 1.2 | 8.13 | 84.76 |

output dimensions up to 3840 and a context window of 32k tokens.

- **llama-embed-nemotron-8b** (Babakhin et al., 2025): Developed by NVIDIA, this model is built on Llama-3.1-8B using bi-directional attention. It is specifically optimized for multilingual RAG and high-precision semantic similarity.

- **Qwen3-Embedding-8B** (Zhang et al., 2025): The latest embedding series from the Qwen team, optimized for 100+ languages and long-text understanding (32k context). It supports user-defined output dimensions via MRL.

- **gemini-embedding-001** (Google, 2025): Google's flagship embedding model via the Gemini API. It is optimized for multi-task performance and utilizes MRL to provide consistent performance across different vector sizes.

- **Linq-Embed-Mistral** (Kim et al., 2024): Developed by Linq AI Research, this model enhances Mistral-7B through task-specific synthetic data generation and sophisticated negative mining.

- **multilingual-e5-large-instruct** (Wang et al., 2024): A widely used multilingual model based on XLM-RoBERTa (560M parameters). Despite its smaller size, it remains highly competitive in cross-lingual retrieval tasks.

- **SFR-Embedding-Mistral** (Meng et al., 2024): Salesforce's 7B parameter model that improves upon Mistral-7B through multi-stage contrastive training and instruction tuning.

- **GritLM-7B** (Muennighoff et al., 2024): A unified model trained via Generative Representational Instruction Tuning, capable of switching between generative and embedding modes while maintaining SOTA performance.

- **text-embedding-3-small** (OpenAI, 2022): OpenAI's most efficient third-generation model, supporting vector shortening and highly optimized for large-scale retrieval applications.

- **mistral-embed** (Jiang et al., 2023): Mistral AI's official API-based embedding model. It provides a balanced 1024-dimensional vector space optimized for retrieval, clustering, and classification tasks.

*Table 10.* **Summary of Evaluated Embedding Models.**

| Model Name | Abbr. | Params | Dim | Base Model |
|---|---|---|---|---|
| KaLM-Gemma3-12B-2511 | kalm | 11.8B | 3840 | Gemma-3-12B |
| llama-embed-nemotron-8b | nemotron | 8.0B | 4096 | Llama-3.1-8B |
| Qwen3-Embedding-8B | qwen | 8.0B | 4096 | Qwen-3 |
| gemini-embedding-001 | gemini | - | 3072 | Unknown |
| Linq-Embed-Mistral | linq | 7.0B | 4096 | Mistral-7B |
| multilingual-e5-large-instruct | e5 | 0.6B | 1024 | XLM-R-Large |
| SFR-Embedding-Mistral | sfr | 7.0B | 4096 | Mistral-7B |
| GritLM-7B | gritlm | 7.0B | 4096 | Mistral-7B |
| text-embedding-3-small | openai | - | 1536 | Unknown |
| mistral-embed | mistral | - | 1024 | Unknown |

# H. Implementation Details

We utilize the **FiQA-2018 → SciFact** transfer setting for these experiments. Specifically, the translators are trained on the FiQA-2018 dataset (Source Domain) and evaluated on the SciFact dataset (Target Domain), representing an Out-of-Distribution

(OOD) scenario. We report the **Recall@100 Drop (%)** relative to the baseline performance (direct pairwise translation) to quantify the stability of the translation under complex composability scenarios.

**Mixing.** Given the set of embedding models {`nemotron`, `kalm`, `openai` }, we designate `nemotron` as the shared target embedding space. Accordingly, we train translators `kalm` → `nemotron` and `openai` → `nemotron`. At inference time, the test collection is partitioned into three disjoint subsets, each of which is translated by its corresponding translator. The resulting translated embeddings are then merged into a single mixed corpus in the `nemotron` space for retrieval evaluation.

**Chaining.** Given the embedding models {`nemotron` → `openai` via `kalm` }, we consider a mixing setting where `kalm` serves as the target embedding space, and `nemotron` is translated to `kalm` via the intermediate model `openai`. Specifically, we train the translators `nemotron` → `openai` and `openai` → `kalm`. At inference time, we first translate the embeddings from the `nemotron` space to the `openai`, and then translate to the `kalm` space. We finally compare the composite translation recall@100 with the direct `nemotron` → `openai`.

# I. Related Work

## I.1. Sentence Embedding Alignment

A substantial body of work studies cross-lingual embedding alignment (Mikolov et al., 2013; Artetxe et al., 2016; Smith et al., 2017; Joulin et al., 2018). Mikolov et al. (Mikolov et al., 2013) formulate the problem as supervised linear regression over seed word pairs. Subsequent studies introduce orthogonality constraints via Procrustes analysis (`PA`) (Smith et al., 2017), Canonical Correlation Analysis (`CCA`) (Artetxe et al., 2016), and optimal transport objectives such as Gromov–Wasserstein (`GW`) alignment (Chen et al., 2020). These approaches exploit the empirical observation that embeddings exhibit *global geometric similarity* across languages, enabling alignment through a single global transformation.

However, such embedding alignment methods are fundamentally limited in capturing the complex and nonlinear relationships induced by modern embedding models (Jha et al., 2025).

## I.2. Embedding Translation

More recent work directly studies *embedding translation* or embedding transformation, which learns a mapping between embedding spaces produced by different sentence encoders. `Vec2Vec` (Jha et al., 2025) proposes an unsupervised framework for cross-model embedding translation, motivated by the Platonic representation hypothesis (Huh et al., 2024). While effective in practice, `Vec2Vec` treats translation as a black box and does not explain when or why translation fails under distribution shift or composed mappings.

Recent neural approaches, such as Embedding Converter (`EmbConv`) (Yoon & Arik, 2025), further improve translation accuracy by employing expressive translation architectures and adversarial or contrastive objectives, and report strong empirical performance. Nevertheless, these methods still lack interpretable error models and do not account for error amplification under multi-model embedding mixing or translation chaining.

In contrast, our work provides a geometric analysis of embedding translation, derives an explicit error bound that explains failure modes under OOD and multi-model settings, and introduces a geometry-aware translation framework that supports robust and risk-aware deployment in vector databases.

# J. Pairwise Translation Results.

We summarize all translation directions among the 10 embedding models. The complete results are reported in Tables 11–22.

**Impact of different embedding translation methods.** The poor performance of rigid methods such as `GW` and `PA`, as well as global linear models like `CCA`, can be attributed to their inability to accommodate the intrinsic non-linear anisotropy and local manifold divergences present in LLM embeddings. Neural approaches such as `Vec2Vec` offer increased flexibility but exhibit significant training instability due to their dependence on adversarial objectives, which are prone to convergence failures and high variance when the training corpus is limited (evidenced by the high standard deviations marked in red across Tables, e.g., `Vec2Vec` on ArguAna in Group 1). Although `EmbConv` increases expressivity via deeper architectures, its single-stream mapping still lacks the capacity for fine-grained adaptation, struggling to reconcile local semantic discrepancies

and achieve precise alignment in challenging cross-domain scenarios.

**Impact of different embedding model pairs.**    Beyond methodological differences, we observe that *architectural affinity* influences the absolute difficulty of the translation task. Pairs sharing similar backbones or training lineages—such as the `mistral`-based cluster (including `sfr`, `linq`, and `gritlm`)—generally exhibit higher absolute recall ceilings due to their homologous embedding topologies. For instance, translating within this family, such as `mistral` →`gritlm` (Table Group 12), yields an exceptionally high Recall@100 of $83.4\%$. Conversely, translating across fundamentally different architectures, such as from the encoder-only `e5` to decoder-only LLMs (e.g., `e5` →`mistral` in Table Group 7), might presents a relatively larger topological gap, resulting in a comparatively lower baseline performance ($74.5\%$). Crucially, however, our method demonstrates **universal robustness** against these structural divergences. While baselines often collapse on heterogeneous pairs (e.g., `EmbConv` drops to $57.4\%$ on `e5` →`qwen`), `H-MoE` consistently maintains high performance gaps over the runner-up. This confirms that while model similarity aids transfer, our hierarchical expert design can be an important factor in successfully bridging diverse geometries, effectively decoupling translation quality from the ancestral proximity of the source and target models.

## K. Sensitivity Experiments

To verify the robustness of `H-MoE` against hyperparameter variations, we conducted a sensitivity analysis using the FiQA-2018 → SciFact transfer setting. We focused on two representative translation directions: `gemini` → `openai` and `kalm` → `qwen`. Figure 10 illustrates the impact of the cluster count $k$, the local loss weight $\alpha$, and the directional constraint weight $\beta$.

**Impact of Cluster Count ($k$).**    Figure 10a evaluates the effect of the number of experts (clusters) on OOD retrieval performance. We observe that performance improves sharply as $k$ increases from 2 to 4, confirming that a single monolithic model (or too few experts) lacks the capacity to capture the complex local geometry of the embedding manifold. Performance stabilizes and reaches a plateau around $k = 4$ to $k = 8$. While increasing $k$ beyond this point offers diminishing returns, the performance remains stable without significant degradation, indicating that our hierarchical routing is robust to over-segmentation. Based on this, we adopted $k = 4$ as the default for most experiments to balance efficiency and effectiveness.

**Impact of Local Loss Weight ($\alpha$).**    Figure 10b analyzes the sensitivity to $\alpha$, which balances the global alignment loss and the local structure-preserving loss ($\mathcal{L}_{local}$). The results show that performance is suboptimal at $\alpha = 0$ (pure regression), validating the necessity of local geometric constraints for OOD generalization. Recall@100 improves significantly as $\alpha$ increases and remains consistently high within the range $\alpha \in [0.3, 0.6]$. Excessively high values ($\alpha > 0.8$) lead to a slight performance dip for some pairs (*e.g.*, `gemini` → `openai`), likely because the optimization becomes overly constrained by local topology at the expense of global alignment. We set $\alpha = 0.5$ in the experiments.

**Impact of Directional Constraint Weight ($\beta$).**    Figure 10c reports the *performance drop* (negative values denote degradation) for multi-model Mixing and Chaining scenarios as we vary the weight $\beta$ of the directional residual constraint ($\mathcal{L}_{dir}$). At $\beta = 0$ (no directional control), we observe severe performance degradation (*e.g.*, a drop of over $12\%$ for chaining). As $\beta$ increases, the degradation is rapidly mitigated, with the drop converging to near-zero (approx. $-2\%$) when $\beta \in [0.4, 0.9]$. This confirms that enforcing directional consistency on residuals is critical for composability. The wide stable region suggests that `H-MoE` is not sensitive to the precise value of $\beta$, provided it is sufficiently non-zero to regularize the residual space. We set $\beta = 0.7$ in the experiments.

**Impact of the routing parameter ($\tau$).**    Figure 10d visualizes the sensitivity of retrieval recall to the confidence threshold. The performance exhibits an inverted U-shape, consistently peaking at $\tau=0.8$. This demonstrates that a calibrated threshold effectively balances the trade-off between utilizing specialized leaf experts and falling back to robust coarser experts for ambiguous queries. We set $\tau = 0.8$ in the experiments.

*Table 11.* **Pairwise Model Translation Performance (Recall@100) - Group 1. Bold and underline denote the best and runner-up average results. Values highlighted in red indicate high-variance outcomes** (std > 10).

| Method | Dataset | gemini→nemotron | gemini→qwen | gemini→kalm | gemini→linq | gemini→e5 | gemini→sfr | gemini→gritlm |
|---|---|---|---|---|---|---|---|---|
| GW | SciDocs | $0.2_{\pm0.1}$ | $1.2_{\pm0.3}$ | $0.1_{\pm0.4}$ | $4.3_{\pm0.3}$ | $3.1_{\pm0.4}$ | $0.6_{\pm0.1}$ | $3.1_{\pm0.1}$ |
| | FiQA-2018 | $3.7_{\pm0.3}$ | $1.1_{\pm0.3}$ | $1.8_{\pm0.1}$ | $3.7_{\pm0.3}$ | $3.6_{\pm0.0}$ | $1.9_{\pm0.2}$ | $0.9_{\pm0.2}$ |
| | ArguAna | $0.5_{\pm0.2}$ | $1.2_{\pm0.3}$ | $1.7_{\pm0.0}$ | $2.0_{\pm0.3}$ | $1.5_{\pm0.0}$ | $1.4_{\pm0.5}$ | $3.2_{\pm0.3}$ |
| | **Avg.** | 1.5 | 1.1 | 1.2 | 3.3 | 2.7 | 1.3 | 2.4 |
| PA | SciDocs | $14.3_{\pm5.7}$ | $6.0_{\pm2.1}$ | $10.2_{\pm4.6}$ | $10.9_{\pm4.4}$ | $1.2_{\pm0.6}$ | $16.1_{\pm7.1}$ | $10.6_{\pm1.2}$ |
| | FiQA-2018 | $1.5_{\pm0.6}$ | $15.4_{\pm3.8}$ | $4.5_{\pm2.2}$ | $2.0_{\pm0.7}$ | $9.5_{\pm1.6}$ | $8.9_{\pm1.2}$ | $9.5_{\pm3.8}$ |
| | ArguAna | $4.3_{\pm2.1}$ | $4.9_{\pm1.6}$ | $2.4_{\pm0.5}$ | $9.4_{\pm4.6}$ | $2.8_{\pm0.9}$ | $14.8_{\pm2.6}$ | $1.0_{\pm0.5}$ |
| | **Avg.** | 6.7 | 8.8 | 5.7 | 7.4 | 4.5 | 13.2 | 7.1 |
| CCA | SciDocs | $33.1_{\pm8.4}$ | $41.2_{\pm7.8}$ | $40.3_{\pm6.7}$ | $32.3_{\pm5.1}$ | $30.7_{\pm8.3}$ | $33.6_{\pm7.9}$ | $35.8_{\pm8.9}$ |
| | FiQA-2018 | $58.8_{\pm4.3}$ | $60.8_{\pm6.4}$ | $60.3_{\pm5.6}$ | $54.0_{\pm5.2}$ | $56.4_{\pm7.3}$ | $58.1_{\pm4.5}$ | $56.9_{\pm5.7}$ |
| | ArguAna | $65.7_{\pm4.4}$ | $69.0_{\pm8.4}$ | $68.0_{\pm6.2}$ | $67.6_{\pm7.2}$ | $67.9_{\pm8.7}$ | $66.9_{\pm6.3}$ | $69.1_{\pm5.0}$ |
| | **Avg.** | 52.5 | 57.0 | 56.2 | 51.3 | 51.7 | 52.8 | 53.9 |
| Vec2Vec | SciDocs | $28.4_{\pm7.9}$ | $36.7_{\pm12.8}$ | $27.6_{\pm12.9}$ | $34.4_{\pm7.1}$ | $30.1_{\pm12.0}$ | $22.4_{\pm12.2}$ | $24.1_{\pm11.0}$ |
| | FiQA-2018 | $38.5_{\pm9.1}$ | $37.0_{\pm7.2}$ | $35.7_{\pm9.2}$ | $39.4_{\pm10.1}$ | $39.2_{\pm6.6}$ | $31.9_{\pm10.3}$ | $27.9_{\pm10.7}$ |
| | ArguAna | $31.0_{\pm6.4}$ | $24.3_{\pm8.8}$ | $39.1_{\pm13.0}$ | $35.0_{\pm6.4}$ | $31.9_{\pm10.7}$ | $31.7_{\pm9.7}$ | $26.3_{\pm11.3}$ |
| | **Avg.** | 32.6 | 32.7 | 34.1 | 36.2 | 33.7 | 28.7 | 26.1 |
| EmbConv | SciDocs | $43.2_{\pm3.5}$ | $48.6_{\pm1.4}$ | $49.2_{\pm2.7}$ | $35.9_{\pm1.5}$ | $42.9_{\pm1.0}$ | $44.0_{\pm3.2}$ | $39.3_{\pm3.0}$ |
| | FiQA-2018 | $69.9_{\pm1.5}$ | $64.5_{\pm2.5}$ | $70.4_{\pm1.3}$ | $64.9_{\pm2.2}$ | $62.6_{\pm1.7}$ | $63.8_{\pm3.1}$ | $61.4_{\pm1.5}$ |
| | ArguAna | $69.9_{\pm3.2}$ | $74.7_{\pm1.0}$ | $73.9_{\pm1.7}$ | $70.9_{\pm1.2}$ | $74.1_{\pm2.1}$ | $77.8_{\pm1.3}$ | $78.1_{\pm3.0}$ |
| | **Avg.** | 61.0 | 62.6 | 64.5 | 57.2 | 59.9 | 61.9 | 59.6 |
| **H−MoE (Ours)** | SciDocs | $59.4_{\pm1.9}$ | $62.5_{\pm3.3}$ | $62.4_{\pm3.3}$ | $52.1_{\pm3.6}$ | $55.4_{\pm3.3}$ | $56.5_{\pm3.9}$ | $58.7_{\pm3.8}$ |
| | FiQA-2018 | $79.9_{\pm1.0}$ | $81.7_{\pm0.6}$ | $81.6_{\pm3.3}$ | $77.9_{\pm1.3}$ | $77.9_{\pm0.6}$ | $78.0_{\pm2.7}$ | $78.9_{\pm3.9}$ |
| | ArguAna | $90.5_{\pm3.5}$ | $90.7_{\pm1.7}$ | $89.8_{\pm3.3}$ | $89.2_{\pm2.1}$ | $88.5_{\pm1.6}$ | $88.9_{\pm3.9}$ | $89.0_{\pm3.9}$ |
| | **Avg.** | **76.6** | **78.3** | **77.9** | **73.1** | **73.9** | **74.5** | **75.6** |

*Table 12.* **Pairwise Model Translation Performance (Recall@100) - Group 2. Bold and underline denote the best and runner-up average results. Values highlighted in red indicate high-variance outcomes** (std > 10).

| Method | Dataset | gemini→mistral | nemotron→gemini | nemotron→qwen | nemotron→kalm | nemotron→e5 | nemotron→sfr | nemotron→gritlm |
|---|---|---|---|---|---|---|---|---|
| GW | SciDocs | $0.6_{\pm0.2}$ | $3.6_{\pm0.0}$ | $3.4_{\pm0.2}$ | $1.5_{\pm0.1}$ | $0.1_{\pm0.1}$ | $4.5_{\pm0.4}$ | $3.4_{\pm0.1}$ |
| | FiQA-2018 | $1.6_{\pm0.3}$ | $2.7_{\pm0.4}$ | $4.3_{\pm0.4}$ | $4.7_{\pm0.3}$ | $2.9_{\pm0.3}$ | $0.5_{\pm0.3}$ | $0.2_{\pm0.1}$ |
| | ArguAna | $2.0_{\pm0.3}$ | $3.4_{\pm0.2}$ | $1.6_{\pm0.2}$ | $3.1_{\pm0.2}$ | $2.3_{\pm0.1}$ | $1.7_{\pm0.4}$ | $4.3_{\pm0.0}$ |
| | **Avg.** | 1.4 | 3.2 | 3.1 | 3.1 | 1.8 | 2.3 | 2.6 |
| PA | SciDocs | $11.0_{\pm1.5}$ | $2.7_{\pm0.9}$ | $13.7_{\pm4.7}$ | $14.1_{\pm4.1}$ | $9.2_{\pm3.1}$ | $4.7_{\pm1.8}$ | $15.3_{\pm4.7}$ |
| | FiQA-2018 | $8.8_{\pm1.5}$ | $16.1_{\pm4.4}$ | $10.4_{\pm5.0}$ | $12.7_{\pm3.3}$ | $2.2_{\pm0.7}$ | $2.0_{\pm0.7}$ | $14.5_{\pm5.7}$ |
| | ArguAna | $1.1_{\pm0.5}$ | $4.9_{\pm2.2}$ | $10.7_{\pm4.0}$ | $4.9_{\pm1.7}$ | $6.1_{\pm1.9}$ | $7.6_{\pm1.2}$ | $1.1_{\pm0.5}$ |
| | **Avg.** | 6.9 | 7.9 | 11.6 | 10.6 | 5.8 | 4.7 | 10.3 |
| CCA | SciDocs | $34.9_{\pm8.8}$ | $38.4_{\pm5.7}$ | $47.3_{\pm6.8}$ | $42.2_{\pm7.8}$ | $30.9_{\pm5.7}$ | $36.9_{\pm4.6}$ | $43.8_{\pm4.6}$ |
| | FiQA-2018 | $55.8_{\pm6.7}$ | $58.0_{\pm4.8}$ | $60.4_{\pm4.7}$ | $58.1_{\pm8.2}$ | $54.8_{\pm7.3}$ | $57.2_{\pm5.1}$ | $59.1_{\pm4.5}$ |
| | ArguAna | $68.8_{\pm5.1}$ | $69.0_{\pm7.1}$ | $71.6_{\pm5.9}$ | $69.9_{\pm4.1}$ | $64.9_{\pm5.0}$ | $66.2_{\pm4.8}$ | $68.8_{\pm7.4}$ |
| | **Avg.** | 53.1 | 55.1 | 59.8 | 56.7 | 50.2 | 53.5 | 57.3 |
| Vec2Vec | SciDocs | $27.9_{\pm11.0}$ | $22.4_{\pm12.9}$ | $36.4_{\pm6.7}$ | $25.4_{\pm11.7}$ | $21.7_{\pm12.2}$ | $24.2_{\pm11.6}$ | $34.4_{\pm9.4}$ |
| | FiQA-2018 | $33.8_{\pm10.4}$ | $26.9_{\pm9.0}$ | $32.7_{\pm11.8}$ | $29.7_{\pm9.0}$ | $29.3_{\pm10.3}$ | $37.6_{\pm10.9}$ | $33.8_{\pm10.5}$ |
| | ArguAna | $35.4_{\pm12.0}$ | $30.1_{\pm8.4}$ | $28.7_{\pm11.1}$ | $25.4_{\pm12.0}$ | $20.6_{\pm12.0}$ | $20.2_{\pm12.6}$ | $35.9_{\pm6.7}$ |
| | **Avg.** | 32.4 | 26.5 | 32.6 | 26.8 | 23.9 | 27.4 | 34.7 |
| EmbConv | SciDocs | $41.2_{\pm1.8}$ | $46.4_{\pm1.6}$ | $54.8_{\pm2.8}$ | $53.7_{\pm2.3}$ | $37.0_{\pm2.4}$ | $45.3_{\pm1.2}$ | $48.9_{\pm2.5}$ |
| | FiQA-2018 | $65.1_{\pm2.3}$ | $67.0_{\pm1.3}$ | $70.3_{\pm1.7}$ | $61.9_{\pm2.4}$ | $64.0_{\pm1.5}$ | $69.4_{\pm1.4}$ | $71.7_{\pm2.3}$ |
| | ArguAna | $72.1_{\pm1.8}$ | $72.1_{\pm1.5}$ | $79.1_{\pm1.6}$ | $79.9_{\pm3.2}$ | $68.8_{\pm2.2}$ | $76.8_{\pm3.5}$ | $76.8_{\pm3.2}$ |
| | **Avg.** | 59.4 | 61.8 | 68.1 | 65.2 | 56.6 | 63.8 | 65.8 |
| **H−MoE (Ours)** | SciDocs | $59.6_{\pm1.6}$ | $58.1_{\pm1.2}$ | $69.2_{\pm2.1}$ | $64.9_{\pm2.1}$ | $56.1_{\pm3.8}$ | $59.3_{\pm0.7}$ | $66.9_{\pm0.9}$ |
| | FiQA-2018 | $79.9_{\pm2.6}$ | $80.0_{\pm1.0}$ | $84.7_{\pm1.5}$ | $82.5_{\pm3.7}$ | $78.3_{\pm2.4}$ | $79.0_{\pm1.6}$ | $83.7_{\pm2.9}$ |
| | ArguAna | $90.1_{\pm3.8}$ | $89.0_{\pm0.8}$ | $91.4_{\pm3.0}$ | $91.4_{\pm2.7}$ | $89.4_{\pm3.7}$ | $90.2_{\pm2.5}$ | $92.2_{\pm2.3}$ |
| | **Avg.** | **76.6** | **75.7** | **81.8** | **79.6** | **74.6** | **76.2** | **80.9** |

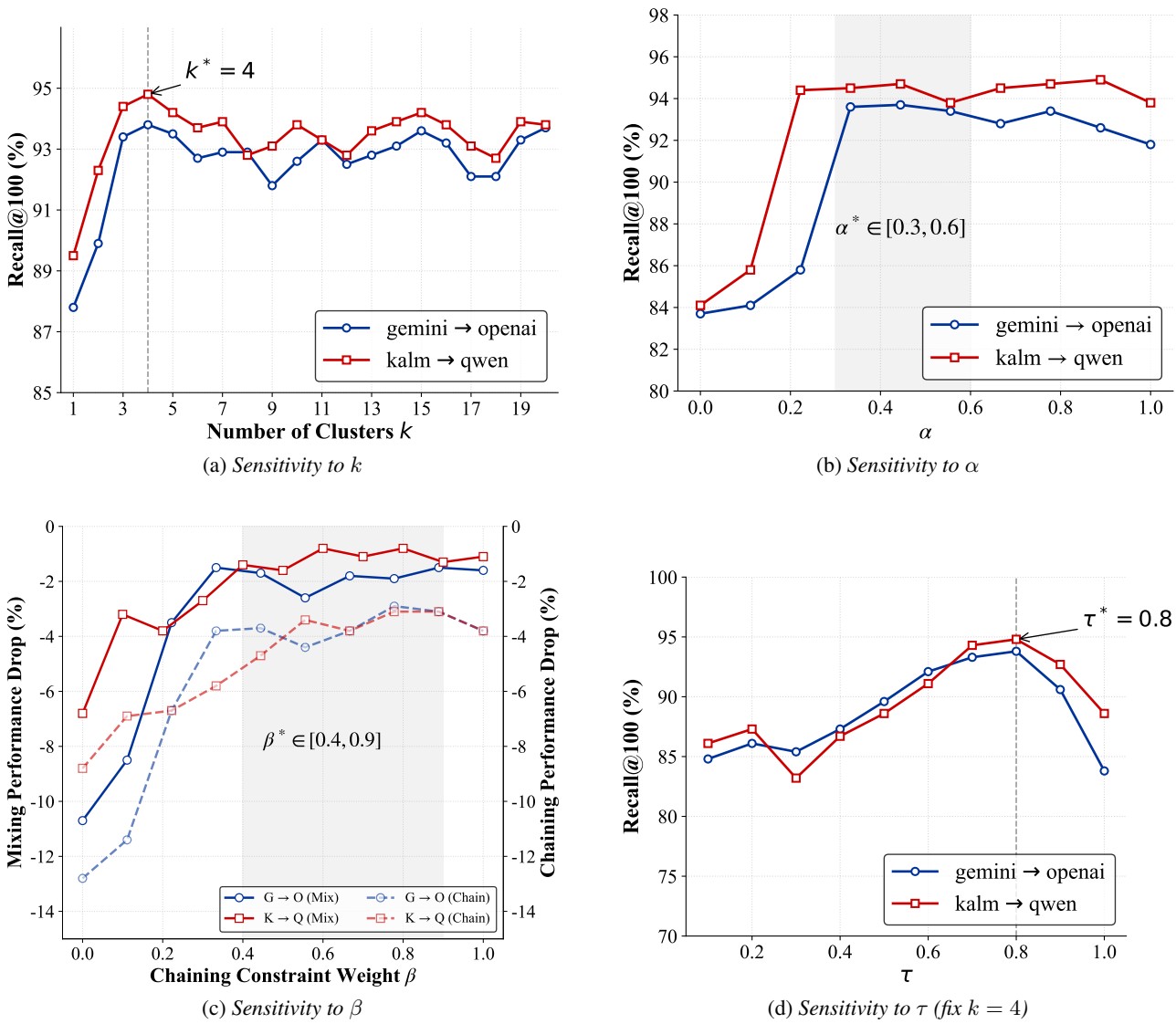

*Figure 10.* **Hyperparameter Sensitivity Analysis. Evaluation under** FiQA-2018 $\rightarrow$ SciFact **transfer setting. (a) Number of clusters** $k$**: performance becomes stable after** $k = 4$**. (b) Local loss weight** $\alpha$**: results are consistent when** $\alpha \in [0.3, 0.6]$**, balancing regression and local structure preservation. (c) Directional constraint weight** $\beta$**: larger** $\beta$ **effectively reduces performance drop in mixing and chaining scenarios. (d) Routing threshold** $\tau$ **(fix** $k = 4$**): controls the hierarchical traversal depth; a balanced** $\tau$ **ensures expert specialization without over-routing.**

*Table 13.* **Pairwise Model Translation Performance (Recall@100) - Group 3. Bold and** underline **denote the best and runner-up average results. Values highlighted in** red **indicate high-variance outcomes** (std > 10)**.**

| Method | Dataset | nemotron→openai | nemotron→mistral | qwen→nemotron | qwen→kalm | qwen→linq | qwen→e5 | qwen→sfr |
|---|---|---|---|---|---|---|---|---|
| GW | SciDocs | $0.5\pm_{0.0}$ | $4.8\pm_{0.5}$ | $1.0\pm_{0.3}$ | $1.7\pm_{0.3}$ | $4.9\pm_{0.2}$ | $3.8\pm_{0.3}$ | $4.9\pm_{0.3}$ |
| | FiQA-2018 | $1.7\pm_{0.1}$ | $1.4\pm_{0.0}$ | $2.5\pm_{0.0}$ | $4.7\pm_{0.4}$ | $4.9\pm_{0.3}$ | $1.1\pm_{0.3}$ | $0.9\pm_{0.3}$ |
| | ArguAna | $4.8\pm_{0.4}$ | $2.4\pm_{0.3}$ | $1.7\pm_{0.3}$ | $2.3\pm_{0.2}$ | $2.1\pm_{0.1}$ | $2.2\pm_{0.3}$ | $0.6\pm_{0.5}$ |
| | **Avg.** | 2.3 | 2.9 | 1.7 | 2.9 | 4.0 | 2.4 | 2.1 |
| PA | SciDocs | $8.3\pm_{0.9}$ | $4.9\pm_{1.6}$ | $7.6\pm_{3.6}$ | $3.3\pm_{1.1}$ | $10.0\pm_{1.6}$ | $14.0\pm_{6.9}$ | $9.3\pm_{2.4}$ |
| | FiQA-2018 | $8.9\pm_{3.3}$ | $15.6\pm_{4.2}$ | $14.0\pm_{2.3}$ | $9.7\pm_{2.9}$ | $1.7\pm_{0.8}$ | $1.6\pm_{0.6}$ | $7.5\pm_{2.6}$ |
| | ArguAna | $5.3\pm_{2.3}$ | $14.7\pm_{2.4}$ | $10.3\pm_{5.1}$ | $6.1\pm_{1.0}$ | $13.8\pm_{3.0}$ | $14.7\pm_{6.5}$ | $8.4\pm_{4.1}$ |
| | **Avg.** | 7.5 | 11.7 | 10.6 | 6.4 | 8.5 | 10.1 | 8.4 |
| CCA | SciDocs | $37.8\pm_{7.2}$ | $46.4\pm_{4.2}$ | $46.3\pm_{8.4}$ | $46.4\pm_{6.7}$ | $33.6\pm_{4.7}$ | $35.3\pm_{6.2}$ | $36.1\pm_{6.4}$ |
| | FiQA-2018 | $57.3\pm_{8.3}$ | $60.5\pm_{8.6}$ | $61.0\pm_{8.5}$ | $59.5\pm_{4.8}$ | $58.3\pm_{5.3}$ | $55.2\pm_{7.2}$ | $56.3\pm_{6.9}$ |
| | ArguAna | $68.4\pm_{7.6}$ | $69.1\pm_{6.2}$ | $67.0\pm_{5.5}$ | $66.1\pm_{7.1}$ | $64.7\pm_{8.5}$ | $67.7\pm_{8.2}$ | $69.2\pm_{8.9}$ |
| | **Avg.** | 54.5 | 58.7 | 58.1 | 57.3 | 52.2 | 52.8 | 53.9 |
| Vec2Vec | SciDocs | $25.5\pm_{10.0}$ | $36.0\pm_{10.2}$ | $32.6\pm_{12.3}$ | $34.9\pm_{7.2}$ | $36.5\pm_{9.4}$ | $37.2\pm_{11.1}$ | $29.1\pm_{10.7}$ |
| | FiQA-2018 | $34.9\pm_{6.3}$ | $32.7\pm_{10.3}$ | $26.2\pm_{9.1}$ | $30.8\pm_{12.8}$ | $34.9\pm_{8.4}$ | $39.3\pm_{11.6}$ | $31.7\pm_{11.8}$ |
| | ArguAna | $29.9\pm_{12.3}$ | $34.4\pm_{10.6}$ | $34.6\pm_{6.6}$ | $38.3\pm_{11.8}$ | $39.3\pm_{7.0}$ | $35.8\pm_{6.1}$ | $22.2\pm_{7.6}$ |
| | **Avg.** | 30.1 | 34.4 | 31.1 | 34.7 | 36.9 | 37.5 | 27.7 |
| EmbConv | SciDocs | $41.8\pm_{2.4}$ | $58.6\pm_{3.4}$ | $53.1\pm_{3.4}$ | $47.6\pm_{1.1}$ | $39.0\pm_{1.9}$ | $40.1\pm_{1.1}$ | $45.9\pm_{3.2}$ |
| | FiQA-2018 | $69.5\pm_{1.2}$ | $69.2\pm_{1.2}$ | $67.0\pm_{1.4}$ | $68.6\pm_{3.4}$ | $67.6\pm_{1.3}$ | $66.6\pm_{1.3}$ | $67.8\pm_{3.3}$ |
| | ArguAna | $71.8\pm_{1.8}$ | $81.3\pm_{2.1}$ | $80.7\pm_{2.2}$ | $81.1\pm_{2.7}$ | $75.8\pm_{2.7}$ | $78.6\pm_{2.9}$ | $70.1\pm_{1.4}$ |
| | **Avg.** | 61.0 | 69.7 | 66.9 | 65.8 | 60.8 | 61.7 | 61.3 |
| **H−MoE (Ours)** | SciDocs | $60.8\pm_{1.4}$ | $70.8\pm_{1.3}$ | $70.3\pm_{3.1}$ | $67.3\pm_{3.8}$ | $58.6\pm_{3.9}$ | $55.8\pm_{3.7}$ | $58.2\pm_{2.7}$ |
| | FiQA-2018 | $80.2\pm_{2.8}$ | $84.2\pm_{1.1}$ | $84.5\pm_{3.2}$ | $82.4\pm_{3.2}$ | $78.8\pm_{2.2}$ | $77.6\pm_{0.8}$ | $79.8\pm_{1.6}$ |
| | ArguAna | $89.3\pm_{0.6}$ | $91.0\pm_{3.1}$ | $91.9\pm_{0.9}$ | $92.1\pm_{2.3}$ | $89.5\pm_{2.9}$ | $89.3\pm_{0.7}$ | $90.7\pm_{2.4}$ |
| | **Avg.** | **76.8** | **82.0** | **82.2** | **80.6** | **75.6** | **74.2** | **76.2** |

*Table 14.* **Pairwise Model Translation Performance (Recall@100) - Group 4. Bold and** underline **denote the best and runner-up average results. Values highlighted in** red **indicate high-variance outcomes** (std > 10)**.**

| Method | Dataset | qwen→gritlm | qwen→openai | qwen→mistral | kalm→gemini | kalm→nemotron | kalm→qwen | kalm→linq |
|---|---|---|---|---|---|---|---|---|
| GW | SciDocs | $3.1\pm_{0.0}$ | $1.2\pm_{0.2}$ | $4.6\pm_{0.0}$ | $2.9\pm_{0.2}$ | $3.0\pm_{0.2}$ | $3.8\pm_{0.3}$ | $4.7\pm_{0.4}$ |
| | FiQA-2018 | $4.3\pm_{0.3}$ | $2.1\pm_{0.0}$ | $0.6\pm_{0.0}$ | $1.8\pm_{0.3}$ | $1.1\pm_{0.3}$ | $4.2\pm_{0.5}$ | $1.0\pm_{0.4}$ |
| | ArguAna | $3.6\pm_{0.1}$ | $3.4\pm_{0.2}$ | $4.1\pm_{0.3}$ | $4.4\pm_{0.4}$ | $0.6\pm_{0.4}$ | $3.1\pm_{0.3}$ | $1.8\pm_{0.0}$ |
| | **Avg.** | 3.7 | 2.2 | 3.1 | 3.0 | 1.6 | 3.7 | 2.5 |
| PA | SciDocs | $9.9\pm_{2.3}$ | $6.8\pm_{2.6}$ | $5.3\pm_{1.9}$ | $11.7\pm_{3.1}$ | $3.2\pm_{0.8}$ | $14.2\pm_{5.8}$ | $10.1\pm_{3.7}$ |
| | FiQA-2018 | $11.0\pm_{0.7}$ | $10.0\pm_{0.9}$ | $8.0\pm_{1.5}$ | $1.7\pm_{0.6}$ | $1.1\pm_{0.5}$ | $14.6\pm_{0.8}$ | $2.8\pm_{0.5}$ |
| | ArguAna | $4.4\pm_{1.2}$ | $3.5\pm_{1.3}$ | $2.8\pm_{0.9}$ | $1.1\pm_{0.5}$ | $7.0\pm_{2.6}$ | $15.6\pm_{6.8}$ | $13.7\pm_{4.4}$ |
| | **Avg.** | 8.5 | 6.8 | 5.4 | 4.9 | 3.8 | 14.8 | 8.8 |
| CCA | SciDocs | $43.4\pm_{4.4}$ | $39.6\pm_{4.8}$ | $43.9\pm_{5.2}$ | $39.6\pm_{7.8}$ | $42.0\pm_{4.8}$ | $44.7\pm_{4.3}$ | $38.7\pm_{8.0}$ |
| | FiQA-2018 | $62.8\pm_{6.9}$ | $60.3\pm_{6.5}$ | $61.6\pm_{7.0}$ | $59.2\pm_{8.6}$ | $59.0\pm_{5.5}$ | $63.6\pm_{8.3}$ | $59.4\pm_{8.0}$ |
| | ArguAna | $69.4\pm_{5.6}$ | $69.3\pm_{7.6}$ | $68.6\pm_{4.9}$ | $67.2\pm_{7.1}$ | $70.0\pm_{6.5}$ | $69.0\pm_{6.5}$ | $68.9\pm_{6.4}$ |
| | **Avg.** | 58.5 | 56.4 | 58.0 | 55.3 | 57.0 | 59.1 | 55.7 |
| Vec2Vec | SciDocs | $21.2\pm_{6.7}$ | $28.8\pm_{9.7}$ | $32.9\pm_{6.5}$ | $26.7\pm_{10.1}$ | $26.2\pm_{8.6}$ | $32.9\pm_{9.0}$ | $22.8\pm_{6.5}$ |
| | FiQA-2018 | $26.9\pm_{11.0}$ | $27.5\pm_{8.0}$ | $33.6\pm_{8.0}$ | $27.0\pm_{12.8}$ | $36.3\pm_{10.7}$ | $27.3\pm_{7.3}$ | $22.0\pm_{11.4}$ |
| | ArguAna | $23.5\pm_{7.7}$ | $26.8\pm_{10.2}$ | $33.1\pm_{12.3}$ | $27.8\pm_{10.2}$ | $34.8\pm_{10.8}$ | $39.0\pm_{9.4}$ | $37.7\pm_{6.3}$ |
| | **Avg.** | 23.9 | 27.7 | 33.2 | 27.2 | 32.4 | 33.0 | 27.5 |
| EmbConv | SciDocs | $49.3\pm_{1.3}$ | $47.3\pm_{2.7}$ | $48.9\pm_{2.2}$ | $46.0\pm_{3.4}$ | $53.5\pm_{3.1}$ | $47.7\pm_{1.6}$ | $48.8\pm_{1.1}$ |
| | FiQA-2018 | $71.5\pm_{3.1}$ | $65.9\pm_{1.7}$ | $70.3\pm_{3.4}$ | $63.5\pm_{1.3}$ | $70.0\pm_{2.9}$ | $69.7\pm_{1.1}$ | $63.4\pm_{2.6}$ |
| | ArguAna | $73.8\pm_{1.4}$ | $76.2\pm_{2.3}$ | $77.7\pm_{1.7}$ | $73.3\pm_{3.2}$ | $71.3\pm_{1.4}$ | $75.4\pm_{1.6}$ | $79.4\pm_{1.7}$ |
| | **Avg.** | 64.9 | 63.2 | 65.6 | 61.0 | 64.9 | 64.3 | 63.9 |
| **H−MoE (Ours)** | SciDocs | $68.0\pm_{2.9}$ | $60.7\pm_{1.4}$ | $66.6\pm_{1.5}$ | $61.5\pm_{0.9}$ | $63.9\pm_{1.0}$ | $67.7\pm_{0.8}$ | $62.5\pm_{2.4}$ |
| | FiQA-2018 | $82.6\pm_{3.0}$ | $81.6\pm_{2.4}$ | $82.9\pm_{0.7}$ | $81.7\pm_{2.6}$ | $82.0\pm_{2.7}$ | $84.3\pm_{0.7}$ | $80.7\pm_{2.1}$ |
| | ArguAna | $91.2\pm_{0.5}$ | $90.7\pm_{2.3}$ | $91.2\pm_{3.3}$ | $89.3\pm_{2.3}$ | $90.3\pm_{2.0}$ | $91.4\pm_{2.3}$ | $90.5\pm_{3.9}$ |
| | **Avg.** | **80.6** | **77.6** | **80.2** | **77.5** | **78.7** | **81.1** | **77.9** |

*Table 15.* **Pairwise Model Translation Performance (Recall@100) - Group 5. Bold and underline denote the best and runner-up average results. Values highlighted in red indicate high-variance outcomes** (std > 10).

| Method | Dataset | kalm→e5 | kalm→sfr | kalm→gritlm | kalm→openai | kalm→mistral | linq→gemini | linq→nemotron |
|---|---|---|---|---|---|---|---|---|
| GW | SciDocs | $0.2_{\pm0.4}$ | $2.8_{\pm0.4}$ | $3.9_{\pm0.2}$ | $2.8_{\pm0.1}$ | $1.9_{\pm0.4}$ | $2.6_{\pm0.4}$ | $3.3_{\pm0.2}$ |
| | FiQA-2018 | $3.7_{\pm0.3}$ | $3.9_{\pm0.2}$ | $0.3_{\pm0.1}$ | $1.0_{\pm0.2}$ | $4.5_{\pm0.2}$ | $0.5_{\pm0.3}$ | $2.2_{\pm0.3}$ |
| | ArguAna | $3.7_{\pm0.2}$ | $3.2_{\pm0.4}$ | $4.7_{\pm0.3}$ | $4.3_{\pm0.2}$ | $2.5_{\pm0.2}$ | $2.8_{\pm0.2}$ | $0.3_{\pm0.4}$ |
| | **Avg.** | 2.5 | 3.3 | 3.0 | 2.7 | 3.0 | 2.0 | 1.9 |
| PA | SciDocs | $16.8_{\pm2.6}$ | $16.1_{\pm1.6}$ | $10.0_{\pm3.2}$ | $11.3_{\pm1.8}$ | $8.1_{\pm4.0}$ | $8.0_{\pm0.8}$ | $14.5_{\pm5.3}$ |
| | FiQA-2018 | $16.0_{\pm1.1}$ | $13.3_{\pm0.7}$ | $16.0_{\pm3.3}$ | $12.6_{\pm3.4}$ | $15.5_{\pm4.1}$ | $14.4_{\pm2.6}$ | $11.1_{\pm4.5}$ |
| | ArguAna | $6.7_{\pm2.8}$ | $13.3_{\pm2.8}$ | $10.1_{\pm2.9}$ | $9.9_{\pm1.7}$ | $14.5_{\pm3.5}$ | $2.0_{\pm0.8}$ | $2.9_{\pm0.7}$ |
| | **Avg.** | 13.2 | 14.2 | 12.0 | 11.3 | 12.7 | 8.1 | 9.5 |
| CCA | SciDocs | $34.0_{\pm4.1}$ | $36.2_{\pm7.6}$ | $40.9_{\pm5.7}$ | $42.1_{\pm4.6}$ | $41.0_{\pm5.3}$ | $27.5_{\pm4.8}$ | $34.6_{\pm7.2}$ |
| | FiQA-2018 | $57.9_{\pm8.0}$ | $57.1_{\pm8.1}$ | $59.0_{\pm7.2}$ | $59.4_{\pm5.6}$ | $59.6_{\pm4.6}$ | $53.9_{\pm7.0}$ | $55.3_{\pm7.8}$ |
| | ArguAna | $66.7_{\pm4.4}$ | $64.5_{\pm7.2}$ | $65.3_{\pm6.2}$ | $68.9_{\pm8.7}$ | $67.7_{\pm6.5}$ | $66.4_{\pm8.1}$ | $65.7_{\pm6.5}$ |
| | **Avg.** | 52.9 | 52.6 | 55.1 | 56.8 | 56.1 | 49.3 | 51.9 |
| Vec2Vec | SciDocs | $25.8_{\pm12.7}$ | $29.7_{\pm7.4}$ | $27.4_{\pm9.9}$ | $32.3_{\pm7.7}$ | $30.3_{\pm9.5}$ | $21.0_{\pm9.5}$ | $34.6_{\pm11.3}$ |
| | FiQA-2018 | $36.1_{\pm10.8}$ | $26.3_{\pm11.0}$ | $29.3_{\pm7.9}$ | $23.2_{\pm8.3}$ | $20.5_{\pm12.6}$ | $33.1_{\pm9.3}$ | $28.0_{\pm10.0}$ |
| | ArguAna | $21.5_{\pm6.4}$ | $37.2_{\pm12.8}$ | $21.9_{\pm8.0}$ | $26.3_{\pm11.3}$ | $39.3_{\pm7.5}$ | $36.9_{\pm9.0}$ | $31.1_{\pm6.5}$ |
| | **Avg.** | 27.8 | 31.1 | 26.2 | 27.3 | 30.0 | 30.3 | 31.2 |
| EmbConv | SciDocs | $41.0_{\pm1.7}$ | $41.8_{\pm2.9}$ | $49.8_{\pm2.5}$ | $47.8_{\pm3.1}$ | $49.6_{\pm3.0}$ | $37.7_{\pm2.8}$ | $45.8_{\pm3.4}$ |
| | FiQA-2018 | $63.8_{\pm1.3}$ | $65.2_{\pm2.3}$ | $68.9_{\pm2.5}$ | $63.9_{\pm1.7}$ | $65.7_{\pm1.1}$ | $65.7_{\pm2.6}$ | $67.5_{\pm1.8}$ |
| | ArguAna | $76.7_{\pm3.0}$ | $77.9_{\pm1.8}$ | $70.2_{\pm2.8}$ | $77.2_{\pm2.7}$ | $71.1_{\pm3.0}$ | $70.4_{\pm1.4}$ | $71.5_{\pm2.0}$ |
| | **Avg.** | 60.5 | 61.7 | 63.0 | 62.9 | 62.1 | 57.9 | 61.6 |
| **H−MoE (Ours)** | SciDocs | $59.2_{\pm3.7}$ | $61.0_{\pm1.6}$ | $65.3_{\pm1.7}$ | $61.9_{\pm2.6}$ | $65.4_{\pm1.5}$ | $52.6_{\pm3.8}$ | $55.5_{\pm3.9}$ |
| | FiQA-2018 | $80.4_{\pm3.1}$ | $81.2_{\pm2.3}$ | $81.8_{\pm3.2}$ | $80.3_{\pm1.4}$ | $81.7_{\pm1.2}$ | $77.8_{\pm1.5}$ | $79.1_{\pm3.8}$ |
| | ArguAna | $88.8_{\pm1.0}$ | $89.8_{\pm1.1}$ | $91.2_{\pm2.8}$ | $90.3_{\pm1.3}$ | $91.2_{\pm1.6}$ | $88.0_{\pm2.9}$ | $89.7_{\pm1.5}$ |
| | **Avg.** | **76.1** | **77.3** | **79.4** | **77.5** | **79.4** | **72.8** | **74.8** |

*Table 16.* **Pairwise Model Translation Performance (Recall@100) - Group 6. Bold and underline denote the best and runner-up average results. Values highlighted in red indicate high-variance outcomes** (std > 10).

| Method | Dataset | linq→qwen | linq→kalm | linq→sfr | linq→openai | linq→mistral | e5→gemini | e5→nemotron |
|---|---|---|---|---|---|---|---|---|
| GW | SciDocs | $3.6_{\pm0.2}$ | $0.8_{\pm0.3}$ | $3.6_{\pm0.1}$ | $2.4_{\pm0.1}$ | $3.6_{\pm0.5}$ | $2.5_{\pm0.5}$ | $1.5_{\pm0.5}$ |
| | FiQA-2018 | $1.4_{\pm0.2}$ | $1.3_{\pm0.3}$ | $0.1_{\pm0.3}$ | $1.7_{\pm0.3}$ | $1.0_{\pm0.4}$ | $4.6_{\pm0.4}$ | $2.5_{\pm0.5}$ |
| | ArguAna | $2.7_{\pm0.5}$ | $3.2_{\pm0.0}$ | $2.1_{\pm0.5}$ | $0.5_{\pm0.3}$ | $4.6_{\pm0.5}$ | $2.9_{\pm0.1}$ | $2.4_{\pm0.1}$ |
| | **Avg.** | 2.6 | 1.8 | 1.9 | 1.6 | 3.1 | 3.3 | 2.1 |
| PA | SciDocs | $3.6_{\pm1.1}$ | $14.5_{\pm6.6}$ | $4.9_{\pm1.0}$ | $11.9_{\pm0.8}$ | $2.9_{\pm1.3}$ | $7.6_{\pm2.3}$ | $2.3_{\pm0.8}$ |
| | FiQA-2018 | $3.9_{\pm1.6}$ | $9.1_{\pm0.9}$ | $12.9_{\pm6.2}$ | $5.8_{\pm1.5}$ | $10.7_{\pm1.6}$ | $9.5_{\pm3.3}$ | $2.2_{\pm0.8}$ |
| | ArguAna | $10.0_{\pm3.9}$ | $3.9_{\pm0.7}$ | $12.4_{\pm0.6}$ | $15.1_{\pm5.5}$ | $3.4_{\pm1.4}$ | $4.9_{\pm2.0}$ | $9.7_{\pm2.7}$ |
| | **Avg.** | 5.8 | 9.2 | 10.1 | 11.0 | 5.7 | 7.3 | 4.7 |
| CCA | SciDocs | $38.1_{\pm5.6}$ | $39.9_{\pm6.6}$ | $41.3_{\pm6.9}$ | $36.5_{\pm6.0}$ | $34.1_{\pm8.5}$ | $32.4_{\pm6.6}$ | $33.2_{\pm8.1}$ |
| | FiQA-2018 | $59.1_{\pm5.0}$ | $60.5_{\pm6.3}$ | $59.5_{\pm4.3}$ | $55.2_{\pm8.9}$ | $56.0_{\pm7.3}$ | $57.8_{\pm4.0}$ | $56.2_{\pm5.4}$ |
| | ArguAna | $69.4_{\pm6.6}$ | $67.2_{\pm6.0}$ | $67.1_{\pm4.3}$ | $67.3_{\pm9.0}$ | $66.5_{\pm5.7}$ | $66.7_{\pm7.4}$ | $66.4_{\pm4.6}$ |
| | **Avg.** | 55.5 | 55.9 | 56.0 | 53.0 | 52.2 | 52.3 | 51.9 |
| Vec2Vec | SciDocs | $28.4_{\pm11.4}$ | $21.0_{\pm11.2}$ | $36.3_{\pm11.1}$ | $24.4_{\pm8.1}$ | $39.4_{\pm12.6}$ | $29.5_{\pm11.9}$ | $22.7_{\pm7.7}$ |
| | FiQA-2018 | $20.3_{\pm7.4}$ | $24.7_{\pm7.5}$ | $32.0_{\pm10.7}$ | $39.0_{\pm12.2}$ | $35.7_{\pm8.2}$ | $34.2_{\pm11.6}$ | $32.9_{\pm12.5}$ |
| | ArguAna | $20.7_{\pm10.2}$ | $21.1_{\pm9.5}$ | $26.0_{\pm7.0}$ | $32.5_{\pm9.0}$ | $23.0_{\pm8.6}$ | $31.2_{\pm12.8}$ | $21.8_{\pm11.1}$ |
| | **Avg.** | 23.1 | 22.3 | 31.4 | 32.0 | 32.7 | 31.6 | 25.8 |
| EmbConv | SciDocs | $45.4_{\pm2.7}$ | $42.8_{\pm1.8}$ | $52.0_{\pm2.5}$ | $42.9_{\pm2.3}$ | $44.3_{\pm2.4}$ | $41.8_{\pm1.9}$ | $40.3_{\pm2.2}$ |
| | FiQA-2018 | $61.1_{\pm1.5}$ | $71.6_{\pm3.3}$ | $62.4_{\pm2.4}$ | $67.5_{\pm3.5}$ | $67.6_{\pm1.7}$ | $64.1_{\pm2.6}$ | $59.8_{\pm1.4}$ |
| | ArguAna | $72.9_{\pm2.4}$ | $80.0_{\pm3.5}$ | $77.5_{\pm2.3}$ | $70.7_{\pm3.3}$ | $74.7_{\pm2.0}$ | $76.9_{\pm3.1}$ | $74.7_{\pm3.3}$ |
| | **Avg.** | 59.8 | 64.8 | 64.0 | 60.3 | 62.2 | 60.9 | 58.3 |
| **H−MoE (Ours)** | SciDocs | $58.8_{\pm2.7}$ | $61.9_{\pm3.2}$ | $63.5_{\pm2.3}$ | $55.7_{\pm2.2}$ | $58.2_{\pm3.9}$ | $56.6_{\pm0.9}$ | $56.4_{\pm1.4}$ |
| | FiQA-2018 | $78.8_{\pm0.6}$ | $80.7_{\pm3.4}$ | $81.9_{\pm2.7}$ | $77.5_{\pm2.4}$ | $78.7_{\pm3.3}$ | $77.6_{\pm1.0}$ | $78.2_{\pm1.8}$ |
| | ArguAna | $90.4_{\pm3.8}$ | $89.7_{\pm3.0}$ | $91.1_{\pm2.7}$ | $88.1_{\pm2.4}$ | $90.0_{\pm3.3}$ | $88.5_{\pm2.3}$ | $88.9_{\pm2.5}$ |
| | **Avg.** | **76.0** | **77.4** | **78.8** | **73.8** | **75.6** | **74.2** | **74.5** |

*Table 17.* **Pairwise Model Translation Performance (Recall@100) - Group 7. Bold and underline denote the best and runner-up average results. Values highlighted in red indicate high-variance outcomes** (std $> 10$).

| Method | Dataset | e5→qwen | e5→kalm | e5→linq | e5→gritlm | e5→openai | e5→mistral | sfr→gemini |
|---|---|---|---|---|---|---|---|---|
| GW | SciDocs | $1.2\pm_{0.3}$ | $0.4\pm_{0.0}$ | $2.7\pm_{0.0}$ | $4.5\pm_{0.3}$ | $2.1\pm_{0.0}$ | $0.8\pm_{0.5}$ | $3.9\pm_{0.5}$ |
| | FiQA-2018 | $3.3\pm_{0.4}$ | $2.5\pm_{0.2}$ | $4.4\pm_{0.4}$ | $4.7\pm_{0.2}$ | $1.3\pm_{0.4}$ | $2.6\pm_{0.5}$ | $3.9\pm_{0.0}$ |
| | ArguAna | $0.6\pm_{0.2}$ | $2.9\pm_{0.4}$ | $0.3\pm_{0.0}$ | $3.0\pm_{0.4}$ | $4.5\pm_{0.2}$ | $2.2\pm_{0.4}$ | $5.0\pm_{0.2}$ |
| | **Avg.** | 1.7 | 1.9 | 2.5 | 4.0 | 2.6 | 1.9 | 4.2 |
| PA | SciDocs | $15.9\pm_{3.7}$ | $3.4\pm_{1.4}$ | $5.3\pm_{0.8}$ | $10.4\pm_{2.0}$ | $10.0\pm_{1.3}$ | $8.7\pm_{0.8}$ | $16.0\pm_{3.0}$ |
| | FiQA-2018 | $11.4\pm_{0.9}$ | $12.7\pm_{2.4}$ | $8.1\pm_{2.5}$ | $5.6\pm_{1.4}$ | $3.2\pm_{0.8}$ | $4.9\pm_{1.7}$ | $13.7\pm_{2.6}$ |
| | ArguAna | $13.6\pm_{5.0}$ | $1.1\pm_{0.5}$ | $1.4\pm_{0.6}$ | $15.2\pm_{6.9}$ | $2.0\pm_{0.7}$ | $4.1\pm_{0.7}$ | $13.9\pm_{6.0}$ |
| | **Avg.** | 13.6 | 5.7 | 4.9 | 10.4 | 5.1 | 5.9 | 14.5 |
| CCA | SciDocs | $36.0\pm_{5.6}$ | $36.6\pm_{6.9}$ | $44.8\pm_{7.4}$ | $35.0\pm_{8.0}$ | $36.0\pm_{9.0}$ | $37.5\pm_{8.5}$ | $35.9\pm_{8.9}$ |
| | FiQA-2018 | $58.3\pm_{5.6}$ | $57.3\pm_{8.1}$ | $58.7\pm_{6.3}$ | $57.0\pm_{7.2}$ | $56.4\pm_{6.5}$ | $56.5\pm_{4.4}$ | $56.8\pm_{7.8}$ |
| | ArguAna | $65.2\pm_{5.5}$ | $68.9\pm_{6.8}$ | $69.6\pm_{5.5}$ | $65.9\pm_{7.6}$ | $66.5\pm_{6.7}$ | $66.9\pm_{5.8}$ | $67.9\pm_{4.1}$ |
| | **Avg.** | 53.2 | 54.3 | 57.7 | 52.6 | 53.0 | 53.6 | 53.5 |
| Vec2Vec | SciDocs | $30.2\pm_{9.6}$ | $26.9\pm_{6.2}$ | $29.1\pm_{7.4}$ | $22.3\pm_{11.9}$ | $31.2\pm_{12.7}$ | $23.9\pm_{10.1}$ | $37.4\pm_{7.3}$ |
| | FiQA-2018 | $31.5\pm_{9.1}$ | $28.3\pm_{12.7}$ | $29.5\pm_{9.1}$ | $33.5\pm_{7.2}$ | $28.1\pm_{7.3}$ | $34.1\pm_{7.4}$ | $29.2\pm_{7.8}$ |
| | ArguAna | $35.4\pm_{10.1}$ | $38.8\pm_{7.0}$ | $26.4\pm_{8.1}$ | $24.9\pm_{7.2}$ | $28.6\pm_{6.2}$ | $36.2\pm_{6.4}$ | $30.6\pm_{10.5}$ |
| | **Avg.** | 32.4 | 31.3 | 28.3 | 26.9 | 29.3 | 31.4 | 32.4 |
| EmbConv | SciDocs | $39.9\pm_{2.5}$ | $48.0\pm_{1.3}$ | $52.8\pm_{1.5}$ | $44.8\pm_{2.1}$ | $40.8\pm_{2.7}$ | $45.5\pm_{2.3}$ | $37.3\pm_{3.1}$ |
| | FiQA-2018 | $59.7\pm_{2.5}$ | $65.8\pm_{1.5}$ | $66.7\pm_{2.6}$ | $65.2\pm_{1.0}$ | $59.9\pm_{2.2}$ | $64.4\pm_{2.9}$ | $60.0\pm_{1.1}$ |
| | ArguAna | $72.7\pm_{1.7}$ | $74.9\pm_{2.0}$ | $76.8\pm_{2.2}$ | $70.8\pm_{3.4}$ | $78.0\pm_{2.9}$ | $74.7\pm_{1.3}$ | $74.1\pm_{3.5}$ |
| | **Avg.** | 57.4 | 62.9 | 65.4 | 60.3 | 59.6 | 61.5 | 57.1 |
| **H−MoE (Ours)** | SciDocs | $55.0\pm_{3.4}$ | $58.4\pm_{3.4}$ | $68.0\pm_{2.6}$ | $57.9\pm_{1.2}$ | $55.9\pm_{1.6}$ | $59.6\pm_{2.9}$ | $56.8\pm_{3.5}$ |
| | FiQA-2018 | $78.9\pm_{1.5}$ | $79.5\pm_{4.0}$ | $83.0\pm_{3.0}$ | $79.1\pm_{0.7}$ | $78.9\pm_{1.5}$ | $78.9\pm_{3.9}$ | $77.7\pm_{3.5}$ |
| | ArguAna | $90.1\pm_{2.1}$ | $90.2\pm_{1.3}$ | $91.6\pm_{1.1}$ | $89.4\pm_{1.5}$ | $88.6\pm_{3.5}$ | $90.2\pm_{3.7}$ | $88.6\pm_{3.0}$ |
| | **Avg.** | **74.7** | **76.0** | **80.9** | **75.5** | **74.5** | **76.2** | **74.4** |

*Table 18.* **Pairwise Model Translation Performance (Recall@100) - Group 8. Bold and underline denote the best and runner-up average results. Values highlighted in red indicate high-variance outcomes** (std $> 10$).

| Method | Dataset | sfr→nemotron | sfr→qwen | sfr→kalm | sfr→linq | sfr→e5 | sfr→gritlm | sfr→openai |
|---|---|---|---|---|---|---|---|---|
| GW | SciDocs | $3.8\pm_{0.3}$ | $0.6\pm_{0.3}$ | $3.7\pm_{0.4}$ | $1.1\pm_{0.1}$ | $0.3\pm_{0.0}$ | $1.1\pm_{0.3}$ | $1.9\pm_{0.5}$ |
| | FiQA-2018 | $2.9\pm_{0.1}$ | $4.7\pm_{0.2}$ | $3.4\pm_{0.2}$ | $1.8\pm_{0.3}$ | $2.4\pm_{0.1}$ | $3.3\pm_{0.1}$ | $2.0\pm_{0.2}$ |
| | ArguAna | $2.3\pm_{0.3}$ | $1.7\pm_{0.0}$ | $2.7\pm_{0.1}$ | $2.5\pm_{0.2}$ | $2.3\pm_{0.0}$ | $1.4\pm_{0.3}$ | $1.2\pm_{0.3}$ |
| | **Avg.** | 3.0 | 2.3 | 3.3 | 1.8 | 1.7 | 1.9 | 1.7 |
| PA | SciDocs | $1.0\pm_{0.5}$ | $7.7\pm_{0.9}$ | $13.2\pm_{5.1}$ | $12.4\pm_{4.1}$ | $10.3\pm_{2.3}$ | $10.6\pm_{4.2}$ | $5.7\pm_{0.6}$ |
| | FiQA-2018 | $4.6\pm_{1.7}$ | $7.1\pm_{2.7}$ | $16.7\pm_{1.6}$ | $7.5\pm_{2.7}$ | $6.0\pm_{0.8}$ | $14.4\pm_{4.9}$ | $16.1\pm_{4.2}$ |
| | ArguAna | $12.0\pm_{5.0}$ | $6.0\pm_{2.7}$ | $1.1\pm_{0.5}$ | $2.4\pm_{0.6}$ | $2.3\pm_{0.9}$ | $9.2\pm_{2.6}$ | $8.9\pm_{2.8}$ |
| | **Avg.** | 5.9 | 6.9 | 10.3 | 7.4 | 6.2 | 11.4 | 10.2 |
| CCA | SciDocs | $33.8\pm_{8.9}$ | $34.4\pm_{7.9}$ | $38.4\pm_{6.5}$ | $41.5\pm_{5.0}$ | $45.9\pm_{7.6}$ | $37.5\pm_{7.7}$ | $35.6\pm_{4.8}$ |
| | FiQA-2018 | $59.9\pm_{8.3}$ | $58.4\pm_{5.3}$ | $57.2\pm_{4.4}$ | $60.0\pm_{6.0}$ | $65.0\pm_{4.4}$ | $56.6\pm_{7.5}$ | $55.0\pm_{5.3}$ |
| | ArguAna | $68.0\pm_{7.4}$ | $68.4\pm_{7.9}$ | $67.2\pm_{5.4}$ | $66.9\pm_{7.4}$ | $69.5\pm_{7.0}$ | $64.8\pm_{6.6}$ | $66.6\pm_{4.1}$ |
| | **Avg.** | 53.9 | 53.7 | 54.3 | 56.1 | 60.1 | 52.9 | 52.4 |
| Vec2Vec | SciDocs | $25.4\pm_{6.4}$ | $27.4\pm_{7.8}$ | $24.9\pm_{9.3}$ | $28.4\pm_{8.3}$ | $23.6\pm_{11.6}$ | $31.1\pm_{6.1}$ | $35.5\pm_{7.4}$ |
| | FiQA-2018 | $31.5\pm_{6.5}$ | $31.5\pm_{11.1}$ | $23.0\pm_{12.5}$ | $34.4\pm_{7.9}$ | $39.1\pm_{6.5}$ | $31.3\pm_{8.1}$ | $35.2\pm_{7.7}$ |
| | ArguAna | $27.2\pm_{9.5}$ | $30.2\pm_{6.3}$ | $31.1\pm_{12.4}$ | $27.5\pm_{9.5}$ | $25.6\pm_{11.9}$ | $29.3\pm_{10.1}$ | $28.2\pm_{7.6}$ |
| | **Avg.** | 28.1 | 29.7 | 26.3 | 30.1 | 29.4 | 30.5 | 33.0 |
| EmbConv | SciDocs | $40.0\pm_{3.2}$ | $46.3\pm_{2.2}$ | $44.6\pm_{1.9}$ | $50.2\pm_{3.4}$ | $57.1\pm_{1.2}$ | $46.0\pm_{3.4}$ | $42.4\pm_{1.1}$ |
| | FiQA-2018 | $70.6\pm_{2.6}$ | $58.8\pm_{2.8}$ | $71.0\pm_{1.2}$ | $69.0\pm_{1.8}$ | $66.2\pm_{2.5}$ | $67.2\pm_{1.7}$ | $67.1\pm_{1.2}$ |
| | ArguAna | $73.1\pm_{1.3}$ | $69.6\pm_{3.2}$ | $71.3\pm_{1.2}$ | $74.4\pm_{2.8}$ | $79.6\pm_{2.2}$ | $77.9\pm_{2.7}$ | $75.6\pm_{2.3}$ |
| | **Avg.** | 61.2 | 58.3 | 62.3 | 64.5 | 67.6 | 63.7 | 61.7 |
| **H−MoE (Ours)** | SciDocs | $59.0\pm_{1.4}$ | $59.2\pm_{1.4}$ | $62.2\pm_{3.9}$ | $64.7\pm_{2.5}$ | $71.0\pm_{2.1}$ | $60.6\pm_{2.1}$ | $57.7\pm_{2.9}$ |
| | FiQA-2018 | $80.6\pm_{0.6}$ | $79.6\pm_{3.3}$ | $80.2\pm_{3.4}$ | $81.6\pm_{3.3}$ | $83.8\pm_{3.7}$ | $80.2\pm_{2.5}$ | $79.0\pm_{1.3}$ |
| | ArguAna | $89.5\pm_{2.1}$ | $88.9\pm_{0.5}$ | $90.0\pm_{2.7}$ | $89.8\pm_{0.7}$ | $92.5\pm_{1.7}$ | $90.9\pm_{1.9}$ | $89.0\pm_{1.5}$ |
| | **Avg.** | **76.4** | **75.9** | **77.4** | **78.7** | **82.5** | **77.2** | **75.3** |

*Table 19.* **Pairwise Model Translation Performance (Recall@100) - Group 9. Bold and underline denote the best and runner-up average results. Values highlighted in red indicate high-variance outcomes (std > 10).**

| Method | Dataset | sfr →mistral | gritlm →gemini | gritlm →nemotron | gritlm →qwen | gritlm →kalm | gritlm →linq | gritlm →e5 |
|---|---|---|---|---|---|---|---|---|
| GW | SciDocs | $0.3\pm_{0.3}$ | $1.1\pm_{0.4}$ | $2.8\pm_{0.2}$ | $3.9\pm_{0.0}$ | $1.4\pm_{0.4}$ | $4.2\pm_{0.4}$ | $4.7\pm_{0.3}$ |
| | FiQA-2018 | $1.5\pm_{0.2}$ | $3.2\pm_{0.4}$ | $2.6\pm_{0.2}$ | $2.7\pm_{0.4}$ | $0.4\pm_{0.5}$ | $3.0\pm_{0.4}$ | $1.0\pm_{0.5}$ |
| | ArguAna | $0.4\pm_{0.1}$ | $4.9\pm_{0.4}$ | $1.7\pm_{0.1}$ | $4.9\pm_{0.1}$ | $4.2\pm_{0.4}$ | $4.6\pm_{0.0}$ | $2.5\pm_{0.2}$ |
| | **Avg.** | 0.7 | 3.1 | 2.3 | 3.8 | 2.0 | 3.9 | 2.8 |
| PA | SciDocs | $2.4\pm_{0.7}$ | $15.5\pm_{6.6}$ | $16.2\pm_{3.3}$ | $13.5\pm_{6.5}$ | $10.3\pm_{1.3}$ | $16.7\pm_{7.7}$ | $3.2\pm_{1.3}$ |
| | FiQA-2018 | $9.3\pm_{0.5}$ | $1.3\pm_{0.6}$ | $7.3\pm_{0.8}$ | $3.8\pm_{1.8}$ | $11.9\pm_{1.2}$ | $10.5\pm_{4.9}$ | $16.5\pm_{1.8}$ |
| | ArguAna | $4.2\pm_{0.7}$ | $13.2\pm_{5.7}$ | $3.2\pm_{0.7}$ | $15.7\pm_{6.3}$ | $16.0\pm_{7.6}$ | $4.5\pm_{1.5}$ | $2.4\pm_{1.0}$ |
| | **Avg.** | 5.3 | 10.0 | 8.9 | 11.0 | 12.7 | 10.6 | 7.4 |
| CCA | SciDocs | $37.4\pm_{6.2}$ | $34.8\pm_{8.8}$ | $44.8\pm_{7.4}$ | $45.6\pm_{5.9}$ | $44.3\pm_{8.7}$ | $35.5\pm_{7.6}$ | $36.2\pm_{6.1}$ |
| | FiQA-2018 | $59.2\pm_{4.3}$ | $58.0\pm_{5.9}$ | $62.3\pm_{8.2}$ | $61.1\pm_{8.5}$ | $57.0\pm_{4.8}$ | $57.8\pm_{5.2}$ | $55.9\pm_{7.0}$ |
| | ArguAna | $69.2\pm_{4.0}$ | $69.3\pm_{6.4}$ | $69.3\pm_{8.8}$ | $67.5\pm_{7.1}$ | $69.0\pm_{4.9}$ | $65.9\pm_{8.2}$ | $68.9\pm_{4.9}$ |
| | **Avg.** | 55.3 | 54.0 | 58.8 | 58.1 | 56.7 | 53.0 | 53.7 |
| Vec2Vec | SciDocs | $21.6\pm_{11.9}$ | $37.7\pm_{8.6}$ | $33.3\pm_{6.6}$ | $28.4\pm_{9.7}$ | $30.0\pm_{12.3}$ | $39.4\pm_{11.7}$ | $25.2\pm_{12.4}$ |
| | FiQA-2018 | $27.9\pm_{9.4}$ | $25.3\pm_{9.3}$ | $30.8\pm_{12.8}$ | $29.5\pm_{12.2}$ | $33.6\pm_{10.3}$ | $28.5\pm_{12.9}$ | $21.3\pm_{8.7}$ |
| | ArguAna | $21.8\pm_{9.8}$ | $25.4\pm_{6.4}$ | $31.1\pm_{11.9}$ | $21.6\pm_{12.3}$ | $35.1\pm_{7.2}$ | $39.9\pm_{10.4}$ | $26.5\pm_{7.9}$ |
| | **Avg.** | 23.8 | 29.4 | 31.8 | 26.5 | 32.9 | 36.0 | 24.3 |
| EmbConv | SciDocs | $42.7\pm_{3.0}$ | $39.8\pm_{3.3}$ | $52.2\pm_{2.5}$ | $54.8\pm_{1.6}$ | $49.1\pm_{2.7}$ | $44.0\pm_{2.6}$ | $44.2\pm_{1.4}$ |
| | FiQA-2018 | $65.9\pm_{2.0}$ | $68.8\pm_{1.7}$ | $69.7\pm_{1.4}$ | $64.6\pm_{3.2}$ | $62.6\pm_{1.3}$ | $66.9\pm_{3.1}$ | $64.2\pm_{1.3}$ |
| | ArguAna | $78.6\pm_{2.5}$ | $76.3\pm_{2.0}$ | $74.6\pm_{1.1}$ | $72.2\pm_{1.8}$ | $77.3\pm_{3.1}$ | $72.6\pm_{3.2}$ | $71.6\pm_{1.1}$ |
| | **Avg.** | 62.4 | 61.6 | 65.5 | 63.9 | 63.0 | 61.2 | 60.0 |
| **H−MoE (Ours)** | SciDocs | $60.9\pm_{3.4}$ | $59.1\pm_{1.5}$ | $66.4\pm_{1.8}$ | $67.0\pm_{1.1}$ | $64.2\pm_{1.1}$ | $59.0\pm_{1.4}$ | $57.7\pm_{3.7}$ |
| | FiQA-2018 | $81.6\pm_{1.4}$ | $80.3\pm_{2.3}$ | $83.5\pm_{1.1}$ | $84.1\pm_{3.1}$ | $82.3\pm_{3.9}$ | $79.5\pm_{0.6}$ | $80.0\pm_{2.6}$ |
| | ArguAna | $90.1\pm_{3.7}$ | $89.9\pm_{0.9}$ | $92.1\pm_{2.6}$ | $91.7\pm_{1.5}$ | $91.5\pm_{2.8}$ | $90.2\pm_{3.2}$ | $88.7\pm_{1.4}$ |
| | **Avg.** | **77.5** | **76.4** | **80.7** | **81.0** | **79.3** | **76.2** | **75.4** |

*Table 20.* **Pairwise Model Translation Performance (Recall@100) - Group 10. Bold and underline denote the best and runner-up average results. Values highlighted in red indicate high-variance outcomes (std > 10).**

| Method | Dataset | gritlm →sfr | gritlm →openai | gritlm →mistral | openai →gemini | openai →nemotron | openai →qwen | openai →kalm |
|---|---|---|---|---|---|---|---|---|
| GW | SciDocs | $1.6\pm_{0.5}$ | $4.1\pm_{0.2}$ | $1.6\pm_{0.1}$ | $1.6\pm_{0.1}$ | $3.6\pm_{0.3}$ | $1.0\pm_{0.4}$ | $4.2\pm_{0.4}$ |
| | FiQA-2018 | $2.2\pm_{0.2}$ | $2.6\pm_{0.3}$ | $4.9\pm_{0.3}$ | $4.5\pm_{0.4}$ | $1.4\pm_{0.2}$ | $2.7\pm_{0.4}$ | $0.6\pm_{0.1}$ |
| | ArguAna | $1.3\pm_{0.2}$ | $1.9\pm_{0.3}$ | $1.4\pm_{0.4}$ | $3.0\pm_{0.3}$ | $0.3\pm_{0.4}$ | $3.4\pm_{0.3}$ | $4.1\pm_{0.4}$ |
| | **Avg.** | 1.7 | 2.9 | 2.6 | 3.0 | 1.8 | 2.4 | 3.0 |
| PA | SciDocs | $11.7\pm_{2.3}$ | $5.6\pm_{2.5}$ | $14.7\pm_{3.1}$ | $15.3\pm_{3.8}$ | $1.1\pm_{0.5}$ | $11.0\pm_{2.5}$ | $14.2\pm_{1.8}$ |
| | FiQA-2018 | $6.6\pm_{2.6}$ | $14.7\pm_{4.5}$ | $1.9\pm_{0.9}$ | $12.3\pm_{4.9}$ | $3.7\pm_{1.5}$ | $7.1\pm_{0.6}$ | $10.9\pm_{1.7}$ |
| | ArguAna | $13.5\pm_{2.6}$ | $2.1\pm_{1.0}$ | $4.2\pm_{1.1}$ | $11.0\pm_{1.4}$ | $13.1\pm_{1.0}$ | $6.9\pm_{0.5}$ | $6.4\pm_{0.9}$ |
| | **Avg.** | 10.6 | 7.5 | 6.9 | 12.9 | 5.9 | 8.3 | 10.5 |
| CCA | SciDocs | $38.4\pm_{4.8}$ | $37.4\pm_{5.7}$ | $48.9\pm_{4.7}$ | $43.9\pm_{6.7}$ | $39.5\pm_{6.7}$ | $40.5\pm_{7.0}$ | $39.8\pm_{5.0}$ |
| | FiQA-2018 | $60.0\pm_{5.5}$ | $59.9\pm_{4.9}$ | $63.3\pm_{4.8}$ | $56.2\pm_{7.3}$ | $60.1\pm_{7.5}$ | $56.0\pm_{4.5}$ | $57.8\pm_{5.0}$ |
| | ArguAna | $67.7\pm_{5.7}$ | $66.5\pm_{6.0}$ | $68.9\pm_{4.2}$ | $71.0\pm_{7.4}$ | $69.0\pm_{7.3}$ | $70.1\pm_{4.6}$ | $69.2\pm_{4.2}$ |
| | **Avg.** | 55.4 | 54.6 | 60.4 | 57.0 | 56.2 | 55.5 | 55.6 |
| Vec2Vec | SciDocs | $24.2\pm_{12.5}$ | $21.8\pm_{9.6}$ | $22.0\pm_{7.2}$ | $20.9\pm_{12.0}$ | $36.3\pm_{7.3}$ | $35.9\pm_{6.8}$ | $33.6\pm_{7.3}$ |
| | FiQA-2018 | $20.1\pm_{10.7}$ | $22.0\pm_{7.8}$ | $25.8\pm_{11.6}$ | $23.4\pm_{8.5}$ | $39.2\pm_{7.9}$ | $21.2\pm_{12.6}$ | $27.2\pm_{6.2}$ |
| | ArguAna | $36.7\pm_{10.7}$ | $39.8\pm_{8.1}$ | $34.8\pm_{6.7}$ | $32.4\pm_{12.1}$ | $37.0\pm_{11.6}$ | $28.3\pm_{10.1}$ | $36.8\pm_{12.8}$ |
| | **Avg.** | 27.0 | 27.9 | 27.5 | 25.6 | 37.5 | 28.5 | 32.5 |
| EmbConv | SciDocs | $41.6\pm_{1.3}$ | $46.3\pm_{2.4}$ | $60.9\pm_{1.9}$ | $53.2\pm_{3.1}$ | $48.4\pm_{2.6}$ | $50.3\pm_{2.2}$ | $45.6\pm_{1.8}$ |
| | FiQA-2018 | $70.3\pm_{1.7}$ | $65.1\pm_{1.8}$ | $72.3\pm_{3.1}$ | $64.1\pm_{1.2}$ | $68.3\pm_{2.1}$ | $70.2\pm_{2.0}$ | $68.8\pm_{2.7}$ |
| | ArguAna | $72.8\pm_{2.7}$ | $73.3\pm_{1.2}$ | $72.7\pm_{1.1}$ | $78.4\pm_{3.4}$ | $76.3\pm_{3.1}$ | $80.3\pm_{1.2}$ | $77.5\pm_{1.1}$ |
| | **Avg.** | 61.6 | 61.6 | 68.6 | 65.2 | 64.4 | 66.9 | 64.0 |
| **H−MoE (Ours)** | SciDocs | $61.0\pm_{2.1}$ | $62.0\pm_{0.9}$ | $74.0\pm_{1.5}$ | $63.7\pm_{3.0}$ | $62.4\pm_{1.9}$ | $61.4\pm_{3.7}$ | $61.9\pm_{3.8}$ |
| | FiQA-2018 | $81.0\pm_{2.7}$ | $81.4\pm_{1.3}$ | $85.8\pm_{3.0}$ | $82.6\pm_{0.9}$ | $80.8\pm_{3.2}$ | $81.6\pm_{2.6}$ | $80.2\pm_{2.2}$ |
| | ArguAna | $90.2\pm_{2.8}$ | $89.9\pm_{3.4}$ | $91.5\pm_{1.1}$ | $91.0\pm_{1.2}$ | $90.7\pm_{0.9}$ | $90.8\pm_{2.1}$ | $90.3\pm_{0.7}$ |
| | **Avg.** | **77.4** | **77.8** | **83.8** | **79.1** | **78.0** | **77.9** | **77.4** |

*Table 21.* **Pairwise Model Translation Performance (Recall@100) - Group 11. Bold and underline denote the best and runner-up average results. Values highlighted in red indicate high-variance outcomes** (std > 10).

| Method | Dataset | openai→linq | openai→e5 | openai→sfr | openai→gritlm | openai→mistral | mistral→gemini | mistral→nemotron |
|---|---|---|---|---|---|---|---|---|
| GW | SciDocs | $4.2_{\pm0.2}$ | $3.0_{\pm0.4}$ | $4.9_{\pm0.2}$ | $3.6_{\pm0.5}$ | $3.3_{\pm0.5}$ | $3.4_{\pm0.2}$ | $0.8_{\pm0.3}$ |
| | FiQA-2018 | $4.6_{\pm0.1}$ | $2.0_{\pm0.5}$ | $2.1_{\pm0.4}$ | $1.1_{\pm0.3}$ | $3.5_{\pm0.2}$ | $4.2_{\pm0.1}$ | $3.8_{\pm0.1}$ |
| | ArguAna | $3.7_{\pm0.1}$ | $3.8_{\pm0.1}$ | $1.6_{\pm0.4}$ | $1.8_{\pm0.3}$ | $1.8_{\pm0.2}$ | $4.1_{\pm0.3}$ | $1.3_{\pm0.4}$ |
| | **Avg.** | 4.2 | 3.0 | 2.8 | 2.2 | 2.9 | 3.9 | 2.0 |
| PA | SciDocs | $5.7_{\pm2.3}$ | $9.9_{\pm1.6}$ | $16.4_{\pm2.8}$ | $15.4_{\pm5.6}$ | $3.8_{\pm1.5}$ | $10.2_{\pm1.7}$ | $10.7_{\pm3.7}$ |
| | FiQA-2018 | $13.9_{\pm1.2}$ | $13.6_{\pm2.3}$ | $1.2_{\pm0.5}$ | $12.9_{\pm3.8}$ | $3.7_{\pm1.2}$ | $12.2_{\pm1.4}$ | $10.0_{\pm2.1}$ |
| | ArguAna | $12.7_{\pm5.6}$ | $15.6_{\pm3.3}$ | $13.3_{\pm5.9}$ | $13.3_{\pm6.4}$ | $7.8_{\pm2.4}$ | $6.1_{\pm0.6}$ | $6.0_{\pm1.1}$ |
| | **Avg.** | 10.7 | 13.0 | 10.3 | 13.9 | 5.1 | 9.5 | 8.9 |
| CCA | SciDocs | $34.8_{\pm6.5}$ | $34.4_{\pm7.8}$ | $34.0_{\pm7.1}$ | $38.6_{\pm5.4}$ | $41.4_{\pm6.6}$ | $35.1_{\pm8.3}$ | $45.8_{\pm7.7}$ |
| | FiQA-2018 | $55.6_{\pm4.5}$ | $53.5_{\pm8.1}$ | $55.1_{\pm7.4}$ | $57.6_{\pm5.9}$ | $60.9_{\pm8.8}$ | $59.0_{\pm6.7}$ | $61.6_{\pm5.7}$ |
| | ArguAna | $67.5_{\pm5.7}$ | $70.0_{\pm7.3}$ | $64.7_{\pm8.7}$ | $68.6_{\pm4.2}$ | $70.3_{\pm4.3}$ | $65.2_{\pm7.9}$ | $70.8_{\pm8.0}$ |
| | **Avg.** | 52.6 | 52.7 | 51.3 | 54.9 | 57.5 | 53.1 | 59.4 |
| Vec2Vec | SciDocs | $31.5_{\pm7.6}$ | $32.8_{\pm10.3}$ | $26.0_{\pm6.6}$ | $27.8_{\pm11.6}$ | $27.5_{\pm12.6}$ | $26.1_{\pm11.2}$ | $36.7_{\pm7.4}$ |
| | FiQA-2018 | $27.8_{\pm10.4}$ | $31.7_{\pm10.9}$ | $28.8_{\pm8.8}$ | $39.2_{\pm7.0}$ | $31.1_{\pm6.9}$ | $26.9_{\pm8.9}$ | $36.0_{\pm12.0}$ |
| | ArguAna | $20.3_{\pm10.7}$ | $38.6_{\pm6.7}$ | $29.5_{\pm9.7}$ | $38.3_{\pm11.6}$ | $31.9_{\pm9.8}$ | $30.3_{\pm7.1}$ | $38.6_{\pm13.0}$ |
| | **Avg.** | 26.5 | 34.4 | 28.1 | 35.1 | 30.2 | 27.7 | 37.1 |
| EmbConv | SciDocs | $40.2_{\pm1.7}$ | $45.1_{\pm2.5}$ | $45.2_{\pm1.8}$ | $50.9_{\pm2.8}$ | $43.4_{\pm3.1}$ | $42.6_{\pm1.1}$ | $56.5_{\pm2.9}$ |
| | FiQA-2018 | $66.6_{\pm2.8}$ | $66.1_{\pm2.1}$ | $69.5_{\pm1.6}$ | $67.0_{\pm1.9}$ | $64.8_{\pm1.7}$ | $65.8_{\pm2.5}$ | $72.1_{\pm2.9}$ |
| | ArguAna | $75.8_{\pm1.3}$ | $76.3_{\pm2.5}$ | $71.7_{\pm2.1}$ | $74.3_{\pm1.4}$ | $80.2_{\pm1.2}$ | $75.0_{\pm2.4}$ | $79.4_{\pm2.2}$ |
| | **Avg.** | 60.9 | 62.5 | 62.1 | 64.1 | 62.8 | 61.1 | 69.3 |
| **H-MoE (Ours)** | SciDocs | $55.8_{\pm1.9}$ | $56.0_{\pm1.1}$ | $59.0_{\pm3.9}$ | $61.5_{\pm0.7}$ | $61.3_{\pm2.1}$ | $59.1_{\pm2.3}$ | $69.6_{\pm2.8}$ |
| | FiQA-2018 | $78.4_{\pm2.8}$ | $78.5_{\pm2.1}$ | $80.1_{\pm2.2}$ | $80.6_{\pm0.6}$ | $80.8_{\pm0.7}$ | $80.3_{\pm3.4}$ | $84.3_{\pm2.8}$ |
| | ArguAna | $89.8_{\pm3.3}$ | $88.5_{\pm3.8}$ | $89.4_{\pm1.3}$ | $89.6_{\pm3.7}$ | $91.2_{\pm1.6}$ | $89.2_{\pm2.4}$ | $92.6_{\pm2.3}$ |
| | **Avg.** | **74.7** | **74.3** | **76.2** | **77.3** | **77.8** | **76.2** | **82.1** |

*Table 22.* **Pairwise Model Translation Performance (Recall@100) - Group 12. Bold and underline denote the best and runner-up average results. Values highlighted in red indicate high-variance outcomes** (std > 10).

| Method | Dataset | mistral→qwen | mistral→kalm | mistral→linq | mistral→e5 | mistral→sfr | mistral→gritlm |
|---|---|---|---|---|---|---|---|
| GW | SciDocs | $4.5_{\pm0.3}$ | $0.9_{\pm0.0}$ | $4.1_{\pm0.4}$ | $2.9_{\pm0.1}$ | $3.0_{\pm0.3}$ | $4.3_{\pm0.1}$ |
| | FiQA-2018 | $4.4_{\pm0.2}$ | $1.1_{\pm0.1}$ | $2.4_{\pm0.1}$ | $0.3_{\pm0.3}$ | $3.6_{\pm0.3}$ | $3.0_{\pm0.4}$ |
| | ArguAna | $3.1_{\pm0.3}$ | $3.6_{\pm0.1}$ | $4.2_{\pm0.0}$ | $1.4_{\pm0.1}$ | $0.5_{\pm0.2}$ | $1.0_{\pm0.3}$ |
| | **Avg.** | 4.0 | 1.9 | 3.6 | 1.5 | 2.4 | 2.8 |
| PA | SciDocs | $12.4_{\pm2.5}$ | $9.0_{\pm2.2}$ | $2.9_{\pm0.9}$ | $6.9_{\pm0.6}$ | $13.0_{\pm4.7}$ | $7.5_{\pm1.1}$ |
| | FiQA-2018 | $10.3_{\pm1.6}$ | $2.5_{\pm0.9}$ | $10.5_{\pm4.4}$ | $4.6_{\pm2.0}$ | $3.1_{\pm1.0}$ | $13.5_{\pm4.1}$ |
| | ArguAna | $2.7_{\pm0.7}$ | $1.6_{\pm0.7}$ | $13.2_{\pm4.5}$ | $7.0_{\pm2.3}$ | $14.6_{\pm2.4}$ | $1.1_{\pm0.5}$ |
| | **Avg.** | 8.5 | 4.4 | 8.9 | 6.2 | 10.2 | 7.3 |
| CCA | SciDocs | $46.1_{\pm5.8}$ | $40.6_{\pm7.8}$ | $36.6_{\pm7.5}$ | $36.9_{\pm6.0}$ | $40.1_{\pm4.7}$ | $48.0_{\pm7.1}$ |
| | FiQA-2018 | $59.4_{\pm5.3}$ | $58.0_{\pm6.4}$ | $55.9_{\pm7.3}$ | $56.6_{\pm7.5}$ | $59.6_{\pm4.3}$ | $62.3_{\pm7.0}$ |
| | ArguAna | $69.9_{\pm5.8}$ | $67.7_{\pm5.3}$ | $67.9_{\pm7.5}$ | $69.3_{\pm8.8}$ | $69.3_{\pm4.2}$ | $68.9_{\pm7.2}$ |
| | **Avg.** | 58.5 | 55.5 | 53.5 | 54.3 | 56.3 | 59.7 |
| Vec2Vec | SciDocs | $24.0_{\pm10.6}$ | $20.7_{\pm7.7}$ | $30.4_{\pm11.2}$ | $25.5_{\pm10.2}$ | $25.9_{\pm12.6}$ | $31.3_{\pm11.4}$ |
| | FiQA-2018 | $25.7_{\pm6.2}$ | $23.1_{\pm13.0}$ | $21.8_{\pm8.0}$ | $38.6_{\pm10.0}$ | $39.1_{\pm12.1}$ | $39.2_{\pm9.2}$ |
| | ArguAna | $30.3_{\pm7.6}$ | $27.1_{\pm8.8}$ | $38.3_{\pm8.4}$ | $28.1_{\pm6.7}$ | $31.5_{\pm10.9}$ | $31.7_{\pm6.2}$ |
| | **Avg.** | 26.7 | 23.6 | 30.2 | 30.8 | 32.2 | 34.1 |
| EmbConv | SciDocs | $55.9_{\pm2.6}$ | $45.1_{\pm1.6}$ | $39.4_{\pm1.7}$ | $47.3_{\pm1.5}$ | $45.7_{\pm2.9}$ | $55.2_{\pm2.2}$ |
| | FiQA-2018 | $67.6_{\pm1.6}$ | $64.6_{\pm3.1}$ | $63.7_{\pm3.3}$ | $63.0_{\pm1.4}$ | $70.5_{\pm2.2}$ | $67.6_{\pm1.7}$ |
| | ArguAna | $71.4_{\pm2.3}$ | $70.6_{\pm2.2}$ | $77.7_{\pm2.8}$ | $69.7_{\pm2.7}$ | $75.5_{\pm3.5}$ | $76.1_{\pm1.1}$ |
| | **Avg.** | 65.0 | 60.1 | 60.3 | 60.0 | 63.9 | 66.3 |
| **H-MoE (Ours)** | SciDocs | $67.9_{\pm2.2}$ | $63.8_{\pm3.9}$ | $58.1_{\pm2.5}$ | $58.1_{\pm2.1}$ | $62.0_{\pm1.8}$ | $73.3_{\pm1.3}$ |
| | FiQA-2018 | $83.4_{\pm2.3}$ | $82.5_{\pm2.1}$ | $79.3_{\pm2.6}$ | $79.9_{\pm1.8}$ | $81.6_{\pm1.5}$ | $85.0_{\pm1.4}$ |
| | ArguAna | $91.1_{\pm2.9}$ | $91.2_{\pm0.5}$ | $89.2_{\pm2.4}$ | $89.7_{\pm1.2}$ | $89.6_{\pm1.0}$ | $91.9_{\pm2.0}$ |
| | **Avg.** | **80.8** | **79.2** | **75.5** | **75.9** | **77.7** | **83.4** |

