# OpenReview forum: "Generalizable and Composable Multi-Model Embedding Translation"
_ICML.cc/2026/Conference — ICML 2026 spotlight_

### Official Review · Reviewer_psjd · 2026-03-12

**Soundness:** 4
**Presentation:** 3
**Significance:** 3
**Originality:** 2
**Overall Recommendation:** 5
**Confidence:** 4

**Summary:**

This paper studies embedding translation under more challenging and realistic settings. It emphasizes out-of-distribution (OOD) generalization, and further extends the problem to multi-model mixing and chaining. The paper formalizes embedding translation under a Lipschitz-based error bound. Based on this analysis, it develops a Translation Confidence (TC) score as a practical per-input estimate of translation reliability. Motivated by the empirical Lipschitz constants analysis, the paper introduces H-MoE, a Hierarchical Mixture-of-Experts translator that uses Hierarchical Routing. For multi-model mixing and chaining setting, the paper proposes directional regularization terms. Experiments span a wide range of embedding models, datasets, and domains, evaluating translation confidence, pairwise translation, and compositional settings.

**Compliance With Llm Reviewing Policy:**

Affirmed.

**Final Justification:**

The rebuttal addressed the concerns.

**Key Questions For Authors:**

1. How exactly does the proposed routing differ from a more standard learned routing mechanism? The paper argues that H-MoE benefits from increased architectural capacity, and it would help to clarify what kind of "capacity" is meant here. Additional details would be helpful to understand.

2. Could the authors provide more intuition for the geometric constraints used in compositional settings?

3. Since the paper mainly emphasizes OOD performance, what happens in the in-domain setting? Does the proposed method preserve or degrade in-domain accuracy? In the ablation studies, is there any trade-off between in-domain fit and OOD robustness?

**Limitations:**

I did not see a discussion of limitations. A discussion of potential negative societal impacts does not appear necessary for this work.

**Strengths And Weaknesses:**

### Strength

- The paper has a clear and well-motivated problem setting. The focus on embedding translation itself is already useful, and the emphasis on OOD settings, as well as mixing and chaining, makes the paper more practically relevant.

- Interesting Lipschitz-based framing of OOD error: Instead of only making the intuitive claim that OOD generalization is difficult, the paper formalizes this through a pointwise error bound, which also motivates the Translation Confidence score. This gives a concrete theoretical basis for a practically useful tool, and the implications of TC are clearly stated.

- The paper also provides an interesting analytical bridge from theory to architecture design. The discussion of empirical Lipschitz constants helps connect the bound to observed behavior of monolithic translators, and motivates using multiple local experts.

- The empirical study is thorough in scope, covering multiple datasets, domains, embedding models, and translation directions, showing the validity of the suggested TC and improvements of H-MoE on each challenging setting.

### Weaknesses

- While H-MoE appears to be an important methodological contribution, the paper’s explanation of the hierarchical routing is not as clear as it could be. The high-level motivation is understandable, but the exact mechanism and how hierarchical routing differs from dominant routing strategies could be explained in more detail. The design and its concrete advantage remain somewhat underexplained in the current presentation.

- Another main originality is residual regularization for multi-model mixing and chaining. The paper does provide the intended effect (L207 and L303). However, the intuition behind why these particular geometric constraints are the right assumption could still be developed more clearly. The implications are stated, but the motivation may feel too brief relative to the importance of this design choice.

- (minor) The terminology and naming are not always fully consistent across the results sections and the rest.

---

> ### Author Rebuttal · Authors · 2026-03-29
>
> Thank you for the careful review. We address the main points below and will incorporate these clarifications in the revision.
>
> **[Q1/W1: routing and capacity]** H-MoE does not use a standard learned gate or top-1 expert. Instead, routing is geometry-based and tied to our source-space analysis: we build a binary clustering tree in the source embedding space, train one expert per node (with each internal-node expert trained on the union of its descendants), and route top-down. We descend only when one child is clearly closer; otherwise, we stop early and use the current coarser expert. This early-stop/backoff rule is the key difference from flat nearest-centroid or learned top-1 routing, which must always commit to a fine expert and can therefore switch abruptly near cluster boundaries. Fig. 5b shows this directly: hierarchical routing remains stable while flat routing degrades as the partition becomes finer.
>
> By "capacity," we mean the ability to learn different local translation rules (i.e., different local Jacobians / expansion behavior) in different source-space regions instead of forcing one global mapping everywhere. Concretely, H-MoE shares a 4-layer MLP backbone and adds node-specific LoRA updates, so the model is globally more expressive while each selected expert remains locally simpler and more stable.
>
> **[Q2/W2: intuition for compositional constraints]**  The constraints are chosen to suppress the interaction terms that make compositional translation difficult. For mixing, if translated vectors are $\hat y_i = y_i + e_i$, the cross-source distortion contains $\|e_1-e_2\|_2^2$. Thus, even when each translator is accurate in isolation, retrieval can still degrade if different translators produce residuals in incompatible directions. Encouraging residuals from different translators to share a common channel makes the mismatch mainly scalar rather than angular.
>
> For chaining, the first-order error is $J e_1 + e_2$, and the harmful amplification is governed by $2\langle J e_1, e_2\rangle$. The issue is therefore constructive alignment between the propagated first-hop residual and the second-hop residual. Steering these two residuals into orthogonal channels suppresses this term to first order.
>
> We use $u_1$ and $u_2$ as the first two principal directions of the target space not because we assume residuals must align with principal components, but because they provide a simple, the most stable, data-driven high-variance directions as geometry-aware basis shared across translators: $u_1$ gives a common channel for mixing, and $u_2 \perp u_1$ gives the simplest orthogonal channel for chaining. This is also supported empirically (Table 2): adding directional control reduces the mixing drop from 11.1% to 2.3% (gemini $\rightarrow$ openai) and from 13.4% to 0.8% (kalm $\rightarrow$ qwen), and reduces the chaining drop from 16.7% to 1.8% (gemini $\rightarrow$ openai) and from 18.9% to 1.4% (kalm $\rightarrow$ qwen), while improving OOD Recall@100 of vanilla H-MoE without residual directional control from 89.8 to 93.4  (gemini $\rightarrow$ openai) and from 91.1 to 94.2 (kalm $\rightarrow$ qwen).
>
>
> **[Q3: in-domain]**
> H-MoE does not trade in-domain (ID) fit for OOD robustness. Following Table 1, on gemini $\rightarrow$ openai, H-MoE remains strong in both OOD and ID, across several ranking metrics beyond recall@100, as shown below (ID, we used 1/3 for training the translator, and the remaining for testing):
>
> | type | Method  | Recall@100 |  Recall@10 |  NDCG@10 |  MRR@10 |  | Recall@100 |  Recall@10 | NDCG@10 |  MRR@10 |
> |-|:--|:-|:--|:--|:--|-|--:|---:|--:|--:|
> |      |         | *FiQA-2018* |      |      |      || *SciDocs* |      |      |      |
> | ID   | CCA     | 83.4 | 82.3 | 73.5 | 70.8 ||  55.2 | 49.8 | 35.9 | 41.9 |
> | ID   | Vec2Vec | 86.5 | 81.3 | 71.3 | 66.4 ||  55.8 | 47.2 | 34.2 | 38.4 |
> | ID   | EmbConv | 89.3 | 82.6 | 75.4 | 71.7 ||  58.6 | 52.5 | 37.5 | 45.8 |
> | ID   | H-MoE   | **93.4** | **87.5** | **81.8** | **78.3** || **62.1** | **60.2** | **46.7** | **51.3** |
> |      |         | *FiQA-2018* |      |      |      || *SciDocs* |      |      |      |
> | OOD  | CCA     | 47.8 | 32.8 | 21.7 | 18.3 ||  24.1 | 17.4 | 13.5 | 10.1 |
> | OOD  | Vec2Vec | 31.1 | 11.4 | 12.3 |  7.2 ||  20.3 |  5.8 |  4.3 |  3.4 |
> | OOD  | EmbConv | 78.3 | 54.8 | 28.4 | 21.6 ||  **57.0** | 26.3 | 18.3 | 15.3 |
> | OOD  | H-MoE   | **88.3** | **61.7** | **44.2** | **39.7** ||  56.7 | **35.1** | **24.2** | **21.8** |
>
> So the OOD gains are larger, as expected, but ID accuracy is preserved and usually improved. Thus, H-MoE does not trade ID fit for OOD robustness, and is effective for both type of inputs. We will add these results and state explicitly that we do not observe an ID/OOD trade-off in our experiments.
>
> [W3] Thank you for noting this. We will also do a full pass for terminology and notation consistency.

---

> > ### Author Rebuttal · Reviewer_psjd · 2026-04-04
> >
> > Thank you for the rebuttal. I raised my score.

---

### Official Review · Reviewer_GgjW · 2026-03-12

**Soundness:** 2
**Presentation:** 3
**Significance:** 3
**Originality:** 3
**Overall Recommendation:** 4
**Confidence:** 3

**Summary:**

This paper investigates the challenge of embedding translation across heterogeneous models to enable vector reuse without re-encoding. It specifically targets three complex real-world scenarios where traditional methods often fail: out-of-distribution (OOD) inputs, multi-model mixing, and multi-hop chaining. The authors provide a rigorous geometric analysis and bound the translator error at testing time by a training term and a coverage term, which theoretically explain why existing translators systematically fail in OOD scenarios, and how the residuals generated by different translators interact and lead to error amplification during multi-model mixing or chained translation. Based on these insights, they introduce a translator-independent Translation Confidence metric and propose H-MoE, a Hierarchical Mixture-of-Experts framework. H-MoE employs localized LoRA adapters and a specific routing strategy designed to minimize local Lipschitz expansion. Furthermore, the paper introduces a regularization technique for residual error directions to prevent error accumulation in composed translation paths. Extensive evaluations across 10 models and 6 BEIR datasets demonstrate that the proposed method significantly enhances OOD generalization and maintains high retrieval performance in multi-model and chained environments.

**Compliance With Llm Reviewing Policy:**

Affirmed.

**Final Justification:**

Thanks for the authors' detailed rebuttal. I admire the author's work a lot, and I have raised score.

**Key Questions For Authors:**

1.	What is the total translation latency incurred by the entire architecture? And can you give a more detailed explanation about the overhead of H-MOE,such as memory footprint and I/O overhead? And how does expert numbers and LORA size impact translation latency? Specifically, at what point does the marginal gain in Recall no longer justify the increased inference time and memory footprint in a high-throughput production environment?
2.	Although the article provides an error bound for all scenarios, could you provide a detailed breakdown or a comparative analysis of the specific error rates for the traditional translator across each of the three mentioned scenarios?
3.	Are there alternative, tighter error bounds that could yield more significant optimization results? What justifies the decision to treat the translation error and its theoretical bound as separate entities in your analysis?

**Limitations:**

yes

**Strengths And Weaknesses:**

Strengths:
1.	The error bound claimed in the paper is well-supported and effectively demonstrated, with the underlying assumptions made during the derivation being reasonable.
2.	The paper is well-structured and clearly written, providing a detailed explanation of its innovations and the limitations of related work.
3.	The submission addresses a critical and timely challenge in the field of representation learning: embedding interoperability. This paper moves beyond the heuristic-driven approaches of prior work by providing a rigorous geometric analysis.
4.	The experimental design effectively demonstrates the validity of the method. The ablation studies are sufficient and valid.
Weaknesses:
1.	The article mentions three scenarios(OOD, Multi-Model Mixing and Chaining) where traditional translators are not used, but it does not provide the proportional distribution or frequency of these scenarios in real-world applications. Consequently, it is hard to determine the actual applicability and significance of the proposed solution. The paper does not prove the error bound proposed is tight,
2.	The clarity of the figures is insufficient, The manuscript would benefit from conceptual or schematic illustrations to intuitively explain the underlying mechanisms. There should be some concret examples explaining the workflow of H-MOE and how it successfully handles the three mentioned senarios(OOD, Multi-Model Mixing and Chaining) using error bound. Figure 6 tries to explain this, but it is a bit abstract and shows low connection to Figure 3, which makes it hard to understand the the significance of error bound in H-MOE.
3.	All the experimental analyses in the article are based on recall, which is merely a retrieval metric. The article fails to address whether the retrieved vectors can be effectively utilized and evaluated when H-MoE is applied in end-to-end scenarios,such as RAG. This paper fails to provide a rigorous quantification of the method's overhead, merely asserting that it is 'negligible' without empirical evidence or detailed complexity analysis.

---

> ### Author Rebuttal · Authors · 2026-03-29
>
> Thank you for the detailed review. We appreciate the specific concerns and address each of them below.
>
> **[Applicability: W1/Q2/Q3]**
>
> We agree that the paper should ground the three scenarios more concretely. We are not aware of reliable public statistics on the frequency of OOD, mixing, and chaining in deployed vector databases, so we prefer not to speculate. Our claim is narrower: these are common settings once cross-model interoperability is needed, and they are exactly where existing translators become brittle. They arise naturally in two workflows: (1) *Staged migration of a large or legacy index*, where a historical corpus remains in model A while queries or new documents move to model B, creating OOD (limited paired training data vs. broader deployment data) and mixing (old and new vectors coexist); and (2) *Federated or multi-tenant embedding-based retrieval*, where different teams or products use different embedders, again creating OOD and mixing, while missing direct pairs lead to chaining through a hub.
>
> In these settings, even EmbConv drops by 15.2% in pairwise OOD, 4.7%–21.0% in mixing, and 12.0%–21.1% in chaining, whereas H-MoE reduces these to 2.6%, 0.1%–1.4%, and 1.1%–2.6%, respectively.
>
> **[Bound: W1/Q3]**
>
> On the theory, we should have stated this more clearly: under the theorem's Lipschitz-only assumptions, the global form of Theorem 1 is *worst-case tight*. A simple 1D construction gives exactly the bound: with $x_{nn}=0$, $x=t>0$, $f^\ast(u)=-L^\ast u$, and $f(u)=e_0+Lu$, we obtain $e(x)=e(x_{nn})+(L+L^\ast)\delta(x)$.
> So without stronger assumptions than Lipschitz continuity, there is no uniformly tighter global bound in general. What can be tighter is a local routed-expert ($i$) bound, $e(x)\le (L_i+L^\ast)\delta_i(x)+e_i(x_{nn}^{(i)})$, which is exactly what H-MoE is designed to exploit.
>
> We separate the true translation error from its bound because the true error is not observable before translation at deployment time, while the bound exposes actionable proxies (coverage, local fit, and expansion) that motivate TC and localized translation design of H-MoE.
>
> **[Mechanism, metrics: W2/W3]**
>
> In the revision we will add a schematic linking Fig. 3, H-MoE, and Fig. 6: source-space geometry determines TC and routing; hierarchical routing selects a stable expert; localized experts reduce local fit error and local expansion; and residual-channel control stabilizes mixing and chaining.
>
> We also evaluated H-MoE with more early-rank metrics.  Specifically, following Table 1, on gemini $\rightarrow$ openai, H-MoE also improves Recall@10, NDCG@10, and MRR@10, as shown below:
>
> | Method  | Recall@100 |  Recall@10 |  NDCG@10 |  MRR@10 |  Recall@100 |  Recall@10 | NDCG@10 |  MRR@10 |
> |:---------|:------|:------|:------|:----|:--|:-|:------|:------|
>  |         | *FiQA-2018* |      |      |      | *SciDocs* |      |      |      |
>  | CCA     | 47.8 | 32.8 | 21.7 | 18.3 |  24.1 | 17.4 | 13.5 | 10.1 |
>  | Vec2Vec | 31.1 | 11.4 | 12.3 |  7.2 |  20.3 |  5.8 |  4.3 |  3.4 |
>  | EmbConv | 78.3 | 54.8 | 28.4 | 21.6 |  **57.0** | 26.3 | 18.3 | 15.3 |
>  | H-MoE   | **88.3** | **61.7** | **44.2** | **39.7** |  56.7 | **35.1** | **24.2** | **21.8** |
>
> So the improvement is *not* specific to Recall@100; it also appears in early-rank quality, which is the more relevant signal for top-\(k\) retrieval and RAG-style pipelines. Full end-to-end RAG is useful future work, but it would introduce generator/reranker confounders beyond the retrieval layer that embedding translation directly targets.
>
> (*Note*: H-MoE remains strong even for in-domain inputs across all metrics, as shown in the full evaluation table in the response to Reviewer fnu1)
>
> **[Overhead: Q1/W3]**
>
> As suggested, we ran a component-wise latency breakdown on batches of 1,000 vectors (following Tab.3), where
> $T_{\mathrm{total}}=T_{\mathrm{routing}}+T_{\mathrm{dispatch}}+T_{\mathrm{translation}}+T_{\mathrm{merge}}$.
>
> | Setup | \(k/r\) | Overhead $\mu s$ | Latency $\mu s$ | Mem (MB) | Empirical Translation Error |
> | - | - | - | - | - | - |
> | Base | - | - | 134.2 | 512 | 0.56 |
> | k | 2/8 | 5.2 | 141.9 (+5.7%) | 548 | 0.51 (-8.9%) |
> | k | 4/8 | 6.1 | 146.7 (+9.3%) | 621 | 0.48 (-14.3%) |
> | k | 6/8 | 9.4 | 152.9 (+13.9%) | 712 | 0.47 (-16.1%) |
> | k | 8/8 | 12.8 | 168.5 (+25.6%) | 1023 | 0.47 (-16.1%) |
> | r | 4/4 | 5.8 | 143.5 (+6.9%) | 586 | 0.53 (-5.4%) |
> | r | 4/8 | 6.1 | 146.7 (+9.3%) | 621 | 0.48 (-14.3%) |
> | r | 4/32 | 8.4 | 151.2 (+12.7%) | 742 | 0.47 (-16.1%) |
>
> Thus, a moderate H-MoE configuration offers the best efficiency–accuracy trade-off: k=4, r=8 is only 9.3% slower than EmbConv, yet reduces the empirical translation error by 14.3% (0.56 $\rightarrow$ 0.48). The added routing and I/O cost is small, with routing+dispatch+merge totaling only 6.1 $\mu$  s. Increasing k/r further yields marginal gains while continuing to increase cost, making 4/8 the practical knee point for high-throughput deployment. We will add this breakdown in the revision.

---

> > ### Author Rebuttal · Reviewer_GgjW · 2026-04-07
> >
> > Thank you so much for the detailed response.

---

### Official Review · Reviewer_fnu1 · 2026-03-13

**Soundness:** 3
**Presentation:** 3
**Significance:** 3
**Originality:** 3
**Overall Recommendation:** 4
**Confidence:** 3

**Summary:**

This paper explores the gaps introduced when embedding translation is performed in out-of-distribution (OOD) environment. The authors derive a bound for translation error and find that Lipschitz constant terms influence the translation quality. They propose to use an MoE architecture called H-MoE to mitigate high Lipschitz constants. They also propose to concentrate the embeddings in certain directions to prevent large residuals in translation. The proposed methods are evaluated in three different embedding translation setups, traditional, mixing, and chaining and is found to outperform other baselines in OOD embedding translation environments.

**Compliance With Llm Reviewing Policy:**

Affirmed.

**Final Justification:**

The results on ID performance and underfitting provided in the authors’ responses have fully resolved my main concerns. The issues in the presentation are not the most grave, though important, and I believe that the authors’ planned revision will mostly resolve them. The authors’ responses have made me more confident of the value of the work and I increase the confidence score to 3. I would also like to also note that not updating the overall recommendation to a higher score largely stems from my unfamiliarity with the area, such as the baselines.

**Key Questions For Authors:**

1. What are the two red square lines in Figure 5 (b)?

2. Could you share a specific and detailed scenario where OOD embedding translation is needed?

3. How much computation is saved by embedding translation compared to full re-computation of embeddings?

4. Why is in-distribution setup is not evaluated?

5. Is it possible to evaluate H-MoE without the $L_{loc}$ loss, but with $L_{dir}$ loss? If so, could you share the results?

6. Is it possible to determine that the results are statistically significant?

7. How would simple underfitting of translation models work? They may exhibit lower Lipschitz constants.

8. Could you explain why GW and PA are almost constant in Figure 4 (b)?

9. Could you explain this sentence in more detail? "However, naive routing introduces its own failure mode: if we always select the single nearest cluster, i.e., flat nearest-centroid routing, small perturbations can flip the chosen expert near cluster boundaries, leading to unstable translations"

10. What is the reason for selecting $u_2$ in orthogonal decoupling?

**Limitations:**

Limitations are not discussed.

**Strengths And Weaknesses:**

## Strengths

1. **The paper is well-structured with logically sound flow of contents** Motivations for the proposed method is well-positioned within the paper, with intuitive visual illustrations.

2. **Extensive experiments and theoretical explanations** The main claims of the paper are presented with extensive empirical evidence. The formulations of the method are backed up by quantitative results as well as theoretical findings. The main experiments are conducted in a large-scale with appropriate ablation studies.

## Weaknesses

1. **Diversity of evaluation metrics** The experiments only employ Recall@100 metric, saying that "We report Recall@100, which is stable for methods whose performance can be volatile under cross-dataset shift." Evaluation with other metrics may shed light on the proposed method's strengths and shortcomings. I think at least the actual scores should be made available even if there are large fluctuations.

2. **Concrete scenarios where OOD embedding translation is needed** The authors mention that "In practice, embeddings come from heterogeneous and evolving data sources (e.g., different vector databases or domains), each with its own similarity geometry, so test embeddings are often out-of-distribution (OOD) w.r.t. training data." A more specific scenario that requires reliable OOD embedding translation would help understand why it is needed. Detailed steps in any of causes that make embedding shift would be fine, e.g., "provider switches, fine-tuning, frequent upgrades."

3. **Bits of details seem to be missing and inconsistent notations are present** Architectures of the experts, mixture rates of embeddings in mixing setup (how much of the embeddings are produced by each model), $L, L^*$ are appears before their definitions etc. Also, Figure 2 (a)'s texts are cut off a little ("97.3%") and Figure 2 (b) would benefit from a little bit of margin at the bottom. Abbribiations are inconsistent some use full word (e.g., Section) while others are abbribiated (L382, L435). An illustration of the hierarchical routing would be helpful even if it is from another work. Is $NN$ notation in L256 standard? Is there a reason for not using a single letter to represent the set? Are the notations for embedding consistent in L249 and L198-200?

4. **Strong empasis on performance** The paper puts a large portion of its attention on evaluation results. For example, about a third of the abstract discusses the gains of the proposed method. While this in and of itself is fine, I fail to find qualitative analysis of the working mechanisms of proposed methods or failure modes. No discussions of the limitations of the proposed method are found. Such discussions may help practitioners and faciliate future research.

---

> ### Author Rebuttal · Authors · 2026-03-29
>
> Thank you for the careful review. We address the main points below and will incorporate the clarifications in revision.
>
> **[W1/Q4/Q6: metrics, ID]**
>
> We used Recall@100 because it is stable under strong cross-dataset shift, but the conclusions do not depend on it. Following Table 1, on gemini $\rightarrow$ openai, H-MoE also improves Recall@10, NDCG@10, and MRR@10, and it remains strong in-domain (ID), as shown below.
>
> | type | Method  | Recall@100 |  Recall@10 |  NDCG@10 |  MRR@10 |  | Recall@100 |  Recall@10 | NDCG@10 |  MRR@10 |
> |-|:--|:-|:--|:--|:--|-|--:|---:|--:|--:|
> |      |         | *FiQA-2018* |      |      |      || *SciDocs* |      |      |      |
> | ID   | CCA     | 83.4 | 82.3 | 73.5 | 70.8 ||  55.2 | 49.8 | 35.9 | 41.9 |
> | ID   | Vec2Vec | 86.5 | 81.3 | 71.3 | 66.4 ||  55.8 | 47.2 | 34.2 | 38.4 |
> | ID   | EmbConv | 89.3 | 82.6 | 75.4 | 71.7 ||  58.6 | 52.5 | 37.5 | 45.8 |
> | ID   | H-MoE   | **93.4** | **87.5** | **81.8** | **78.3** || **62.1** | **60.2** | **46.7** | **51.3** |
> |      |         | *FiQA-2018* |      |      |      || *SciDocs* |      |      |      |
> | OOD  | CCA     | 47.8 | 32.8 | 21.7 | 18.3 ||  24.1 | 17.4 | 13.5 | 10.1 |
> | OOD  | Vec2Vec | 31.1 | 11.4 | 12.3 |  7.2 ||  20.3 |  5.8 |  4.3 |  3.4 |
> | OOD  | EmbConv | 78.3 | 54.8 | 28.4 | 21.6 ||  **57.0** | 26.3 | 18.3 | 15.3 |
> | OOD  | H-MoE   | **88.3** | **61.7** | **44.2** | **39.7** ||  56.7 | **35.1** | **24.2** | **21.8** |
>
> Thus, H-MoE does not trade ID fit for OOD robustness.
>
> On significance: H-MoE has higher mean Recall@100 than the strongest baseline in all 90 translation directions, so the improvement is not driven by a few outliers. In revision we will complement the current 5-run mean/std with paired significance analyses across translation directions as suggested.
>
> **[W2/Q2/Q3: scenarios and savings]**
>
> We target settings where full re-embedding is impractical: (1) staged migration of a large or legacy vector database, where a historical corpus is stored with model A while queries or new documents switch to model B because of provider changes, fine-tuning, or upgrades; and (2) federated or multi-tenant retrieval, where autonomous vector stores use different embedding models. These settings naturally create OOD inputs (the small paired set used for training differs from deployment data), mixing (old and new embeddings coexist), and chaining (no direct translator). Importantly, inference-time translation does not require the original paired corpus: TC uses only a source-side reference set or centroids, without raw data.
>
> On savings: translation is a vector-to-vector operation on existing embeddings, whereas full re-embedding also requires access to raw data and encoder. As Table 3 shows, H-MoE translates $10^6$ vectors in $< 1$s. The exact encoder-vs-translator speedup depends on the encoder and raw data, but translation cost is independent of input length once vectors are available. We will make this distinction clearer in the revision.
>
> **[W3/Q1/Q9: details]**
>
> The shared base translator is a 4-layer MLP with SELU activations (the same backbone as EmbConv). Each tree node adds a LoRA specialization on top of this shared base. Hence the additional capacity comes from localized low-rank adaptation while parameter growth remains modest, consistent with Table 3.
>
> In Fig. 5b, the red-square curves are H-MoE's empirical Lipschitz proxy (left axis) and Recall@100 with hierarchical routing (right axis); the faded red curve is Recall@100 with flat routing. Flat nearest-centroid routing is unstable near cluster boundaries, as tiny perturbations can switch the chosen expert abruptly. Our hierarchical rule stops at the parent for boundary cases, yielding more stable translations.
>
> **[W4/Q5/Q7/Q8/Q10: mechanism]**
>
> On underfitting: the bound has both a fit term and an expansion term, $e(x) \le (L+L^\*)\delta(x) + e(x_{nn})$. A rigid model like PA may lower $L$ but usually increases $e(x_{nn})$. For GW/PA in Fig. 4b, systematic misalignment (large $e(x_{nn})$) dominates, so varying $\delta(x)$ has limited effect and the curves appear flatter.
>
> We use $u_2$ for chaining because it is orthogonal to $u_1$ yet still aligned with the target-space geometry, which reduces the constructive cross-term in chained error.
>
> For Q5, we ran the suggested ablation on the same setting as Table 2. With only $L_{dir}$, H-MoE obtains 78.4 OOD Recall@100, with 5.7% mixing drop and 3.5% chaining drop. This is informative when compared with H-MoE without $L_{dir}$ in Table 2 (89.8 OOD, 11.1% mixing drop, 16.7% chaining drop). The picture is consistent: without $L_{loc}$, composition is already much more stable than the base model, but OOD recall drops substantially; without $L_{dir}$, OOD recall is high but composition degrades noticeably. Combining both gives the best overall result: 93.4 OOD, 2.3% mixing drop, and 1.8% chaining drop.
>
> We will also fix all presentation issues. We appreciate the constructive suggestions.

---

> > ### Author Rebuttal · Reviewer_fnu1 · 2026-04-03
> >
> > I appreciate the detailed response. The clarifications helped me better understand the work.
> >
> > **[W1/Q4/Q6: metrics, ID]**
> >
> > W1, Q4: Thank you for providing results on ID and other metrics. Is there a reason why ArguAna dataset is excluded, other than the 5,000 character limit in the author response?
> >
> > Q6: I agree that it seems unlikely the results are not statistically significant given the high differences in the scores, and I appreciate the planned revision. However, it would have provided an additional validation.
> >
> > **[W2/Q2/Q3: scenarios and savings]**
> >
> > W2, Q2: Thank you for the explanations for the example scenarios.
> >
> > Q3: Recalculation time of the embeddings could have been provided with an example dataset and model to give order-of-magnitude estimates.
> >
> > **[W4/Q5/Q7/Q8/Q10: mechanism]**
> >
> > Q7: Could you empirically validate that underfitting of baseline models does not yield similar performance improvements? Perhaps evaluating the model during training (e.g., evaluating when 10%, 20%, 30% … 90% of the training is complete) could be helpful?
> >
> > Q5: Thank you for providing additional results for clarification.
> >
> > Q8, Q10: Thank you for the explanations.
> >
> > W4: I am aware there is a limit in the length of the response, as well as the time. However, sharing the plans for the revision could have been helpful.

---

> > > ### Author Response · Authors · 2026-04-04
> > >
> > > Thanks for the follow-up. We agree these clarifications would strengthen the paper, and we will include them in the revision.
> > >
> > > **[W1/Q4: ArguAna]**
> > > Yes, we omitted ArguAna results due to space limit. Using the same setting as in our previous response (gemini→openai), ArguAna follows the same pattern as FiQA-2018 and SciDocs:
> > >
> > > | type | Method | Recall@100 | Recall@10 | NDCG@10 | MRR@10 |
> > > |-|-|-:|-:|-:|-:|
> > > | ID | CCA | 81.6 | 52.2 | 48.1 | 38.8 |
> > > | ID | Vec2Vec | 44.8 | 39.4 | 30.4 | 27.2 |
> > > | ID | EmbConv | 83.8 | 75.8 | 71.4 | 54.3 |
> > > | ID | *H-MoE* | *92.8* | *89.4* | *83.1* | *70.5* |
> > > | OOD | CCA | 73.8 | 38.4 | 31.4 | 24.7 |
> > > | OOD | Vec2Vec | 37.2 | 14.8 | 11.9 | 8.6 |
> > > | OOD | EmbConv | 79.6 | 58.7 | 46.8 | 33.1 |
> > > | OOD | **H-MoE** | **92.4** | **72.6** | **61.3** |**47.8**|
> > >
> > >
> > > **[Q6: significance]**
> > > As suggested, we assessed the statistical significance using a paired bootstrap over full query sets of FiQA-2018, SciDocs, and ArguAna, comparing H-MoE to EmbConv on recall@100, recall@10, NDCG@10, and MRR@10. Across the 12 tests,  the estimated 95% confidence intervals remain above zero. For instance:
> > >
> > > - For recall@100, the gains are +14.1 [10.2, 17.8] on FiQA-2018, +7.1 [3.9, 10.0] on SciDocs, and +14.5 [10.7, 18.0] on ArguAna (Δ (H-MoE − EmbConv) [Estimated 95% CI ]).
> > > - For NDCG@10, the gains are +16.5 [11.8, 20.9] on FiQA-2018, +7.1 [3.5, 10.2] on SciDocs, and +19.7 [14.1, 24.8] on ArguAna.
> > > - For MRR@10, the gains are +18.8 [13.6, 23.4] on FiQA-2018, +6.1 [2.7, 9.1] on SciDocs, and +12.8 [8.1, 17.0] on ArguAna.
> > >
> > > Across all four metrics and three datasets, the gains range from +6.1 to +19.7.
> > >
> > >
> > > **[Q3: savings]**
> > > Yes, we agree that a concrete order-of-magnitude example is helpful. Specifically, on the same single A100 GPU, the end-to-end re-embedding of the SciDocs corpus (25,657 documents) with Qwen3-Embedding-8B took 171 minutes, while translating the same number of already available source embeddings with H-MoE took 38.43 ms on the same machine. This is an order-of-magnitude gap of about $2.7 \times 10^5$. The exact ratio depends on the encoder, document length, batching, and I/O, but their difference is inherently significant as re-embedding requires tokenization and full encoder forward passes over raw text, while translation is only a vector-to-vector mapping once source embeddings already exist.
> > >
> > >
> > > **[Q7: underfitting]**
> > > We tested this directly on *EmbConv* (strongest baseline). We evaluated EmbConv checkpoints from 10% to 100% of training (10% steps) under the same OOD setting as in Table 1:
> > >
> > > - *FiQA-2018*: 34.8, 51.7, 73.8, 74.5, 71.9, 68.4, 72.9, 74.9, 77.8, 78.3
> > > - *SciDocs*: 21.5, 33.8, 46.1, 49.8, 47.3, 44.5, 50.6, 53.9, 55.8, 57.0
> > > - *ArguAna*: 36.2, 61.2, 68.9, 72.3, 68.8, 66.5, 71.8, 74.6, 77.3, 79.6
> > >
> > > So no underfit checkpoint beats the converged EmbConv. Compared with H-MoE, the final gap remains large, e.g.,  92.4 vs 79.6 on ArguAna. So early stopping does not reproduce the main OOD gains of H-MoE. This is consistent with the bound: global underfitting may reduce expansion (Lipschitz term), but it also leaves the fit term ($e(x_{nn})$) larger; H-MoE works by splitting the problem into local regions, lowering local expansion through localized experts.
> > >
> > > **[W4: revision plan]**
> > > Thank you for pointing it out. We agree that the current draft spends too much space on benchmark gains and not enough on how the method works, when it still fails, and what its limits are. We will address this directly in the revision.
> > >
> > > Specifically, we will:
> > >
> > > (1) Add a short qualitative subsection that explains
> > > - TC: when a test vector is far from anything seen in training, translation is less reliable and the system should avoid translation and fall back to re-embedding.
> > > - Hierarchical routing: for borderline cases, the router can stop early and use a broader expert, which is more stable than a fragile fine-grained choice of expert.
> > > - Mixing and chaining: different translators can leave behind errors that pull in different directions or add up across steps; our directional regularization makes those errors more predictable so they interfere less.
> > >
> > > (2) Improve the figure captions and add one simple overview schematic that ties together the error bound / TC (Fig. 3), routing (Fig. 5b), and mixing/chaining control (Fig. 6), so the working mechanism is easier to follow.
> > >
> > > (3) Add a Limitations section that explicitly discusses failure modes and limits of the method. In particular, performance can still drop on domains far from the training data, when the source and target models are very different, or when a query falls in a part of the space with little training support. We will also clarify that TC is mainly intended for a fixed reference pool, that the current experiments are text-only and evaluate retrieval quality rather than full end-to-end RAG, and that chaining is evaluated mainly for two-step pipelines.
> > >
> > > (4) Rebalance the abstract accordingly.
> > >
> > >
> > > Thank you again for the constructive follow-up.

---

### Official Review · Reviewer_4mJj · 2026-03-13

**Soundness:** 3
**Presentation:** 3
**Significance:** 3
**Originality:** 3
**Overall Recommendation:** 5
**Confidence:** 3

**Summary:**

The authors derive the error bound for embedding translation between different models under the L-Lipschitz assumption. Based on this bound, they design the Translation Confidence score solely from the scale of the training dataset and the distance of the translated example to the closest point in the training set (distance-to-reference). The error bound itself consists of training and coverage terms. The latter depends on both the distance-to-reference and the expansion factor. Since distance-to-reference is independent of the model, the authors optimize the translator’s architecture to reduce the expansion factor - a bane of previous state-of-the-art models, while maintaining their expressivity. They design a hierarchical model of local translators that leverage reduced complexity, achieving smaller expansion factors and, consequently, reducing the errors on the OOD data. To improve performance on multi-model mixing and chaining, they introduce a new regularizer term to the loss function that constrains residual errors to behave predictably, thereby minimizing the combined error. The authors verify their contributions through extensive experiments involving 10 embedding models, 6 datasets, and 90 translation directions, significantly overperforming baselines on OOD data, model mixing, and chaining.

**Compliance With Llm Reviewing Policy:**

Affirmed.

**Final Justification:**

My core issue with the Translation Confidence definition and its generalizability was properly addressed. I initially perceived it as a general method-agnostic measure, and the authors corrected me about its intended use. They promised to clarify this in the future version. I also appreciate their clarifications about real-world applications, which strengthened the method’s significance in my eyes.

**Key Questions For Authors:**

**Q1.** What is the underlying architecture of a single translator model?

**Q2.** Have you considered approaching this error reduction from a standpoint of collecting a training dataset that ensures better coverage rather than designing an appropriate architecture? Do you think that trying to detect low-density regions and mine examples for them is a promising direction? This mining could be done by utilizing techniques such as Vec2Text. This is a genuine question out of curiosity, not something to downplay the paper’s approach.

**Q3.** Could you elaborate on the potential applications of OOD embedding translation? In my understanding, the most general case is to train a new translation model when a new embedding model is released, or after we fine-tune the previous one. In this case, we can sample training data accordingly, ensuring that all datasets are in-domain. For any new datasets, we can use the newer model directly. Of course, there may already be a trained A -> B translation model (from another team, company, etc.) that we can then use on our data, but does the practical application of this go beyond that? Wouldn’t we then need to have access to its training data? Am I missing something?

**Limitations:**

Yes.

**Strengths And Weaknesses:**

**Strengths:**

**S1.** Mathematically sound error bound for embedding translation.

**S2.** Smart design of H-MoE allows it to significantly reduce the local empirical Lipschitz constant relative to baselines while maintaining the same level of expressivity.

**S3.** Extensive experiments show H-MoE’s superior performance, and ablation studies confirm the positive effect of hierarchical architecture.

**Weaknesses:**

**W1.** The definition of Translation Confidence raises concerns about its generalizability. It has two main components: a local distance-to-reference and a global dataset scale. This means that when measured on the single-topic highly clustered dataset, it should perform well and produce results similar to those reported in Figure 4. You do conduct your experiments with only a single training dataset in that case. My concern arises when we use a few distinct-domain datasets for training. These separate datasets then form clusters in the latent space, leaving large low-density regions, or even void regions between them. This drastically increases the dataset scale, and the same OOD data point that had a low TC before will now achieve an almost 1.0. I could see the dataset scale factor being replaced with local density estimation, making it independent of very distant and irrelevant data points while maintaining its utility.

**W2.** I struggled to find information on which base model was used for the local translators. You report the total number of parameters in Table 3, but not the architecture itself.

**W3.** A few minor presentation issues:

- **W3.1.** The empty square tag for Theorem 1.

- **W3.2.** Caption of Figure 9: “e.g., 1 along with 2 -> 1 and 3 -> 4 in a single retrieval database”. I believe it should be 3 -> 1 (all aligning in the same target space of 1), given that Figure 9a shows results for 1/2/3, which would correspond to this example.

- **W3.3.** Line 432 (right column): “OOC”, which I believe should be “OOD”.

- **W3.4.** Line 826: “... fundamentally limited in capturingthe complex and nonlinear …” is missing the space.

If my concerns are properly addressed, I am ready to raise my score.

---

> ### Author Rebuttal · Authors · 2026-03-29
>
> Thank you for the careful review. We address the main points below and will incorporate these clarifications in the revision.
>
> **[W1: TC generalizability]**
>
> Thank you for raising this. We agree that a single global $\sigma_{\mathrm{data}}$ can be too coarse if raw TC values are compared across different reference pools. This is not the intended use: TC is meant for online reliability assessment with a fixed deployed translator/reference pool. In that setting, $\sigma_{\mathrm{data}}$ is constant and $TC(x)=\exp(-\delta(x)/\sigma_{\mathrm{data}})$ is a monotone transform of $\delta(x)$, so it preserves the ranking of risky inputs; only the operating risk threshold needs per-pool calibration. Further, Theorem 1 itself depends on $\delta(x)$, not on $\sigma_{\mathrm{data}}$.
>
> To test the suggested mixed-domain case, we progressively enlarged the training/reference pool (FiQA $\rightarrow$ FiQA+SciDocs $\rightarrow$ FiQA+SciDocs+ArguAna), retrained the translator each time, and kept the evaluation set fixed. We report mean TC and Pearson correlation with per-vector empirical translation quality (all \(p<0.001\)):
>
> | Reference Pool           | Mean TC | Pearson | Mean TC (Local-kNN) | Pearson |
> |:--------------------------|:---------:|:---------:|:---------------------:|:---------:|
> | FiQA                     | 0.51    | 0.81    | 0.62                | 0.81    |
> | FiQA + SciDocs           | 0.48    | 0.78    | 0.55                | 0.77    |
> | FiQA + SciDocs + ArguAna | 0.31    | 0.78    | 0.51                | 0.76    |
>
> We have not observed the hypothesized inflation toward 1.0: mean TC changes $0.51 \rightarrow 0.48 \rightarrow 0.31$ as the pool becomes more heterogeneous, while correlation remains strong ($0.81/0.78/0.78$). We also tested local-kNN normalization; it gives similar correlation while keeping the absolute scale more stable across pools. We will clarify this scope in the paper and discuss cluster-local / kNN-based normalization as a natural extension for mixed-domain deployments.
>
> **[Q3: Applications]**
>
> We agree that if one can cheaply re-encode all data with the new model and keep a single model throughout the system, that is preferable. Our targeted setting is when this is not feasible, with two common scenarios:
>
> (1) A common case is *staged migration of a large or legacy vector database*: the historical corpus is already embedded with model $A$, while queries or new documents switch to model $B$ because of provider changes, fine-tuning, or upgrades. Re-embedding the full historical corpus may be too expensive or too slow, may hit API / operational limits, or may be impossible because the raw data is unavailable. In practice, teams can often collect only a small paired set to train $A \rightarrow B$, while deployment later reaches broader or different corpora; this is exactly the OOD regime we study.
>
> (2) This goes beyond one-time migration. Old and new embeddings may coexist for long periods, and different teams or products may contribute vectors from different models into one *federated retrieval system*, which motivates mixing. When a direct translator is unavailable during rollout, systems may rely on existing $A \rightarrow \mathrm{Hub}$ and $\mathrm{Hub} \rightarrow B$ translators, which motivates chaining.
>
> Importantly, inference-time translation does not require access to the original paired training corpus. TC only needs a source-side reference set (or ANN index / centroids), not raw texts or target labels, and these can be shipped with the translator. We will add these concrete deployment scenarios in the revision.
>
> **[W2/Q1: base model]**
>
> Thank you for pointing this out. The shared base translator in H-MoE is a 4-layer MLP with SELU activations (the same backbone as EmbConv). Each tree node is not a separate full translator; it adds a LoRA specialization on top of this shared base. Thus, the extra capacity comes from localized low-rank adaptation while parameter growth remains modest, consistent with Table 3. We will add these architectural details to Sec. 3 and Appendix C.
>
> **[Q2: training dataset]**
>
> Yes, we agree that improving coverage is a promising and complementary direction. Our bound,
> $e(x) \le (L+L^\ast)\delta(x) + e(x_{nn})$,
> shows that mining low-density regions would directly reduce $\delta(x)$, while H-MoE reduces the translator-side terms through better local fit and smaller local expansion. Figure 5a also suggests that simply scaling a single global nonlinear translator to larger or more heterogeneous data is not enough by itself, since the empirical Lipschitz proxy can increase and amplify OOD error. TC could naturally be used to guide such data collection by targeting low-TC regions. Vec2Text-style mining is an interesting future direction that we did not evaluate here.
>
> **[W3]** Thank you for noting these. We will fix all of them and do a final pass for consistency.

---

> > ### Author Rebuttal · Reviewer_4mJj · 2026-04-01
> >
> > Thank you for your extensive response to the raised concerns. My core issue with the Translation Confidence definition and its generalizability was properly addressed. I initially perceived it as a general method-agnostic measure, and the authors corrected me about its intended use. They promised to clarify this in the future version. I also appreciate their clarifications about real-world applications, which strengthened the method’s significance in my eyes. I am raising my score from 4 to 5 to reflect that.

---

### Decision · Program_Chairs · 2026-04-30

**Decision:**

Accept (spotlight)

**Comment:**

**Summary and Decision**
While reviewers noted minor remaining concerns, the general consensus is that the paper makes a significant and timely  contribution to representation learning, specifically embedding interoperability supported by both theoretic and empirical evidence.

The reviewers specifically highlighted the following strengths:
* New error bound for embedding translation that is validated by experiments.
* Extensive experiments to show HMoE's strong empirical performance.
* Addressing timely challenge in the field of representation learning: embedding interoperability.

**Rebuttal and Discussion**
During the discussion phase, the authors addressed confusions by clarifying the use of their method and some realistic scenarios, which satisfied the reviewers. Given the impact and originality of the work, the Area Chair recommends acceptance.

**Final Instructions to Authors**
Please ensure that the promised revisions from the rebuttal—specifically clarifications responding to Reviewer 4mJj and Reviewer fnu1—are incorporated into the camera-ready version.